# Unified Risk Analysis for Weakly Supervised Learning

**Chao-Kai Chiang**                                                    *chaokai@k.u-tokyo.ac.jp*
**Masashi Sugiyama**                                                       *sugi@k.u-tokyo.ac.jp*
*Department of Complexity Science and Engineering*
*Graduate School of Frontier Sciences*
*The University of Tokyo*
*5-1-5 Kashiwanoha, Kashiwa-shi, Chiba 277-8561, Japan*

**Reviewed on OpenReview:** *https://openreview.net/forum?id=RGsdAwWuu6*

## Abstract

Among the flourishing research of weakly supervised learning (WSL), we recognize the lack of a unified interpretation of the mechanism behind the weakly supervised scenarios, let alone a systematic treatment of the risk rewrite problem, a crucial step in the empirical risk minimization approach. In this paper, we introduce a framework providing a comprehensive understanding and a unified methodology for WSL. The formulation component of the framework, leveraging a contamination perspective, provides a unified interpretation of how weak supervision is formed and subsumes fifteen existing WSL settings. The induced reduction graphs offer comprehensive connections over WSLs. The analysis component of the framework, viewed as a decontamination process, provides a systematic method of conducting risk rewrite. In addition to the conventional inverse matrix approach, we devise a novel strategy called marginal chain aiming to decontaminate distributions. We justify the feasibility of the proposed framework by recovering existing rewrites reported in the literature.

## 1 Introduction

Accurate labels allow one to generalize to unseen data via empirical risk minimization (ERM) and analyze the generalization error in terms of the classification risk. In practice, there are various situations in which acquiring accurate labels is hard or even impossible. One obstacle preventing us from acquiring accurate labels is labeling restrictions, such as imperfect supervision due to imperceptibility, time constraints, annotation costs, and even data sensitivity. Another obstacle is the disruption by unavoidable noise from the environment.

To address the first obstacle of restrictions, various formulations have been studied under the notion of weakly supervised learning (WSL) (Zhou, 2018; Sugiyama et al., 2022). Based on various types of available label information, it evolves to thriving topics, including the conventional settings (Lu et al., 2019; 2020; 2021; Elkan & Noto, 2008; du Plessis et al., 2014; 2015; Niu et al., 2016; Kiryo et al., 2017; Sansone et al., 2019) that investigating the potential of unlabeled data, complementary-label learning (Ishida et al., 2017; 2019; Yu et al., 2018; Feng et al., 2020a; Katsura & Uchida, 2020; Chou et al., 2020), partial-label learning (Cour et al., 2011; Wang et al., 2019; Lv et al., 2020; Feng et al., 2020b; Wu et al., 2023), learning with confidence information (Ishida et al., 2018; Cao et al., 2021a;b; Berthon et al., 2021; Ishida et al., 2023), and learning with comparative information (Bao et al., 2018; Shimada et al., 2021; Feng et al., 2021; Cao et al., 2021b). Developing to resolve the second obstacle of noise, learning with noisy labels (LNL) can be categorized into two major formulations; one is called mutually contaminated distributions (MCD) (Scott et al., 2013; Menon et al., 2015; Katz-Samuels et al., 2019) in which class-conditional distributions contaminate each other, and the other is named class-conditional random label noise (CCN) (Natarajan et al., 2013; 2017) where a label is flipped by random noise.

Despite fruitful results and tremendous impact, we recognize a lack of global understanding and systematic treatment of WSL. From the perspective of *formulation*, there are only scattered links among WSLs. Lu et al. (2019) and Feng et al. (2021) showed that parameter substitution could reduce unlabeled-unlabeled to similar-unlabeled and positive-unlabeled settings. Figure 1 in Wu et al. (2023) showed relationships among four WSLs of partial- and complementary-labels. A similar observation can be found in the intersection of WSLs and LNLs. Several WSLs were shown to be special cases of the MCD model, and some other WSLs are special cases of the CCN model. For details, please refer to the discussions in Sections 8.2.3 and 9.2.4 of Sugiyama et al. (2022). These connections encourage us to consider the possibility that there exists a unique interpretation that explains the mechanism behind WSL. Luckily, from the *methodological* viewpoint, most of the existing WSL research adopted certain forms of the ERM approach. A crucial shared step is to perform the risk rewrite, a way of rephrasing the uncomputable risk to a computable one in terms of the data-generating distributions. A successful rewrite is the starting point of many downstream tasks, including but not limited to the following: Devising a practical or robust objective for training, comparing the strengths and properties of loss functions, proving the consistency, and analyzing generalization error bounds. However, many rewrite forms (summarized in Tables 4 and 5) look independent as if they are tailored to fit each problem's unique form of supervision and are not adaptable to each other. These seemingly non-adaptable estimators post a practical challenge: When facing a new form of weak (or noisy) supervision, we do not have a guideline or general strategy to leverage developed methods to address the new situation.

These observations raise the following questions we aim to answer in this paper: What is the essence of WSL? From a formulation perspective, can a unique interpretation be found to explain the mechanism behind WSL? Does a methodology exist to address as many WSLs as possible?

This paper proposes a framework with the following contributions to answer the research questions.

1. To the best of our knowledge, the framework is the first systematic attempt to address how and why WSLs are related. The framework consists of a formulation component and an analysis component, and subsumes fifteen weakly supervised scenarios. Table 10 summarizes the results obtained from our framework. This paper brings forth the next two new insights.

2. The formulation component, modeling from a *contamination* perspective, provides a coherent interpretation of the weakly supervised data-generating processes. It produces a comprehensive relationship graph, shown in Figure 1, consisting of Tables 7, 8, and 9. Figure1 summarizes the WSLs and reveals connections between scenarios that were previously unknown to the community. Figure 1 is our answer to the second research question. Figure 1 also unveils a distinctive confidence-based type of WSLs that do not belong to the prominent MCD or CCN categories.

3. The analysis component, leveraging the *decontamination* concept, establishes a generic methodology for conducting risk rewrites for all WSLs discussed in this paper. Thus, rewrite derivations that previously seemed irrelevant can now be systematically analyzed, and the final research question is answered. In addition, the analysis component also applies to new scenarios.

4. Regarding the technical contributions, the proposed framework distinguishes two approaches, the inversion method and the marginal chain method presented in Theorems 1 and 2, to implement the decontamination idea. The marginal chain method is a newly developed invertibility-free loss correction approach. We also illustrate the subtle adjustments to develop simplified and intuitive proofs for the existing risk rewrites. These alternative proofs have their own logic stemming from the proposed framework.

The idea of decontamination has been widely implemented and investigated. There are two major approaches, loss correction, and label correction, in LNL. Closest to the current paper, Cid-Sueiro (2012), van Rooyen & Williamson (2017), Katz-Samuels et al. (2019), Patrini et al. (2017), and van Rooyen & Williamson (2015) exploited the inverse matrix, sometimes known as the backward method (Patrini et al., 2017), to construct a corrected training loss to obtain an unbiased estimator. There were deep learning methods leveraging the contamination assumption, sometimes called the forward method (Patrini et al., 2017), to train a classifier (Patrini et al., 2017; Yu et al., 2018; Sukhbaatar & Fergus, 2015; Goldberger & Ben-Reuven, 2017; Berthon

et al., 2021). Besides modifying the loss function, one has two other strategies to manipulate the corrupted labels. The (iterative) pseudo-label method modified the labels for training (Ma et al., 2018; Tanaka et al., 2018; Reed et al., 2015). Filtering clean data points for training is the other option (Northcutt et al., 2017; 2021; Jiang et al., 2018; Han et al., 2018; Yu et al., 2019). Apart from classification, a different research branch studies conditions and methods for recovering the base distributions (Katz-Samuels et al., 2019; Blanchard & Scott, 2014; Blanchard et al., 2016).

The current work is close to the loss correction approach in LNL. Most previous loss correction methods exploited invertibility to construct the corrected losses. In contrast, the marginal chain approach we propose in this paper adopts the conditional probability formula to build the corrected losses. Many of the existing work targeted either the MCD or the CCN models. Scott & Zhang (2020), Berthon et al. (2021), Patrini et al. (2017), Goldberger & Ben-Reuven (2017), Sukhbaatar & Fergus (2015), Yu et al. (2018), Natarajan et al. (2013), Natarajan et al. (2017), Northcutt et al. (2017), and Northcutt et al. (2021) were based on the CCN model, and Katz-Samuels et al. (2019), Blanchard & Scott (2014), and Blanchard et al. (2016) were based on the MCD model. Menon et al. (2015), van Rooyen & Williamson (2017), and Katz-Samuels et al. (2019) studied multiple noise models at the same time. However, the current paper investigates the connections between MCD, CCN, and confidence-based settings simultaneously through the lens of matrix decontamination as broadly as possible to identify a generic methodology for WSLs. Different from the current paper aiming for risk minimization, research also studied various performance measures, such as the balanced error rate (Scott & Zhang, 2020; 2019; Menon et al., 2015; du Plessis et al., 2013), the area under the receiver operating characteristic curve (Charoenphakdee et al., 2019; Sakai et al., 2018; Menon et al., 2015), and cost-sensitive measures (Charoenphakdee et al., 2021; Natarajan et al., 2017). We choose the classification risk as the only measure due to the focus of this paper.

The remaining sections are organized as follows. Section 2 reviews ERM in supervised learning, the risk rewrite problem, and the existing results. Section 3 presents the proposed framework. We show that the proposed framework provides a unified way to formulate diverse weakly supervised scenarios in Section 4. Section 5 demonstrates how to instantiate the framework to conduct risk rewrite. We demonstrate the applicability of the proposed framework to new scenarios in Section 6. Finally, we conclude the paper and discuss outlooks in Section 7. The current organization of Sections 4 and 5 aims at connecting multiple WSLs under one framework. We note that this paper can serve multiple purposes for the study of WSLs. A summary of possible use cases of the paper is provided in Appendix A.

## 2 Preliminaries

Let $(y, x)$ be a data example where the instance $x \in \mathcal{X}$ and the label $y \in \mathcal{Y}$. For binary classification, the label space $\mathcal{Y}$ is $\{p, n\}$, and for multiclass classification with $K$ classes, $\mathcal{Y} = \{1, 2, \ldots, K\} := [K]$. The joint distribution is $\mathbb{P}(Y, X)$, the class prior is $\mathbb{P}(Y)$, the class-conditional distribution is $\mathbb{P}(X|Y)$, and the class probability function is $\mathbb{P}(Y|X)$. Given a space of hypotheses $\mathcal{G}$, we denote the loss of a hypothesis $g \in \mathcal{G}$ on predicting $y$ of $(y, x)$ as $\ell_{Y=y}(g(x))$. To accommodate concise expressions and readability for all WSLs considered in this paper simultaneously, we use alias notations when the context is unambiguous. Table 1 provides a set of common notations used in this paper.

We use $(y, x)$ instead of the convention $(x, y)$ to represent a data example because, in the current paper, we focus on discussing different types of supervision. Placing the label before the instance emphasizes the type of supervision under investigation in theorems and derivations.

### 2.1 Supervised Learning and the ERM Method

In supervised learning with $K$ classes, the observed data is of the form

$$\{x_i^y\}_{i=1}^{n_y} \overset{\text{i.i.d.}}{\sim} \mathcal{P}_{X|Y=y}, \forall y \in [K].$$

Table 1: Alias of Common Notations.

| Name of the notation | Expression | Aliases |
|---|---|---|
| Binary classes | $\{p, n\}$ | |
| Multiple classes | $\{1, \ldots, K\}$ | $[K]$ |
| Compound set of $[K]$ | $2^{[K]} \setminus \{\emptyset, [K]\}$ | $\mathcal{S}$ |
| Joint distribution | $\mathbb{P}(Y = y, X = x)$ | $\mathcal{P}_{Y=y,x}$, $\mathcal{P}_{Y=y,X}$, or $\mathcal{P}_{Y,X}$ |
| Hypothesis and its space | $g \in \mathcal{G}$ | |
| Loss of $g$ | $\ell_{Y=y}(g(x))$ | $\ell_y$, $\ell_y(X)$, or $\ell_Y(g(X))$ |
| Classification risk | $\mathbb{E}_{Y,X}[\ell_Y(g(X))]$ | $R(g)$ |
| The $j$-th entry of vector $V$ | $(V)_j$ | $V_j$ |
| Class prior | $\mathbb{P}(Y = y)$ | $\pi_y$ |
| Marginal | $\mathbb{P}(X)$ | $\mathcal{P}_X$ |
| Class-conditional | $\mathbb{P}(X = x\|Y = y)$ | $\mathcal{P}_{X|Y}$, $\mathcal{P}_{X|Y=y}$, or $\mathcal{P}_{x|Y=y}$ |
| Confidence | $\mathbb{P}(Y = y\|X = x)$ | $r_y(X)$, $r_y(x)$, or $r(X)$ if $y = p$ |

Notation $x_i^y$ denotes the shorthand of $(y, x_i)$. The goal of learning is to find a classifier $g \in \mathcal{G}$ that minimizes the classification risk

$$R(g) := \mathbb{E}_{Y,X}[\ell_Y(g(X))] = \sum_{y=1}^{K} \int_{x \in \mathcal{X}} \mathcal{P}_{Y=y,x} \, \ell_{Y=y}(g(x)) \, \mathrm{d}x. \tag{1}$$

To find such a classifier, ERM first constructs an empirical risk estimator with the data in hand:

$$\hat{R}(g) = \sum_{y=1}^{K} \frac{1}{n_y} \sum_{i=1}^{n_y} \pi_y \ell_{Y=y}(g(x_i^y)). \tag{2}$$

The estimator approximates $R(g)$ consistently since it can be shown that (2) approaches (1) as $N \to \infty$ (Tewari & Bartlett, 2014; Kiryo et al., 2017) and (Sugiyama et al., 2022, Chapter 3). Then, ERM takes $\hat{R}(g)$ as the training objective and optimizes it to find the optimal classifier

$$g^* = \arg\min_{g \in \mathcal{G}} \hat{R}(g) \tag{3}$$

in the hypothesis space $\mathcal{G}$ as the output of ERM.

## 2.2 The Risk Rewrite Problem and Existing Results

Sections 2.2.1 to 2.2.17 review the learning scenarios including WSLs, MCD, and CCN that will be discussed in this paper. A knowledgeable reader may refer directly to summary Tables 2 through 6 and proceed to Section 3.

In every WSL scenario, the goal of learning is the same as supervised learning. However, the observed data is no longer as perfectly labeled as in supervised learning. That said, there are differences in the formulations of the observed data and the ways of estimating the classification risk. We begin with reviewing WSLs derived from binary classes. For $K = 2$, we denote $\mathcal{Y} := \{p, n\}$.

### 2.2.1 Positive-Unlabeled (PU) learning

The observed data in PU learning (du Plessis et al., 2015) is of the form

$$\begin{aligned}
\{x_i^p\}_{i=1}^{n_p} &\overset{\text{i.i.d.}}{\sim} \mathcal{P}_P := \mathcal{P}_{X|Y=p}, \\
\{x_j^u\}_{j=1}^{n_u} &\overset{\text{i.i.d.}}{\sim} \mathcal{P}_U := \pi_p \, \mathcal{P}_{X|Y=p} + \pi_n \, \mathcal{P}_{X|Y=n},
\end{aligned} \tag{4}$$

where $x_j^{\mathrm{u}}$ is viewed as the shorthand of $(\mathrm{u}, x_j)$ symbolizing the unlabeled data[1]. The unlabeled data set $\{x_j^{\mathrm{u}}\}_j$ consists of a mixture of samples from $\mathcal{P}_{X|Y=\mathrm{p}}$ and $\mathcal{P}_{X|Y=\mathrm{n}}$ with proportion $\pi_{\mathrm{p}}$. Since the information of negatively sampled data is unavailable, (2) is uncomputable, causing directly optimizing (3) infeasibility. Therefore, to make ERM applicable, the *risk rewrite problem* (Sugiyama et al., 2022) asks:

Can one rephrase the classification risk $R(g)$ (1) in terms of the given data formulation?

du Plessis et al. (2015) rewrote the classification risk in terms of the data-generating distributions $\mathcal{P}_{\mathrm{P}}$ and $\mathcal{P}_{\mathrm{U}}$ as

$$R(g) = \mathbb{E}_{\mathrm{P}}\left[\pi_{\mathrm{p}}\ell_{\mathrm{p}} - \pi_{\mathrm{p}}\ell_{\mathrm{n}}\right] + \mathbb{E}_{\mathrm{U}}\left[\ell_{\mathrm{n}}\right]. \tag{5}$$

### 2.2.2 Positive-confidence (Pconf) Learning Learning

The observed data in Pconf learning (Ishida et al., 2018) is of the form

$$\{x_i, r(x_i)\}_{i=1}^{n},$$

where

$$\begin{aligned} x_i &\overset{\text{i.i.d.}}{\sim} \mathcal{P}_{\mathrm{P}} := \mathcal{P}_{X|Y=\mathrm{p}}, \\ r(x_i) &:= \mathcal{P}_{Y=\mathrm{p}|X=x_i}. \end{aligned} \tag{6}$$

The function $r(x)$ represents how confident an example $x$ would be positively labeled. Ishida et al. (2018) rewrote the classification risk as

$$R(g) = \pi_{\mathrm{p}}\mathbb{E}_{\mathrm{P}}\left[\ell_{\mathrm{p}} + \frac{1 - r(X)}{r(X)}\ell_{\mathrm{n}}\right]. \tag{7}$$

### 2.2.3 Unlabeled-Unlabeled (UU) learning

The observed data in UU learning (Lu et al., 2019) is of the form

$$\begin{aligned} \{x_i^{\mathrm{u}_1}\}_{i=1}^{n_{\mathrm{u}_1}} &\overset{\text{i.i.d.}}{\sim} \mathcal{P}_{\mathrm{U}_1} := (1 - \gamma_1)\,\mathcal{P}_{X|Y=\mathrm{p}} + \gamma_1\,\mathcal{P}_{X|Y=\mathrm{n}}, \\ \{x_j^{\mathrm{u}_2}\}_{j=1}^{n_{\mathrm{u}_2}} &\overset{\text{i.i.d.}}{\sim} \mathcal{P}_{\mathrm{U}_2} := \gamma_2\,\mathcal{P}_{X|Y=\mathrm{p}} + (1 - \gamma_2)\,\mathcal{P}_{X|Y=\mathrm{n}}, \end{aligned} \tag{8}$$

where $x_i^{\mathrm{u}_1}$ (resp. $x_j^{\mathrm{u}_2}$) being the shorthand of $(\mathrm{u}_1, x_i)$ (resp. $(\mathrm{u}_2, x_j)$) represents $x_i$ (resp. $x_j$) belonging to the unlabeled data whose mixture parameter is $\gamma_1$ (resp. $\gamma_2$). Notice a difference that the mixture proportion of the unlabeled data in PU learning is $\pi_{\mathrm{p}}$. Lu et al. (2019) rewrote the classification risk in terms of the data-generating distributions $\mathcal{P}_{\mathrm{U}_1}$ and $\mathcal{P}_{\mathrm{U}_2}$ as follows: Assume $\gamma_1 + \gamma_2 \neq 1$. Then,

$$R(g) = \mathbb{E}_{\mathrm{U}_1}\left[\frac{(1 - \gamma_2)\pi_{\mathrm{p}}}{1 - \gamma_1 - \gamma_2}\ell_{\mathrm{p}} + \frac{-\gamma_2\pi_{\mathrm{n}}}{1 - \gamma_1 - \gamma_2}\ell_{\mathrm{n}}\right] + \mathbb{E}_{\mathrm{U}_2}\left[\frac{-\gamma_1\pi_{\mathrm{p}}}{1 - \gamma_1 - \gamma_2}\ell_{\mathrm{p}} + \frac{(1 - \gamma_1)\pi_{\mathrm{n}}}{1 - \gamma_1 - \gamma_2}\ell_{\mathrm{n}}\right]. \tag{9}$$

### 2.2.4 Similar-Unlabeled (SU) learning

The observed data in SU learning (Bao et al., 2018) is of the form

$$\begin{aligned} \left\{\left(x_i^{\mathrm{s}}, x_i^{\mathrm{s}'}\right)\right\}_{i=1}^{n_{\mathrm{s}}} &\overset{\text{i.i.d.}}{\sim} \mathcal{P}_{\mathrm{S}} := \frac{\pi_{\mathrm{p}}^2 \mathcal{P}_{X|Y=\mathrm{p}}\mathcal{P}_{X'|Y=\mathrm{p}} + \pi_{\mathrm{n}}^2 \mathcal{P}_{X|Y=\mathrm{n}}\mathcal{P}_{X'|Y=\mathrm{n}}}{\pi_{\mathrm{p}}^2 + \pi_{\mathrm{n}}^2}, \\ \{x_j^{\mathrm{u}}\}_{j=1}^{n_{\mathrm{u}}} &\overset{\text{i.i.d.}}{\sim} \mathcal{P}_{\mathrm{U}} := \pi_{\mathrm{p}}\,\mathcal{P}_{X|Y=\mathrm{p}} + \pi_{\mathrm{n}}\,\mathcal{P}_{X|Y=\mathrm{n}}. \end{aligned} \tag{10}$$

The word "similar" means the examples in every $(x^{\mathrm{s}}, x^{\mathrm{s}'})$ pair have the same label; either both are positive, or both are negative. Under the assumption $\pi_{\mathrm{p}} \neq \pi_{\mathrm{n}}$, Bao et al. (2018) rewrote the classification risk as

$$R(g) = \left(\pi_{\mathrm{p}}^2 + \pi_{\mathrm{n}}^2\right)\mathbb{E}_{\mathrm{S}}\left[\frac{\mathcal{L}(X) + \mathcal{L}(X')}{2}\right] + \mathbb{E}_{\mathrm{U}}\left[\mathcal{L}_-(X)\right], \tag{11}$$

---

[1]Seemingly being redundant, but it is helpful to use $(\mathrm{u}, x_j)$ to distinguish it from the positively labeled instance $(\mathrm{p}, x_i)$.

where

$$\begin{aligned}
\mathcal{L}(X) &:= \frac{1}{\pi_{\mathrm{p}} - \pi_{\mathrm{n}}}\ell_{\mathrm{p}}(X) - \frac{1}{\pi_{\mathrm{p}} - \pi_{\mathrm{n}}}\ell_{\mathrm{n}}(X), \\
\mathcal{L}_{-}(X) &:= -\frac{\pi_{\mathrm{n}}}{\pi_{\mathrm{p}} - \pi_{\mathrm{n}}}\ell_{\mathrm{p}}(X) + \frac{\pi_{\mathrm{p}}}{\pi_{\mathrm{p}} - \pi_{\mathrm{n}}}\ell_{\mathrm{n}}(X).
\end{aligned}$$

### 2.2.5 Dissimilar-Unlabeled (DU) learning

The observed data in DU learning (Shimada et al., 2021) is of the form

$$\begin{aligned}
\left\{\left(x_i^{\mathrm{d}}, x_i^{\mathrm{d}'}\right)\right\}_{i=1}^{n_{\mathrm{d}}} &\overset{\text{i.i.d.}}{\sim} \mathcal{P}_{\mathrm{D}} := \frac{\mathcal{P}_{X|Y=\mathrm{p}}\mathcal{P}_{X'|Y=\mathrm{n}} + \mathcal{P}_{X|Y=\mathrm{n}}\mathcal{P}_{X'|Y=\mathrm{p}}}{2}, \\
\left\{x_j^{\mathrm{u}}\right\}_{j=1}^{n_{\mathrm{u}}} &\overset{\text{i.i.d.}}{\sim} \mathcal{P}_{\mathrm{U}} := \pi_{\mathrm{p}}\,\mathcal{P}_{X|Y=\mathrm{p}} + \pi_{\mathrm{n}}\,\mathcal{P}_{X|Y=\mathrm{n}}.
\end{aligned} \tag{12}$$

The word "dissimilar" means the examples in every $(x^{\mathrm{d}}, x^{\mathrm{d}'})$ pair have distinct labels. Under the assumption $\pi_{\mathrm{p}} \neq \pi_{\mathrm{n}}$, Shimada et al. (2021) rewrote the classification risk as

$$R(g) = 2\pi_{\mathrm{p}}\pi_{\mathrm{n}}\mathbb{E}_{\mathrm{D}}\left[-\frac{\mathcal{L}(X) + \mathcal{L}(X')}{2}\right] + \mathbb{E}_{\mathrm{U}}\left[\mathcal{L}_{+}(X)\right], \tag{13}$$

where

$$\begin{aligned}
\mathcal{L}(X) &= \frac{1}{\pi_{\mathrm{p}} - \pi_{\mathrm{n}}}\ell_{\mathrm{p}}(X) - \frac{1}{\pi_{\mathrm{p}} - \pi_{\mathrm{n}}}\ell_{\mathrm{n}}(X), \\
\mathcal{L}_{+}(X) &:= \frac{\pi_{\mathrm{p}}}{\pi_{\mathrm{p}} - \pi_{\mathrm{n}}}\ell_{\mathrm{p}}(X) - \frac{\pi_{\mathrm{n}}}{\pi_{\mathrm{p}} - \pi_{\mathrm{n}}}\ell_{\mathrm{n}}(X).
\end{aligned}$$

Note that $\mathcal{L}(X)$ has been defined in the SU setting. We repeat it here for clarity.

### 2.2.6 Similar-Dissimilar (SD) learning

The observed data in SD learning (Shimada et al., 2021) is of the form

$$\begin{aligned}
\left\{\left(x_i^{\mathrm{s}}, x_i^{\mathrm{s}'}\right)\right\}_{i=1}^{n_{\mathrm{s}}} &\overset{\text{i.i.d.}}{\sim} \mathcal{P}_{\mathrm{S}} := \frac{\pi_{\mathrm{p}}^2\mathcal{P}_{X|Y=\mathrm{p}}\mathcal{P}_{X'|Y=\mathrm{p}} + \pi_{\mathrm{n}}^2\mathcal{P}_{X|Y=\mathrm{n}}\mathcal{P}_{X'|Y=\mathrm{n}}}{\pi_{\mathrm{p}}^2 + \pi_{\mathrm{n}}^2}, \\
\left\{\left(x_i^{\mathrm{d}}, x_i^{\mathrm{d}'}\right)\right\}_{i=1}^{n_{\mathrm{d}}} &\overset{\text{i.i.d.}}{\sim} \mathcal{P}_{\mathrm{D}} := \frac{\mathcal{P}_{X|Y=\mathrm{p}}\mathcal{P}_{X'|Y=\mathrm{n}} + \mathcal{P}_{X|Y=\mathrm{n}}\mathcal{P}_{X'|Y=\mathrm{p}}}{2}.
\end{aligned} \tag{14}$$

Under the assumption $\pi_{\mathrm{p}} \neq \pi_{\mathrm{n}}$, Shimada et al. (2021) rewrote the classification risk as

$$R(g) = \left(\pi_{\mathrm{p}}^2 + \pi_{\mathrm{n}}^2\right)\mathbb{E}_{\mathrm{S}}\left[\frac{\mathcal{L}_{+}(X) + \mathcal{L}_{+}(X')}{2}\right] + 2\pi_{\mathrm{p}}\pi_{\mathrm{n}}\mathbb{E}_{\mathrm{D}}\left[\frac{\mathcal{L}_{-}(X) + \mathcal{L}_{-}(X')}{2}\right], \tag{15}$$

where

$$\begin{aligned}
\mathcal{L}_{+}(X) &= \frac{\pi_{\mathrm{p}}}{\pi_{\mathrm{p}} - \pi_{\mathrm{n}}}\ell_{\mathrm{p}}(X) - \frac{\pi_{\mathrm{n}}}{\pi_{\mathrm{p}} - \pi_{\mathrm{n}}}\ell_{\mathrm{n}}(X), \\
\mathcal{L}_{-}(X) &= -\frac{\pi_{\mathrm{n}}}{\pi_{\mathrm{p}} - \pi_{\mathrm{n}}}\ell_{\mathrm{p}}(X) + \frac{\pi_{\mathrm{p}}}{\pi_{\mathrm{p}} - \pi_{\mathrm{n}}}\ell_{\mathrm{n}}(X).
\end{aligned}$$

Note that $\mathcal{L}_{+}(X)$ and $\mathcal{L}_{-}(X)$ have been defined in the DU and SU settings. We repeat them here for clarity.

### 2.2.7 Pairwise Comparison (Pcomp) Learning

The observed data in Pcomp learning (Feng et al., 2021) is of the form

$$\left\{\left(x_i^{\mathrm{pc}}, x_i^{\mathrm{pc}'}\right)\right\}_{i=1}^{n_{\mathrm{pc}}} \overset{\text{i.i.d.}}{\sim} \mathcal{P}_{\mathrm{PC}} := \frac{\pi_{\mathrm{p}}^2\mathcal{P}_{X|Y=\mathrm{p}}\mathcal{P}_{X'|Y=\mathrm{p}} + \pi_{\mathrm{p}}\pi_{\mathrm{n}}\mathcal{P}_{X|Y=\mathrm{p}}\mathcal{P}_{X'|Y=\mathrm{n}} + \pi_{\mathrm{n}}^2\mathcal{P}_{X|Y=\mathrm{n}}\mathcal{P}_{X'|Y=\mathrm{n}}}{\pi_{\mathrm{p}}^2 + \pi_{\mathrm{p}}\pi_{\mathrm{n}} + \pi_{\mathrm{n}}^2}.$$

$$\tag{16}$$

The pairwise comparison encodes a meaning that each $x^{\mathrm{pc}}$ "can not be more negative" than $x^{\mathrm{pc}'}$ in the $(x^{\mathrm{pc}}, x^{\mathrm{pc}'})$ pair. That is, the labels in $(x^{\mathrm{pc}}, x^{\mathrm{pc}'})$ are of the form $(\mathrm{p}, \mathrm{p})$, $(\mathrm{p}, \mathrm{n})$, or $(\mathrm{n}, \mathrm{n})$. Feng et al. (2021) rewrote the classification risk as

$$R(g) = \mathbb{E}_{\mathrm{Sup}}\left[\ell_{\mathrm{p}} - \pi_{\mathrm{p}}\ell_{\mathrm{n}}\right] + \mathbb{E}_{\mathrm{Inf}}\left[-\pi_{\mathrm{n}}\ell_{\mathrm{p}} + \ell_{\mathrm{n}}\right], \tag{17}$$

where the expectations are computed over the following distributions

$$\begin{aligned} \mathcal{P}_{\mathrm{Sup}} &:= \int_{x' \in \mathcal{X}} \mathcal{P}_{\mathrm{PC}}\,\mathrm{d}x', \\ \mathcal{P}_{\mathrm{Inf}} &:= \int_{x \in \mathcal{X}} \mathcal{P}_{\mathrm{PC}}\,\mathrm{d}x. \end{aligned}$$

### 2.2.8 Similarity-Confidence Learning (Sconf) Learning

The observed data in Sconf learning (Cao et al., 2021b) is of the form

$$\left\{ x_i^{\mathrm{sc}}, x_i^{\mathrm{sc}'}, r\left(x_i^{\mathrm{sc}}, x_i^{\mathrm{sc}'}\right) \right\}_{i=1}^n,$$

where

$$\begin{aligned} x_i^{\mathrm{sc}} &\stackrel{\mathrm{i.i.d.}}{\sim} \mathcal{P}_X := \pi_{\mathrm{p}}\,\mathcal{P}_{X|Y=\mathrm{p}} + \pi_{\mathrm{n}}\,\mathcal{P}_{X|Y=\mathrm{n}}, \\ x_i^{\mathrm{sc}'} &\stackrel{\mathrm{i.i.d.}}{\sim} \mathcal{P}_{X'} := \pi_{\mathrm{p}}\,\mathcal{P}_{X'|Y=\mathrm{p}} + \pi_{\mathrm{n}}\,\mathcal{P}_{X'|Y=\mathrm{n}}, \\ r\left(x_i^{\mathrm{sc}}, x_i^{\mathrm{sc}'}\right) &:= \mathcal{P}_{Y=y_i^{\mathrm{sc}}=Y'=y_i^{\mathrm{sc}'}|X=x_i^{\mathrm{sc}},X'=x_i^{\mathrm{sc}'}}. \end{aligned} \tag{18}$$

Cao et al. (2021b) rewrote the classification risk as

$$R(g) = \mathbb{E}_{X,X'}\left[\frac{r(X,X') - \pi_{\mathrm{n}}}{\pi_{\mathrm{p}} - \pi_{\mathrm{n}}}\mathcal{L}_{\mathrm{p}}(X,X') + \frac{\pi_{\mathrm{p}} - r(X,X')}{\pi_{\mathrm{p}} - \pi_{\mathrm{n}}}\mathcal{L}_{\mathrm{n}}(X,X')\right], \tag{19}$$

where

$$\begin{aligned} \mathcal{L}_{\mathrm{p}}(X,X') &:= \frac{\ell_{\mathrm{p}}(X) + \ell_{\mathrm{p}}(X')}{2}, \\ \mathcal{L}_{\mathrm{n}}(X,X') &:= \frac{\ell_{\mathrm{n}}(X) + \ell_{\mathrm{n}}(X')}{2}. \end{aligned}$$

### 2.2.9 Complementary-Label (CL) Learning

One can also formulate weak supervision from multiclass classification. For $K$ classes, we denote $\mathcal{Y} := [K]$.

The observed data in CL learning (Ishida et al., 2019) is of the form

$$\{(\bar{s}_i, x_i)\}_{i=1}^n \stackrel{\mathrm{i.i.d.}}{\sim} \mathcal{P}_{\bar{S},X} := \frac{1}{K-1} \sum_{Y \neq \bar{S}} \mathcal{P}_{Y,X}. \tag{20}$$

As is named "complementary," $\bar{s} \in [K]$ represents that the true label $y$ of $x$ cannot be $\bar{s}$. Ishida et al. (2019) rewrote the classification risk as

$$R(g) = \mathbb{E}_{\bar{S},X}\left[\sum_{y=1}^K \ell_y - (K-1)\ell_{\bar{S}}\right]. \tag{21}$$

### 2.2.10 Multi-Complementary-Label (MCL) Learning

The observed data in MCL learning (Feng et al., 2020a) is of the form

$$\{(\bar{s}_i, x_i)\}_{i=1}^n \overset{\text{i.i.d.}}{\sim} \mathcal{P}_{\bar{S},X} := \begin{cases} \sum_{d=1}^{K-1} \mathcal{P}_{|\bar{S}|=d} \cdot \frac{1}{\binom{K-1}{|\bar{S}|}} \sum_{Y \notin \bar{S}} \mathcal{P}_{Y,X}, & \text{if } |\bar{S}| = d, \\ 0, & \text{otherwise.} \end{cases} \tag{22}$$

Generalized from CL, $\bar{s} \subset [K]$ in MCL is a set of classes of size $d \in [K-1]$, representing multiple exclusions. In other words, CL is the special case of MCL with $d = 1$. Feng et al. (2020a) rewrote the classification risk as

$$R(g) = \sum_{d=1}^{K-1} \mathcal{P}_{|\bar{S}|=d} \mathbb{E}_{\bar{S},X||\bar{S}|=d} \left[ \sum_{y \notin \bar{S}} \ell_y - \frac{K-1-|\bar{S}|}{|\bar{S}|} \sum_{\bar{s} \in \bar{S}} \ell_{\bar{s}} \right]. \tag{23}$$

### 2.2.11 Provably Consistent Partial-Label (PCPL) Learning

The observed data in PCPL learning (Feng et al., 2020b) is of the form

$$\{(s_i, x_i)\}_{i=1}^n \overset{\text{i.i.d.}}{\sim} \mathcal{P}_{S,X} := \frac{1}{2^{K-1}-1} \sum_{Y \in S} \mathcal{P}_{Y,X}. \tag{24}$$

A partial-label $s \subset [K]$ is a set of classes containing the true label $y$ of $x$. Feng et al. (2020b) rewrote the classification risk as

$$R(g) = \frac{1}{2} \mathbb{E}_{S,X} \left[ \sum_{y=1}^K \frac{\mathcal{P}_{Y=y|X}}{\sum_{a \in S} \mathcal{P}_{Y=a|X}} \ell_y \right]. \tag{25}$$

### 2.2.12 Proper Partial-Label (PPL) Learning

The observed data in PPL learning (Wu et al., 2023) is of the form

$$\{(s_i, x_i)\}_{i=1}^n \overset{\text{i.i.d.}}{\sim} \mathcal{P}_{S,X} := C(S, X) \sum_{Y \in S} \mathcal{P}_{Y,X}. \tag{26}$$

The weight $\frac{1}{2^{K-1}-1}$ in PCPL is generalized to $C(S, X)$, a function of the partial-label and the instance, allowing one to characterize the "properness" of a partial-label. Wu et al. (2023) rewrote the classification risk as

$$R(g) = \mathbb{E}_{S,X} \left[ \sum_{y \in S} \frac{\mathcal{P}_{Y=y|X}}{\sum_{a \in S} \mathcal{P}_{Y=a|X}} \ell_y \right]. \tag{27}$$

### 2.2.13 Single-Class Confidence (SC-Conf) Learning

The observed data in SC-Conf learning (Cao et al., 2021a) is of the form

$$\{x_i, r_1(x_i), \ldots, r_K(x_i)\}_{i=1}^n,$$

where

$$\begin{aligned} x_i &\overset{\text{i.i.d.}}{\sim} \mathcal{P}_{X|Y=y_s} \text{ with } y_s \in [K], \\ r_k(x_i) &:= \mathcal{P}_{Y=k|X=x_i} \text{ for each } k \in [K]. \end{aligned} \tag{28}$$

The constraint of SC-Conf is that the examples are sampled from a specific class $y_s$. The key to risk rewrite is the availability of confident information $r_k(x)$ about each class. Cao et al. (2021a) rewrote the classification risk as

$$R(g) = \pi_{y_s} \mathbb{E}_{X|Y=y_s} \left[ \sum_{y=1}^K \frac{r_y(X)}{r_{y_s}(X)} \ell_y \right]. \tag{29}$$

### 2.2.14 Subset Confidence (Sub-Conf) Learning

The observed data in Sub-Conf learning (Cao et al., 2021a) is of the form

$$\{x_i, r_1(x_i), \ldots, r_K(x_i)\}_{i=1}^n,$$

where

$$
\begin{aligned}
x_i &\overset{\text{i.i.d.}}{\sim} \mathcal{P}_{X|Y \in \mathcal{Y}_s} \text{ with } \mathcal{Y}_s \subset [K], \\
r_k(x_i) &:= \mathcal{P}_{Y=k|X=x_i} \text{ for each } k \in [K].
\end{aligned}
\tag{30}
$$

Sub-Conf is a relaxed setting of SC-Conf where the samples come from a set of classes $\mathcal{Y}_s$. Cao et al. (2021a) rewrote the classification risk as

$$R(g) = \pi_{\mathcal{Y}_s} \mathbb{E}_{X|Y \in \mathcal{Y}_s} \left[ \sum_{y=1}^K \frac{r_y(X)}{r_{\mathcal{Y}_s}(X)} \ell_y \right], \tag{31}$$

where $\pi_{\mathcal{Y}_s} := \sum_{j \in \mathcal{Y}_s} \pi_j$, and $r_{\mathcal{Y}_s}(X) := \mathcal{P}_{Y \in \mathcal{Y}_s|X} = \sum_{j \in \mathcal{Y}_s} \mathcal{P}_{Y=j|X}$.

### 2.2.15 Soft-Label Learning

Ishida et al. (2023) formulated soft-label learning under the binary setting, in which the observed data is of the form

$$\{x_i, r(x_i)\}_{i=1}^n,$$

where

$$
\begin{aligned}
x_i &\overset{\text{i.i.d.}}{\sim} \mathcal{P}_X := \mathcal{P}_{Y=\text{p},X} + \mathcal{P}_{Y=\text{n},X}, \\
r(x_i) &:= \mathcal{P}_{Y=\text{p}|X=x_i}.
\end{aligned}
\tag{32}
$$

It is straightforward to obtain a corresponding formulation under the multiclass setting:

$$\{x_i, r_1(x_i), \ldots, r_K(x_i)\}_{i=1}^n,$$

where

$$
\begin{aligned}
x_i &\overset{\text{i.i.d.}}{\sim} \mathcal{P}_X := \sum_{k=1}^K \mathcal{P}_{Y=k,X}, \\
r_k(x_i) &:= \mathcal{P}_{Y=k|X=x_i} \text{ for each } k \in [K].
\end{aligned}
\tag{33}
$$

The difference between SC-Conf and multiclass soft-label (resp. the difference between Pconf and binary soft-label) is the sample distribution of $x$. We rewrote the classification risk as

$$R(g) = \mathbb{E}_X \left[ \sum_{y=1}^K r_y(X) \ell_y \right]. \tag{34}$$

### 2.2.16 Summary of Existing WSL Formulations and Risk Rewrites

We summarize the weakly supervised scenarios discussed and their risk rewrite results. The formulations are divided into the binary classification settings in Table 2 and the multiclass classification settings in Table 3. We list the formulations in chronological order, according to their publication order. Tables 4 and 5 are the corresponding rewrites.

Table 2: Binary WSL formulations.

| WSL | Formulation | Equation |
|---|---|---|
| PU | $\{x_i^{\mathrm{p}}\}_{i=1}^{n_{\mathrm{p}}} \overset{\text{i.i.d.}}{\sim} \mathcal{P}_{\mathrm{P}} := \mathcal{P}_{X|Y=\mathrm{p}},$ 
 $\{x_j^{\mathrm{u}}\}_{j=1}^{n_{\mathrm{u}}} \overset{\text{i.i.d.}}{\sim} \mathcal{P}_{\mathrm{U}} := \pi_{\mathrm{p}} \, \mathcal{P}_{X|Y=\mathrm{p}} + \pi_{\mathrm{n}} \, \mathcal{P}_{X|Y=\mathrm{n}}.$ | (4) |
| Pconf | $\{x_i, r(x_i)\}_{i=1}^{n}, \text{ where}$ 
 $\quad x_i \overset{\text{i.i.d.}}{\sim} \mathcal{P}_{\mathrm{P}} := \mathcal{P}_{X|Y=\mathrm{p}},$ 
 $\quad r(x_i) := \mathcal{P}_{Y=\mathrm{p}|X=x_i}.$ | (6) |
| UU | $\{x_i^{\mathrm{u}_1}\}_{i=1}^{n_{\mathrm{u}_1}} \overset{\text{i.i.d.}}{\sim} \mathcal{P}_{\mathrm{U}_1} := (1-\gamma_1) \, \mathcal{P}_{X|Y=\mathrm{p}} + \gamma_1 \, \mathcal{P}_{X|Y=\mathrm{n}},$ 
 $\{x_j^{\mathrm{u}_2}\}_{j=1}^{n_{\mathrm{u}_2}} \overset{\text{i.i.d.}}{\sim} \mathcal{P}_{\mathrm{U}_2} := \gamma_2 \, \mathcal{P}_{X|Y=\mathrm{p}} + (1-\gamma_2) \, \mathcal{P}_{X|Y=\mathrm{n}}.$ | (8) |
| SU | $\left\{\left(x_i^{\mathrm{s}}, x_i^{\mathrm{s}'}\right)\right\}_{i=1}^{n_{\mathrm{s}}} \overset{\text{i.i.d.}}{\sim} \mathcal{P}_{\mathrm{S}} := \dfrac{\pi_{\mathrm{p}}^2 \mathcal{P}_{X|Y=\mathrm{p}} \mathcal{P}_{X'|Y=\mathrm{p}} + \pi_{\mathrm{n}}^2 \mathcal{P}_{X|Y=\mathrm{n}} \mathcal{P}_{X'|Y=\mathrm{n}}}{\pi_{\mathrm{p}}^2 + \pi_{\mathrm{n}}^2},$ 
 $\quad \{x_j^{\mathrm{u}}\}_{j=1}^{n_{\mathrm{u}}} \overset{\text{i.i.d.}}{\sim} \mathcal{P}_{\mathrm{U}} := \pi_{\mathrm{p}} \, \mathcal{P}_{X|Y=\mathrm{p}} + \pi_{\mathrm{n}} \, \mathcal{P}_{X|Y=\mathrm{n}}.$ | (10) |
| DU | $\left\{\left(x_i^{\mathrm{d}}, x_i^{\mathrm{d}'}\right)\right\}_{i=1}^{n_{\mathrm{d}}} \overset{\text{i.i.d.}}{\sim} \mathcal{P}_{\mathrm{D}} := \dfrac{\mathcal{P}_{X|Y=\mathrm{p}} \mathcal{P}_{X'|Y=\mathrm{n}} + \mathcal{P}_{X|Y=\mathrm{n}} \mathcal{P}_{X'|Y=\mathrm{p}}}{2},$ 
 $\quad \{x_j^{\mathrm{u}}\}_{j=1}^{n_{\mathrm{u}}} \overset{\text{i.i.d.}}{\sim} \mathcal{P}_{\mathrm{U}} := \pi_{\mathrm{p}} \, \mathcal{P}_{X|Y=\mathrm{p}} + \pi_{\mathrm{n}} \, \mathcal{P}_{X|Y=\mathrm{n}}.$ | (12) |
| SD | $\left\{\left(x_i^{\mathrm{s}}, x_i^{\mathrm{s}'}\right)\right\}_{i=1}^{n_{\mathrm{s}}} \overset{\text{i.i.d.}}{\sim} \mathcal{P}_{\mathrm{S}} := \dfrac{\pi_{\mathrm{p}}^2 \mathcal{P}_{X|Y=\mathrm{p}} \mathcal{P}_{X'|Y=\mathrm{p}} + \pi_{\mathrm{n}}^2 \mathcal{P}_{X|Y=\mathrm{n}} \mathcal{P}_{X'|Y=\mathrm{n}}}{\pi_{\mathrm{p}}^2 + \pi_{\mathrm{n}}^2},$ 
 $\left\{\left(x_i^{\mathrm{d}}, x_i^{\mathrm{d}'}\right)\right\}_{i=1}^{n_{\mathrm{d}}} \overset{\text{i.i.d.}}{\sim} \mathcal{P}_{\mathrm{D}} := \dfrac{\mathcal{P}_{X|Y=\mathrm{p}} \mathcal{P}_{X'|Y=\mathrm{n}} + \mathcal{P}_{X|Y=\mathrm{n}} \mathcal{P}_{X'|Y=\mathrm{p}}}{2}.$ | (14) |
| Pcomp | $\left\{\left(x_i^{\mathrm{pc}}, x_i^{\mathrm{pc}'}\right)\right\}_{i=1}^{n_{\mathrm{pc}}} \overset{\text{i.i.d.}}{\sim} \mathcal{P}_{\mathrm{PC}}$ 
 $:= \dfrac{\pi_{\mathrm{p}}^2 \mathcal{P}_{X|Y=\mathrm{p}} \mathcal{P}_{X'|Y=\mathrm{p}} + \pi_{\mathrm{p}} \pi_{\mathrm{n}} \mathcal{P}_{X|Y=\mathrm{p}} \mathcal{P}_{X'|Y=\mathrm{n}} + \pi_{\mathrm{n}}^2 \mathcal{P}_{X|Y=\mathrm{n}} \mathcal{P}_{X'|Y=\mathrm{n}}}{\pi_{\mathrm{p}}^2 + \pi_{\mathrm{p}} \pi_{\mathrm{n}} + \pi_{\mathrm{n}}^2}.$ | (16) |
| Sconf | $\left\{x_i^{\mathrm{sc}}, x_i^{\mathrm{sc}'}, r\left(x_i^{\mathrm{sc}}, x_i^{\mathrm{sc}'}\right)\right\}_{i=1}^{n_{\mathrm{sc}}}, \text{ where}$ 
 $\quad x_i^{\mathrm{sc}} \overset{\text{i.i.d.}}{\sim} \mathcal{P}_X := \pi_{\mathrm{p}} \, \mathcal{P}_{X|Y=\mathrm{p}} + \pi_{\mathrm{n}} \, \mathcal{P}_{X|Y=\mathrm{n}},$ 
 $\quad x_i^{\mathrm{sc}'} \overset{\text{i.i.d.}}{\sim} \mathcal{P}_{X'} := \pi_{\mathrm{p}} \, \mathcal{P}_{X'|Y=\mathrm{p}} + \pi_{\mathrm{n}} \, \mathcal{P}_{X'|Y=\mathrm{n}},$ 
 $\quad r\left(x_i^{\mathrm{sc}}, x_i^{\mathrm{sc}'}\right) := \mathcal{P}_{Y=y_i^{\mathrm{sc}}=Y'=y_i^{\mathrm{sc}'}|X=x_i^{\mathrm{sc}}, X'=x_i^{\mathrm{sc}'}}.$ | (18) |

Table 3: Multiclass WSL formulations.

| WSL | Formulation | Equation |
|---|---|---|
| CL | $\{(\bar{s}_i, x_i)\}_{i=1}^n \overset{\text{i.i.d.}}{\sim} \mathcal{P}_{\bar{S},X} := \frac{1}{K-1} \sum_{Y \neq \bar{S}} \mathcal{P}_{Y,X}.$ | (20) |
| MCL | $\{(\bar{s}_i, x_i)\}_{i=1}^n \overset{\text{i.i.d.}}{\sim} \mathcal{P}_{\bar{S},X} := \begin{cases} \sum_{d=1}^{K-1} \mathcal{P}_{\|\bar{S}\|=d} \cdot \frac{1}{\binom{K-1}{\|\bar{S}\|}} \sum_{Y \notin \bar{S}} \mathcal{P}_{Y,X}, & \text{if } \|\bar{S}\| = d, \\ 0, & \text{otherwise.} \end{cases}$ | (22) |
| PCPL | $\{(s_i, x_i)\}_{i=1}^n \overset{\text{i.i.d.}}{\sim} \mathcal{P}_{S,X} := \frac{1}{2^{K-1}-1} \sum_{Y \in S} \mathcal{P}_{Y,X}.$ | (24) |
| PPL | $\{(s_i, x_i)\}_{i=1}^n \overset{\text{i.i.d.}}{\sim} \mathcal{P}_{S,X} := C(S,X) \sum_{Y \in S} \mathcal{P}_{Y,X}.$ | (26) |
| SC-Conf | $\{x_i, r_1(x_i), \ldots, r_K(x_i)\}_{i=1}^n,$ where $x_i \overset{\text{i.i.d.}}{\sim} \mathcal{P}_{X\|Y=y_s}$ with $y_s \in [K],$ $r_k(x_i) := \mathcal{P}_{Y=k\|X=x_i}$ for each $k \in [K].$ | (28) |
| Sub-Conf | $\{x_i, r_1(x_i), \ldots, r_K(x_i)\}_{i=1}^n,$ where $x_i \overset{\text{i.i.d.}}{\sim} \mathcal{P}_{X\|Y \in \mathcal{Y}_s}$ with $\mathcal{Y}_s \subset [K],$ $r_k(x_i) := \mathcal{P}_{Y=k\|X=x_i}$ for each $k \in [K].$ | (30) |
| Soft-label | $\{x_i, r_1(x_i), \ldots, r_K(x_i)\}_{i=1}^n,$ where $x_i \overset{\text{i.i.d.}}{\sim} \mathcal{P}_X,$ $r_k(x_i) := \mathcal{P}_{Y=k\|X=x_i}$ for each $k \in [K].$ | (33) |

Table 4: Risk rewrites for binary WSLs.

| WSL | Risk rewrite for $R(g) = \mathbb{E}_{Y,X}\left[\ell_Y(g(X))\right]$ (1) | Equation |
|---|---|---|
| PU | $R(g) = \mathbb{E}_{\mathrm{P}}\left[\pi_{\mathrm{p}}\ell_{\mathrm{p}} - \pi_{\mathrm{p}}\ell_{\mathrm{n}}\right] + \mathbb{E}_{\mathrm{U}}\left[\ell_{\mathrm{n}}\right].$ | (5) |
| Pconf | $R(g) = \pi_{\mathrm{p}}\mathbb{E}_{\mathrm{P}}\left[\ell_{\mathrm{p}} + \dfrac{1 - r(X)}{r(X)}\ell_{\mathrm{n}}\right].$ | (7) |
| UU | $R(g) = \mathbb{E}_{\mathrm{U}_1}\left[\dfrac{(1-\gamma_2)\pi_{\mathrm{p}}}{1 - \gamma_1 - \gamma_2}\ell_{\mathrm{p}} + \dfrac{-\gamma_2\pi_{\mathrm{n}}}{1 - \gamma_1 - \gamma_2}\ell_{\mathrm{n}}\right] + \mathbb{E}_{\mathrm{U}_2}\left[\dfrac{-\gamma_1\pi_{\mathrm{p}}}{1 - \gamma_1 - \gamma_2}\ell_{\mathrm{p}} + \dfrac{(1-\gamma_1)\pi_{\mathrm{n}}}{1 - \gamma_1 - \gamma_2}\ell_{\mathrm{n}}\right].$ | (9) |
| SU | $R(g) = \left(\pi_{\mathrm{p}}^2 + \pi_{\mathrm{n}}^2\right)\mathbb{E}_{\mathrm{S}}\left[\dfrac{\mathcal{L}(X) + \mathcal{L}(X')}{2}\right] + \mathbb{E}_{\mathrm{U}}\left[\mathcal{L}_{-}(X)\right],$ where $$\mathcal{L}(X) := \frac{1}{\pi_{\mathrm{p}} - \pi_{\mathrm{n}}}\ell_{\mathrm{p}}(X) - \frac{1}{\pi_{\mathrm{p}} - \pi_{\mathrm{n}}}\ell_{\mathrm{n}}(X),$$ $$\mathcal{L}_{-}(X) := -\frac{\pi_{\mathrm{n}}}{\pi_{\mathrm{p}} - \pi_{\mathrm{n}}}\ell_{\mathrm{p}}(X) + \frac{\pi_{\mathrm{p}}}{\pi_{\mathrm{p}} - \pi_{\mathrm{n}}}\ell_{\mathrm{n}}(X).$$ | (11) |
| DU | $R(g) = 2\pi_{\mathrm{p}}\pi_{\mathrm{n}}\mathbb{E}_{\mathrm{D}}\left[-\dfrac{\mathcal{L}(X) + \mathcal{L}(X')}{2}\right] + \mathbb{E}_{\mathrm{U}}\left[\mathcal{L}_{+}(X)\right],$ where $\mathcal{L}(X)$ is defined in the SU setting, and $$\mathcal{L}_{+}(X) := \frac{\pi_{\mathrm{p}}}{\pi_{\mathrm{p}} - \pi_{\mathrm{n}}}\ell_{\mathrm{p}}(X) - \frac{\pi_{\mathrm{n}}}{\pi_{\mathrm{p}} - \pi_{\mathrm{n}}}\ell_{\mathrm{n}}(X).$$ | (13) |
| SD | $R(g) = \left(\pi_{\mathrm{p}}^2 + \pi_{\mathrm{n}}^2\right)\mathbb{E}_{\mathrm{S}}\left[\dfrac{\mathcal{L}_{+}(X) + \mathcal{L}_{+}(X')}{2}\right] + 2\pi_{\mathrm{p}}\pi_{\mathrm{n}}\mathbb{E}_{\mathrm{D}}\left[\dfrac{\mathcal{L}_{-}(X) + \mathcal{L}_{-}(X')}{2}\right],$ where $\mathcal{L}_{+}(X)$ and $\mathcal{L}_{-}(X')$ are defined in the SU and DU settings. | (15) |
| Pcomp | $R(g) = \mathbb{E}_{\mathrm{Sup}}\left[\ell_{\mathrm{p}} - \pi_{\mathrm{p}}\ell_{\mathrm{n}}\right] + \mathbb{E}_{\mathrm{Inf}}\left[-\pi_{\mathrm{n}}\ell_{\mathrm{p}} + \ell_{\mathrm{n}}\right],$ where $$\mathcal{P}_{\mathrm{Sup}} := \int_{x' \in \mathcal{X}} \mathcal{P}_{\mathrm{PC}}\,\mathrm{d}x',$$ $$\mathcal{P}_{\mathrm{Inf}} := \int_{x \in \mathcal{X}} \mathcal{P}_{\mathrm{PC}}\,\mathrm{d}x.$$ | (17) |
| Sconf | $R(g) = \mathbb{E}_{X,X'}\left[\dfrac{r(X,X') - \pi_{\mathrm{n}}}{\pi_{\mathrm{p}} - \pi_{\mathrm{n}}}\dfrac{\ell_{\mathrm{p}}(X) + \ell_{\mathrm{p}}(X')}{2} + \dfrac{\pi_{\mathrm{p}} - r(X,X')}{\pi_{\mathrm{p}} - \pi_{\mathrm{n}}}\dfrac{\ell_{\mathrm{n}}(X) + \ell_{\mathrm{n}}(X')}{2}\right].$ | (19) |

Table 5: Risk rewrites for multiclass WSLs.

| WSL | Risk rewrite for $R(g) = \mathbb{E}_{Y,X}\left[\ell_Y(g(X))\right]$ (1) | Equation |
|---|---|---|
| CL | $R(g) = \mathbb{E}_{\bar{S},X}\left[\sum_{y=1}^{K} \ell_y - (K-1)\ell_{\bar{S}}\right].$ | (21) |
| MCL | $R(g) = \sum_{d=1}^{K-1} \mathcal{P}_{|\bar{S}|=d}\, \mathbb{E}_{\bar{S},X||\bar{S}|=d}\left[\sum_{y\notin\bar{S}} \ell_y - \frac{K-1-|\bar{S}|}{|\bar{S}|}\sum_{\bar{s}\in\bar{S}} \ell_{\bar{s}}\right].$ | (23) |
| PCPL | $R(g) = \frac{1}{2}\mathbb{E}_{S,X}\left[\sum_{y=1}^{K} \frac{\mathcal{P}_{Y=y|X}}{\sum_{a\in S}\mathcal{P}_{Y=a|X}}\ell_y\right].$ | (25) |
| PPL | $R(g) = \mathbb{E}_{S,X}\left[\sum_{y\in S} \frac{\mathcal{P}_{Y=y|X}}{\sum_{a\in S}\mathcal{P}_{Y=a|X}}\ell_y\right].$ | (27) |
| SC-Conf | $R(g) = \pi_{y_\mathrm{s}}\mathbb{E}_{X|Y=y_\mathrm{s}}\left[\sum_{y=1}^{K} \frac{r_y(X)}{r_{y_\mathrm{s}}(X)}\ell_y\right].$ | (29) |
| Sub-Conf | $R(g) = \pi_{\mathcal{Y}_\mathrm{s}}\mathbb{E}_{X|Y\in\mathcal{Y}_\mathrm{s}}\left[\sum_{y=1}^{K} \frac{r_y(X)}{r_{\mathcal{Y}_\mathrm{s}}(X)}\ell_y\right].$ | (31) |
| Soft-label | $R(g) = \mathbb{E}_X\left[\sum_{y=1}^{K} r_y(X)\ell_y\right].$ | (34) |

From the above tables, finding a way to reexpress the classification risk $R(g)$ (1) in terms of the data-generating distributions becomes the crux when applying ERM for most WSL studies. The rewrites also replace loss functions $\ell_Y$ defining (1) with various modified losses (shown inside the expectations). These modified loss functions are sometimes called corrected losses, which is why the approach is also called loss correction. Proposing a generic methodology that finds properly corrected losses to achieve risk rewrite in different scenarios is a main topic we would like to elaborate on in this paper.

### 2.2.17 Learning with Noisy Labels (LNL) Formulations

Next, we review two related formulations in LNL, the MCD and CCN settings, in Table 6. The observed instances in MCD and CCN are still labeled by $\{\mathrm{p},\mathrm{n}\}$ but are polluted by certain noise models. We use $\bar{Y}$ to represent a polluted label, compared to an unpolluted $Y$. In MCD, a small portion of the negatively labeled data $\gamma_\mathrm{p}\mathcal{P}_{X|Y=\mathrm{n}}$ contaminates the positively labeled data $\mathcal{P}_{X|Y=\mathrm{p}}$. Likewise, a small portion of the positive data $\gamma_\mathrm{n}\mathcal{P}_{X|Y=\mathrm{p}}$ contaminates the negatively labeled data $\mathcal{P}_{X|Y=\mathrm{n}}$ (Scott et al., 2013). In the CCN setting, a label $Y$ is flipped to become $\bar{Y}$ with probability $\mathcal{P}_{\bar{Y}|Y,X}$ (Natarajan et al., 2013). Although they are formulated for the study of noisy labels, their formulations share similar structures with many WSLs above. In Section 4, we will use the similarities to categorize WSLs and provide a bird's eye view to reveal connections among WSLs.

Table 6: MCD and CCN formulations.

| Scenario | Formulation |
|---|---|
| MCD | $\left\{x_i^{\bar{p}}\right\}_{i=1}^{n_{\bar{p}}} \overset{\text{i.i.d.}}{\sim} \mathcal{P}_{X|\bar{Y}=\mathrm{p}} := (1 - \gamma_\mathrm{p})\,\mathcal{P}_{X|Y=\mathrm{p}} + \gamma_\mathrm{p}\,\mathcal{P}_{X|Y=\mathrm{n}}.$ 
 $\left\{x_j^{\bar{n}}\right\}_{j=1}^{n_{\bar{n}}} \overset{\text{i.i.d.}}{\sim} \mathcal{P}_{X|\bar{Y}=\mathrm{n}} := \gamma_\mathrm{n}\,\mathcal{P}_{X|Y=\mathrm{p}} + (1 - \gamma_\mathrm{n})\,\mathcal{P}_{X|Y=\mathrm{n}}.$ |
| CCN | $\{(\bar{y}_i, x_i)\}_{i=1}^{n} \overset{\text{i.i.d.}}{\sim} \mathcal{P}_{\bar{Y}=\bar{y}_i,X} := \sum_{k \in \{\mathrm{p},\mathrm{n}\}} \mathcal{P}_{\bar{Y}=\bar{y}_i|Y=k,X}\mathcal{P}_{Y=k,X}, \forall \bar{y}_i \in \{\mathrm{p},\mathrm{n}\}.$ |

## 3 A Framework for Risk Rewrite

We illustrate the proposed framework in this section. Its job is to provide a unified treatment and understanding of WSL. It consists of a formulation component and an analysis component. The analysis component suggests a generic methodology to solve the risk rewrite problem. Moreover, diving into the formulation component's logic, we can interpret multiple WSL formulations and the diverse risk rewrites from a single perspective.

Before introducing the framework, we first define several abstract notations that will be used throughout the paper. There are three main characters and one supporting character in the framework. The main characters are the vector of data-generating distributions $\bar{P}$, the vector of risk-defining distributions $P$, and the vector of base distributions $B$. The supporting character is the vector of loss functions $L$. The reason for using vectorized pseudonyms is that the proposed framework uses matrix multiplication as a basic mathematical operation. $\bar{P}$ contains distributions that produce the observational data. For instance, $\bar{P} = \left(\begin{smallmatrix} \mathcal{P}_\mathrm{P} \\ \mathcal{P}_\mathrm{U} \end{smallmatrix}\right)$ in PU learning (Table 2) and $\mathcal{P}_{\bar{S}=k,X}$ is the $k$-th entry of $\bar{P}$ in CL learning (Table 3). We use classification risk (1) to illustrate our framework. So $P$ consists of joint distributions $\mathcal{P}_{X,Y}$. We can look at Tables 2 and 3 and see that there are basic elements that define a data-generating distribution. These are class-conditionals $\mathcal{P}_{X|Y}$ in Table 2 and joint distributions $\mathcal{P}_{X,Y}$ in Table 3. Since these basic elements do not necessarily coincide with the entries of $P$, we denote them as $B$. $L$ consists of loss functions and its $k$-th entry is $\ell_{Y=k}(g(X))^2$.

### 3.1 The Formulation Component of the Framework

The construction of the formulation component is to study the connections among WSLs and provide a foundation for developing the generic methodology. We draw inspiration from Section 2.2. Each WSL formulation represents a type of weaken information of the joint distribution $\mathcal{P}_{Y,X}$ in supervised learning. For instance, unlabeled data discards the label information (Lu et al., 2020; 2021), the complementary-label is a label that cannot be the ground truth (Ishida et al., 2017; Yu et al., 2018), and the similarity encodes a comparative relationship of two ground truth labels (Bao et al., 2018; Shimada et al., 2021; Cao et al., 2021b). Thus, we are motivated to search for a general way to link data-generating distributions with the joint distribution.

We start by linking the data-generating distributions $\bar{P}$ and the base distributions $B$. This involves finding matrix correspondences to Tables 2 and 3. We assume that a matrix $M_\mathrm{corr}$ formalizes the link:

$$\bar{P} = M_\mathrm{corr}B. \tag{35}$$

Taking PU learning (4) for example, $M_\mathrm{corr}$ aims to connect $\bar{P} = \left(\begin{smallmatrix} \mathcal{P}_\mathrm{P} \\ \mathcal{P}_\mathrm{U} \end{smallmatrix}\right)$ with $B = \left(\begin{smallmatrix} \mathcal{P}_{X|Y=\mathrm{p}} \\ \mathcal{P}_{X|Y=\mathrm{n}} \end{smallmatrix}\right)$. To keep the framework as abstract as possible, we would like to defer the discussion of all other $\bar{P}$ and $B$ until we realize their corresponding $M_\mathrm{corr}$ in Section 4.

The matrix formulation has two advantages. First, it provides a unified way to characterize a wide range of WSL settings. By studying the entries of a matrix, we can easily link one WSL scenario to another to form

---

[2]We reserve $\bar{P}$, $P$, and $B$ for vectors of distributions and $L$ and $\bar{L}$ for vectors of loss functions. We address them as "the distributions" and "the losses" to avoid the verbose "the vector of distributions/losses."

reduction graphs of WSLs. As the first main topic of this work, Section 4 shows, for a given WSL setting, how to find the corresponding matrix $M_{\text{corr}}$, and Tables 7 – 9 summarize fifteen WSL settings covered by our matrix formulation and depict a reduction graph rooted from $M_{\text{corr}}$. The following subsection illustrates the second advantage of aiding the construction of a generic methodology for conducting risk rewrite.

### 3.2 The Analysis Component of the Framework

Note that the conventional expression (1) can be simplified, by the inner product, to be $R(g) = \int_{x \in \mathcal{X}} L^\top P \mathrm{d}x$. It is immediately possible to rewrite the risk under data-generating distributions $\bar{P}$ by showing $L^\top P = \bar{L}^\top \bar{P}$, where $\bar{L}$ is called the vector of corrected losses and its role will be clarified later. Therefore, it is imperative to establish the connection between $\bar{P}$ and $P$. The goal can be achieved by linking $B$ and $P$ since we have assumed that $\bar{P} = M_{\text{corr}}B$ in the previous subsection. Recall from the beginning of this section that the base distributions $B$ are of the forms $\mathcal{P}_{X|Y}$ or $\mathcal{P}_{X,Y}$, and the risk-defining distributions $P$ are of the form $\mathcal{P}_{X,Y}$. Given their label-relevant nature (i.e., they are either the joint distribution or the class-conditionals), we assume that there exists a transformation matrix $M_{\text{trsf}}$ that satisfies $B = M_{\text{trsf}}P$. Thus, (35) becomes

$$\bar{P} = M_{\text{corr}}M_{\text{trsf}}P. \tag{36}$$

Having the freedom to choose $M_{\text{trsf}}$ allows the framework to handle different base distributions $B$ and to adapt to various performance measures that define $P$, as we will discuss in Sections 5.1.1, 5.2.1, and 6.1, respectively.

The reason why connecting $P$ with $\bar{P}$ (36) helps the construction of the corrected losses is that if we manage to find a way to compensate for the combined effect of $M_{\text{corr}}$ and $M_{\text{trsf}}$, we can implement the compensation mechanism on the "corrected" losses $\bar{L}$. Specifically, suppose there exists a matrix $M_{\text{corr}}^\dagger$ satisfying

$$P = M_{\text{corr}}^\dagger \bar{P}. \tag{37}$$

Then, the corrected losses defined by

$$\bar{L}^\top := L^\top M_{\text{corr}}^\dagger \tag{38}$$

allows us to rephrase the classification risk as

$$\begin{aligned}
\int_{x \in \mathcal{X}} \bar{L}^\top \bar{P} \, \mathrm{d}x &= \int_{x \in \mathcal{X}} L^\top M_{\text{corr}}^\dagger \bar{P} \, \mathrm{d}x \\
&= \int_{x \in \mathcal{X}} L^\top P \, \mathrm{d}x = R(g),
\end{aligned} \tag{39}$$

providing a rewrite for $R(g)$ with respect to $\bar{P}$.

The above procedure describes a generic methodology for the risk rewrite problem. As the second main topic, we instantiate the framework by presenting the corresponding matrices $M_{\text{corr}}^\dagger$ and $M_{\text{trsf}}$ for each learning scenario in Section 5 to demonstrate its applicability.

### 3.3 Intuition of the Framework

The key equations discussed in Sections 3.1 and 3.2 are

$$\bar{P} \stackrel{(35)}{=} M_{\text{corr}}B \stackrel{(36)}{=} M_{\text{corr}}M_{\text{trsf}}P,$$

$$\bar{L}^\top \bar{P} \stackrel{(38)}{=} L^\top M_{\text{corr}}^\dagger M_{\text{corr}}M_{\text{trsf}}P \stackrel{(37)}{=} L^\top P.$$

The logic behind them is succinct and interpretive. First, from a formulation perspective, viewing matrix $M_{\text{corr}}$ as a contamination matrix that corrupts the base $B$ to become the contaminated $\bar{P}$ (35), we interpret this *contamination mechanism* as sacrificing certain information in exchange for certain saved costs or privacy, reflecting the essence underlying WSL formulations. In addition to formulating the data-generating

mechanism, the link between $B$ and the risk-defining distributions $P$ (36) connects $\bar{P}$ and $P$ to motivate the methodology design. This novel viewpoint of connecting the data distributions via the explicit two-stage formulation facilitates the unification work in this paper.

Second, regarding the methodological design, it becomes easier to devise a countermeasure when the connection between $\bar{P}$ and $P$ is in good shape. The design of $\bar{L}^\top = L^\top M^\dagger_{\text{corr}}$ involves $M^\dagger_{\text{corr}}$, which captures a common idea behind risk rewrite: Restoration of the risk-defining distributions and the original loss functions is accomplished by the *decontamination* (37) provided by $\bar{L}$. Furthermore, the instantiations of $\bar{L}^\top = L^\top M^\dagger_{\text{corr}}$ (38) justify that the apparently different forms of corrected losses reported in the literature (i.e., referred papers that contribute to Tables 4 and 5, and those referred to as recoveries in Section 5) essentially stem from $M^\dagger_{\text{corr}}$. In summary, the proposed framework is abstract and flexible enough that we use it in the current paper to formulate the contamination mechanisms and provide a generic methodology for a wide range of WSLs.

### 3.4 Building Blocks: The Inversion and the Marginal Chain Approaches

We describe two building blocks, the inversion method and the marginal chain method, that will be used to devise $M^\dagger_{\text{corr}}$ that satisfies (37) in each scenario we study later.

**Proposition 1** (The inversion method)**.** *Let $P$ and $\bar{P}$ be vectors. Suppose $\bar{P} = MP$ holds for an invertible matrix $M$. Then, choosing $M^\dagger_{\text{corr}} = M^{-1}$, we have $P = M^\dagger_{\text{corr}}\bar{P}$.*

*Proof.* For any invertible $M$, it is easy to see that, by assigning $M^\dagger_{\text{corr}} = M^{-1}$, one has

$$M^\dagger_{\text{corr}}\bar{P} = M^{-1}\bar{P} = M^{-1}MP = P.$$

$\square$

We remark that this simple strategy was adopted in many LNL works. A handful of related papers are Cid-Sueiro (2012), Blanchard & Scott (2014), Menon et al. (2015), van Rooyen & Williamson (2015), Patrini et al. (2017), van Rooyen & Williamson (2017), and Katz-Samuels et al. (2019). Hence, it can be applied to WSLs that are special cases of certain LNL scenarios.

**Proposition 2** (The marginal chain method)**.** *Let $Y = k \in [K]$ be a class label, where $[K]$ is the set of classes associated with the classification risk. Let $\mathcal{S} = \{s_1, s_2, \ldots, s_{|\mathcal{S}|}\} \subseteq 2^{[K]}$ be the set of class sets and $S$ be the random variable of the observational outcome. Denote*

$$P = \begin{pmatrix} \mathcal{P}_{Y=1,X} \\ \vdots \\ \mathcal{P}_{Y=K,X} \end{pmatrix} \text{ and } \bar{P} = \begin{pmatrix} \mathcal{P}_{S=s_1,X} \\ \vdots \\ \mathcal{P}_{S=s_{|\mathcal{S}|},X} \end{pmatrix}.$$

*Then,*

$$M = \begin{pmatrix} \mathcal{P}_{S=s_1|Y=1,X} & \mathcal{P}_{S=s_1|Y=2,X} & \cdots & \mathcal{P}_{S=s_1|Y=K,X} \\ \mathcal{P}_{S=s_2|Y=1,X} & \mathcal{P}_{S=s_2|Y=2,X} & \cdots & \mathcal{P}_{S=s_2|Y=K,X} \\ \vdots & \vdots & \ddots & \vdots \\ \mathcal{P}_{S=s_{|\mathcal{S}|}|Y=1,X} & \mathcal{P}_{S=s_{|\mathcal{S}|}|Y=2,X} & \cdots & \mathcal{P}_{S=s_{|\mathcal{S}|}|Y=K,X} \end{pmatrix} \tag{40}$$

*satisfies $\bar{P} = MP$, and*

$$M^\dagger_{\text{corr}} = \begin{pmatrix} \mathcal{P}_{Y=1|S=s_1,X} & \mathcal{P}_{Y=1|S=s_2,X} & \cdots & \mathcal{P}_{Y=1|S=s_{|\mathcal{S}|},X} \\ \mathcal{P}_{Y=2|S=s_1,X} & \mathcal{P}_{Y=2|S=s_2,X} & \cdots & \mathcal{P}_{Y=2|S=s_{|\mathcal{S}|},X} \\ \vdots & \vdots & \ddots & \vdots \\ \mathcal{P}_{Y=K|S=s_1,X} & \mathcal{P}_{Y=K|S=s_2,X} & \cdots & \mathcal{P}_{Y=K|S=s_{|\mathcal{S}|},X} \end{pmatrix} \tag{41}$$

*satisfies* $P = M_{\text{corr}}^{\dagger} \bar{P}$.

The role of $S$ is to represent a weak supervision that encodes some combinatorial information about the unobservable true label $Y$. We will discuss this concept in detail in Sections 4.2 and 5.2.

*Proof.* It suffices to show $\left(MP\right)_j = \bar{P}_j$ for any $j \in [|\mathcal{S}|]$. Taking the inner product of the $j$-th row of $M$ and $P$, we have

$$\sum_{k=1}^{K} \mathcal{P}_{S=s_j|Y=k,X} \mathcal{P}_{Y=k,X} = \sum_{k=1}^{K} \mathcal{P}_{S=s_j,Y=k,X} = \mathcal{P}_{S=s_j,X}$$

that verifies (40).

Next, we prove $P = M_{\text{corr}}^{\dagger} \bar{P}$ by showing $\left(M_{\text{corr}}^{\dagger} \bar{P}\right)_i = P_i$. For each $i \in [K]$,

$$
\begin{aligned}
\left(M_{\text{corr}}^{\dagger} \bar{P}\right)_i = \left(M_{\text{corr}}^{\dagger} MP\right)_i &= \sum_{j=1}^{|\mathcal{S}|} \mathcal{P}_{Y=i|S=s_j,X} \sum_{k=1}^{K} \mathcal{P}_{S=s_j|Y=k,X} \mathcal{P}_{Y=k,X} \\
&\overset{(a)}{=} \sum_{j=1}^{|\mathcal{S}|} \mathcal{P}_{Y=i|S=s_j,X} \mathcal{P}_{S=s_j,X} \\
&\overset{(b)}{=} \mathcal{P}_{Y=i,X} = P_i.
\end{aligned}
\tag{42}
$$

$\square$

Besides finding the inverse matrix, we propose a new approach called the *marginal chain* to achieve (37). The development of this approach begins with the observation that $\mathcal{P}_{S=s_j,X}$ in $\bar{P} = MP$ is a distribution where $Y$ is marginalized out. It inspires an idea that one could perform another marginalization to restore the original distribution $\mathcal{P}_{Y,X}$; specifically, by marginalizing out $S$. The design of $M_{\text{corr}}^{\dagger}$ in (41) aims to carry out the idea. As shown by (a) and (b) in the proof, two consecutive marginalization steps on $Y$ and then $S$ give the name of the marginal chain.

Both the inversion and marginal chain methods have strengths and weaknesses. The inversion method only requires $P$ as a real vector but needs the invertible assumption on the contamination matrix $M$. In contrast, the marginal chain method exploits that $P$, in fact, is a distributional vector, allowing it to find a decontamination matrix $M_{\text{corr}}^{\dagger}$ even for a non-invertible $M$. A restriction of the marginal chain method is that the construction of $M_{\text{corr}}^{\dagger}$ is regulated by probability equations.

We are ready to justify the proposed framework through the following two sections. Section 4 discusses weakly supervised scenarios that can be subsumed by the formulation component (35). Section 5 verifies the analysis component by instantiating (38) to conduct the risk rewrite for each scenario mentioned in Section 4. In both sections, we divide the scenarios into three categories. The first two are WSLs that can be viewed as special cases in either the prevalent MCD or CCN settings. The third category contains confidence-based scenarios. The notations listed in Table 1 will still be functional. For all notations and their abbreviations required in the coming sections, please refer to Appendix B.

## 4 Contamination as Weak Supervision

In this section, we instantiate the contamination matrix for each weakly supervised scenario listed in Table 2 and Table 3. Tables 7 – 9 summarize the contamination matrices developed in this section. Each table also represents a reduction graph of WSL settings. These reduction graphs cluster WSL settings into three main categories, providing a hierarchy of relationships. With this hierarchy, we can understand, compare with, and relate to different settings or even grow the hierarchy by adding new branches. Next are the notations

for reading the graphs. For two contamination mechanisms, U and V, we use $M_\mathrm{U} \to M_\mathrm{V}$ to denote "$M_\mathrm{U}$ is reduced to $M_\mathrm{V}$" or "$M_\mathrm{U}$ is realized as $M_\mathrm{V}$", and $M_\mathrm{U} \rightsquigarrow M_\mathrm{V}$ means "$M_\mathrm{U}$ is generalized to $M_\mathrm{V}$".

The proposed framework provides a generic strategy for formulating multiple weakly supervised scenarios. Thus, the proofs will have a certain degree of similarity. To avoid repeating similar proofs, we provide proofs that appear for the first time. For auxiliary lemmas and results whose proofs are similar to the previous ones, we refer to the omitted proofs in Appendix C. In particular, the omitted proofs in Section 4.1 can be found in Appendix C.1, and those in Section 4.2 can be found in Appendix C.2.

Table 7: Contamination matrices of MCD category in Section 4.1.

| WSLs | Entry Parameter | Contamination Matrix | Reduction path |
|---|---|---|---|
| MCD | $\gamma_\mathrm{p}, \gamma_\mathrm{n}$ | $M_\mathrm{MCD}$ (45) | $M_\mathrm{corr} \to M_\mathrm{MCD}$ |
| UU | $\gamma_1, \gamma_2$ | $M_\mathrm{UU}$ (49) | $M_\mathrm{corr} \to M_\mathrm{UU} \approx M_\mathrm{MCD}$ |
| PU | $\gamma_1 = 0, \gamma_2 = \pi_\mathrm{p}$ | $M_\mathrm{PU}$ (50) | $M_\mathrm{UU} \to M_\mathrm{PU}$ |
| SU | $\gamma_1 = \frac{\pi_\mathrm{n}^2}{\pi_\mathrm{p}^2 + \pi_\mathrm{n}^2}, \gamma_2 = \pi_\mathrm{p}$ | $M_\mathrm{SU}$ (51) | $M_\mathrm{UU} \to M_\mathrm{SU}$ |
| Pcomp | $\gamma_1 = \frac{\pi_\mathrm{n}^2}{\pi_\mathrm{p} + \pi_\mathrm{n}^2}, \gamma_2 = \frac{\pi_\mathrm{p}^2}{\pi_\mathrm{p}^2 + \pi_\mathrm{n}}$ | $M_\mathrm{Pcomp}$ (52) | $M_\mathrm{UU} \to M_\mathrm{Pcomp}$ |
| DU | $\gamma_1 = 1/2, \gamma_2 = \pi_\mathrm{p}$ | $M_\mathrm{DU}$ (53) | $M_\mathrm{UU} \to M_\mathrm{DU}$ |
| SD | $\gamma_1 = \frac{\pi_\mathrm{n}^2}{\pi_\mathrm{p}^2 + \pi_\mathrm{n}^2}, \gamma_2 = 1/2$ | $M_\mathrm{SD}$ (54) | $M_\mathrm{UU} \to M_\mathrm{SD}$ |
| Sconf | – | $M_\mathrm{Sconf}$ (55) | $M_\mathrm{corr} \to M_\mathrm{Sconf}$ |

Table 8: Contamination matrices of CCN category in Section 4.2.

| WSLs | Entry Parameter | Contamination Matrix | Reduction path |
|---|---|---|---|
| CCN | $\mathcal{P}_{\bar{Y}\mid Y,X}$ (59) | $M_\mathrm{CCN}$ (60) | $M_\mathrm{corr} \to M_\mathrm{CCN}$ |
| Generalized CCN | $\mathcal{P}_{S\mid Y,X}$ (61) | $M_\mathrm{gCCN}$ (64) | $M_\mathrm{corr} \to M_\mathrm{CCN} \rightsquigarrow M_\mathrm{gCCN}$ |
| PPL | $C(S,X)\mathbb{I}\left[Y \in S\right]$ (65) | $M_\mathrm{PPL}$ (66) | $M_\mathrm{gCCN} \to M_\mathrm{PPL}$ |
| PCPL | $\frac{1}{2^{K-1}-1}\mathbb{I}\left[Y \in S\right]$ | $M_\mathrm{PCPL}$ (68) | $M_\mathrm{gCCN} \to M_\mathrm{PPL} \to M_\mathrm{PCPL}$ |
| MCL | $\frac{q_{|\bar{S}|}}{\binom{K-1}{|\bar{S}|}}\mathbb{I}\left[Y \notin \bar{S}\right]$ (75) | $M_\mathrm{MCL}$ (71) | $M_\mathrm{gCCN} \to M_\mathrm{PPL} \to M_\mathrm{MCL}$ |
| CL | $\lvert S\rvert = 1, \frac{1}{K-1}\mathbb{I}\left[Y \in S\right]$ | $M_\mathrm{CL}$ (76) | $M_\mathrm{gCCN} \to M_\mathrm{PPL} \to M_\mathrm{MCL} \to M_\mathrm{CL}$ |

Table 9: Contamination matrices of confidence-based category in Section 4.3.

| WSLs | Entry Parameter | Contamination Matrix | Reduction path |
|---|---|---|---|
| Sub-Conf | $\frac{\mathcal{P}_{Y \in \mathcal{Y}_\mathrm{s}\mid X}}{\mathcal{P}_{Y = k\mid X}}$ | $M_\mathrm{Sub}$ (80) | $M_\mathrm{corr} \to M_\mathrm{Sub}$ |
| SC | $\mathcal{Y}_\mathrm{s} = \{y_\mathrm{s}\}$ in $M_\mathrm{Sub}$ | $M_\mathrm{SC}$ (81) | $M_\mathrm{Sub} \to M_\mathrm{SC}$ |
| Pconf | $K = 2, y_\mathrm{s} = \mathrm{p}$ in $M_\mathrm{SC}$ | $M_\mathrm{Pconf}$ (82) | $M_\mathrm{Sub} \to M_\mathrm{SC} \to M_\mathrm{Pconf}$ |
| Soft | $\frac{1}{\mathcal{P}_{Y = k\mid X}}$ | $M_\mathrm{Soft}$ (84) | $M_\mathrm{Sub} \to M_\mathrm{Soft}$ |

### 4.1 MCD Scenarios

As listed in Table 6, in binary classification, the MCD model (Menon et al., 2015) corrupts the clean class-conditionals $\mathcal{P}_{X|Y=p}$ and $\mathcal{P}_{X|Y=n}$ via parameters $\gamma_p$ and $\gamma_n$ as follows:

$$
\begin{aligned}
\mathcal{P}_{X|\bar{Y}=p} &:= (1 - \gamma_p)\,\mathcal{P}_{X|Y=p} + \gamma_p\,\mathcal{P}_{X|Y=n}, \\
\mathcal{P}_{X|\bar{Y}=n} &:= \gamma_n\,\mathcal{P}_{X|Y=p} + (1 - \gamma_n)\,\mathcal{P}_{X|Y=n},
\end{aligned}
\tag{43}
$$

where $\gamma_p, \gamma_n \in [0,1]$ and $\gamma_p + \gamma_n < 1$. Viewing the contamination targets $\mathcal{P}_{X|Y=p}$ and $\mathcal{P}_{X|Y=n}$ as the base distributions

$$
B := \begin{pmatrix} \mathcal{P}_{X|Y=p} \\ \mathcal{P}_{X|Y=n} \end{pmatrix}
$$

and denoting the vector of data-generating distributions as

$$
\bar{P} := \begin{pmatrix} \mathcal{P}_{X|\bar{Y}=p} \\ \mathcal{P}_{X|\bar{Y}=n} \end{pmatrix},
$$

we can express (43) in the following matrix form

$$
\begin{pmatrix} \mathcal{P}_{X|\bar{Y}=p} \\ \mathcal{P}_{X|\bar{Y}=n} \end{pmatrix} = \begin{pmatrix} 1 - \gamma_p & \gamma_p \\ \gamma_n & 1 - \gamma_n \end{pmatrix} \begin{pmatrix} \mathcal{P}_{X|Y=p} \\ \mathcal{P}_{X|Y=n} \end{pmatrix}.
\tag{44}
$$

Comparing (44) with $\bar{P} = M_{\mathrm{corr}} B$ (35), we find that the contamination matrix $M_{\mathrm{corr}}$ is realized as

$$
M_{\mathrm{MCD}} := \begin{pmatrix} 1 - \gamma_p & \gamma_p \\ \gamma_n & 1 - \gamma_n \end{pmatrix}
\tag{45}
$$

in the MCD setting.

#### 4.1.1 Unlabeled-Unlabeled (UU) Learning (Lu et al., 2019)

Next, we show how to characterize UU learning by a contamination matrix. Naming

$$
\pi_p\,\mathcal{P}_{X|Y=p} + \pi_n\,\mathcal{P}_{X|Y=n}
$$

as $\mathcal{P}_U$ is feasible since $\pi_p\,\mathcal{P}_{X|Y=p} + \pi_n\,\mathcal{P}_{X|Y=n} = \mathcal{P}_X$ generates data that statistically equals to data sampled from $\mathcal{P}_{Y,X}$ with labels removed. Viewing $\pi_p$ as the mixture rate of samples from $\mathcal{P}_{X|Y=p}$ and $\mathcal{P}_{X|Y=n}$, $\mathcal{P}_U$ is parameterized by $\pi_p$. Therefore, we can interpret (8), recalled as follows, as formulating two unlabeled data distributions w.r.t. mixture rates $(1 - \gamma_1)$ and $\gamma_2$, respectively:

$$
\begin{aligned}
\mathcal{P}_{U_1} &= (1 - \gamma_1)\,\mathcal{P}_{X|Y=p} + \gamma_1\,\mathcal{P}_{X|Y=n}, \\
\mathcal{P}_{U_2} &= \gamma_2\,\mathcal{P}_{X|Y=p} + (1 - \gamma_2)\,\mathcal{P}_{X|Y=n}.
\end{aligned}
$$

Taking the class-conditionals as the base distributions

$$
B := \begin{pmatrix} \mathcal{P}_{X|Y=p} \\ \mathcal{P}_{X|Y=n} \end{pmatrix}
\tag{46}
$$

and converting (8) to the matrix form, we express the data-generating distributions of UU learning

$$
\bar{P} := \begin{pmatrix} \mathcal{P}_{U_1} \\ \mathcal{P}_{U_2} \end{pmatrix}
\tag{47}
$$

as

$$\begin{pmatrix} \mathcal{P}_{\mathrm{U}_1} \\ \mathcal{P}_{\mathrm{U}_2} \end{pmatrix} = \begin{pmatrix} 1 - \gamma_1 & \gamma_1 \\ \gamma_2 & 1 - \gamma_2 \end{pmatrix} \begin{pmatrix} \mathcal{P}_{X|Y=\mathrm{p}} \\ \mathcal{P}_{X|Y=\mathrm{n}} \end{pmatrix}, \tag{48}$$

and we arrive at the following lemma.

**Lemma 3.** *Let $B$ (46) be the base distributions and $\bar{P}$ (47) be the data-generating distributions. For $\gamma_1, \gamma_2 \in [0, 1]$ such that $\gamma_1 + \gamma_2 \neq 1$, the contamination matrix*

$$M_{\mathrm{UU}} := \begin{pmatrix} 1 - \gamma_1 & \gamma_1 \\ \gamma_2 & 1 - \gamma_2 \end{pmatrix} \tag{49}$$

*characterizes the data-generating process of UU learning (8) via (48).*

Comparing (48) with the formulation framework $\bar{P} = M_{\mathrm{corr}} B$ (35), we see that in UU learning, $M_{\mathrm{corr}}$ is realized as $M_{\mathrm{UU}}$:

$$M_{\mathrm{corr}} \rightarrow M_{\mathrm{UU}}.$$

Like MCD, we assume $\gamma_1 + \gamma_2 \neq 1$. Our assumption is equivalent to that of MCD since the case of swapping $P_{\mathrm{corr}}$ and $Q_{\mathrm{corr}}$ in Menon et al. (2015) corresponds to $\gamma_1 + \gamma_2 > 1$ in our case. For details, refer to the discussion in Section 2.2 of Menon et al. (2015). The need for $\gamma_1 + \gamma_2 \neq 1$ can be explained by examining the entries in $M_{\mathrm{UU}}$. The constraint $\gamma_1 + \gamma_2 \neq 1$ guarantees distinct rows in $M_{\mathrm{UU}}$, implying the observed data sets are sampled from two distinct distributions. On the contrary, allowing $\gamma_1 + \gamma_2 = 1$ ends up observing one unlabeled data set (i.e., $\mathcal{P}_{\mathrm{U}_1} = \mathcal{P}_{\mathrm{U}_2}$) since $1 - \gamma_1 = \gamma_2$. Lu et al. (2019) proved in Section 3 that it is impossible to conduct a risk rewrite if one only observes one unlabeled data set.

Assigning $\gamma_1 = \gamma_{\mathrm{p}}$ and $\gamma_2 = \gamma_{\mathrm{n}}$ implies that MCD and UU have essentially the same data-generating process from the contamination perspective, as (44) and (48) have the identical right-hand sides (i.e., the same contamination targets and the same contamination matrix). However, they bear different meanings in respective research topics (i.e., distinct notions on the left-hand sides of the equations): In MCD, one still observes data with labels, nonetheless noisy, while in the UU setting, one observes two distinct unlabeled data sets. We use "$\approx$" to denote their relation in the UU row of Table 7.

Connecting UU learning with MCD, and later the generalized CCN with CCN in Section 4.2.1, allows us to categorize WSLs from the LNL perspective into Sections 4.1 and 4.2. In the rest of this subsection, we collect WSLs whose base distributions are class-conditionals and show $M_{\mathrm{UU}}$ instantiates their formulations via respective assignments of $\gamma_1$ and $\gamma_2$.

### 4.1.2 Positive-Unlabeled (PU) Learning (Kiryo et al., 2017)

Recall from (4) that $\mathcal{P}_{\mathrm{P}} = \mathcal{P}_{X|Y=\mathrm{p}}$ and $\mathcal{P}_{\mathrm{U}} = \mathcal{P}_X$. The following lemma describes the contamination matrix of PU learning.

**Lemma 4.** *Let $B$ (46) be the base distributions and*

$$\bar{P} := \begin{pmatrix} \mathcal{P}_{\mathrm{P}} \\ \mathcal{P}_{\mathrm{U}} \end{pmatrix}$$

*be the data-generating distributions. Define the contamination matrix*

$$M_{\mathrm{PU}} := \begin{pmatrix} 1 & 0 \\ \pi_{\mathrm{p}} & \pi_{\mathrm{n}} \end{pmatrix}. \tag{50}$$

*Then, $\bar{P} = M_{\mathrm{PU}} B$, and $M_{\mathrm{PU}}$ characterizes the data-generating process of PU learning (4).*

*Proof.* We apply the same proof strategy as in Lemma 3. By definitions,

$$
M_{\mathrm{PU}}B = \begin{pmatrix} 1 & 0 \\ \pi_{\mathrm{p}} & \pi_{\mathrm{n}} \end{pmatrix} \begin{pmatrix} \mathcal{P}_{X|Y=\mathrm{p}} \\ \mathcal{P}_{X|Y=\mathrm{n}} \end{pmatrix} = \begin{pmatrix} 1 \cdot \mathcal{P}_{X|Y=\mathrm{p}} + 0 \cdot \mathcal{P}_{X|Y=\mathrm{n}} \\ \pi_{\mathrm{p}} \cdot \mathcal{P}_{X|Y=\mathrm{p}} + \pi_{\mathrm{n}} \cdot \mathcal{P}_{X|Y=\mathrm{n}} \end{pmatrix}.
$$

Since $1 \cdot \mathcal{P}_{X|Y=\mathrm{p}} + 0 \cdot \mathcal{P}_{X|Y=\mathrm{n}} = \mathcal{P}_{X|Y=\mathrm{p}}$ and $\pi_{\mathrm{p}} \cdot \mathcal{P}_{X|Y=\mathrm{p}} + \pi_{\mathrm{n}} \cdot \mathcal{P}_{X|Y=\mathrm{n}} = \mathcal{P}_X$, we obtain $M_{\mathrm{PU}}B = \bar{P}$. ☐

Comparing with (35), we see that the contamination matrix $M_{\mathrm{corr}}$ is instantiated as $M_{\mathrm{PU}}$ (50) in PU learning. Further, $M_{\mathrm{PU}}$ can be obtained by assigning $\gamma_1 = 0$ and $\gamma_2 = \pi_{\mathrm{p}}$ in $M_{\mathrm{UU}}$ (49), and hence, we obtain the reduction path

$$
M_{\mathrm{corr}} \to M_{\mathrm{UU}} \to M_{\mathrm{PU}}.
$$

### 4.1.3 Similar-Unlabeled (SU) Learning (Bao et al., 2018)

Recall $\mathcal{P}_{\mathrm{S}}$ (10) generates the pair of data points $(x, x')$ who share the same label. In addition to the pairwise distribution $\mathcal{P}_{\mathrm{S}}$, a pointwise distribution

$$
\mathcal{P}_{\tilde{\mathrm{S}}} = \frac{\pi_{\mathrm{p}}^2 \mathcal{P}_{X|Y=\mathrm{p}} + \pi_{\mathrm{n}}^2 \mathcal{P}_{X|Y=\mathrm{p}}}{\pi_{\mathrm{p}}^2 + \pi_{\mathrm{n}}^2}
$$

is also defined for single data point $x$ (Bao et al., 2018, Lemma 1). Therefore, we choose $\mathcal{P}_{\tilde{\mathrm{S}}}$ as the data-generating distribution when constructing the contamination matrix in the following lemma.

**Lemma 5.** *Let B (46) be the base distributions and*

$$
\bar{P} := \begin{pmatrix} \mathcal{P}_{\tilde{\mathrm{S}}} \\ \mathcal{P}_{\mathrm{U}} \end{pmatrix}.
$$

*Then, the contamination matrix*

$$
M_{\mathrm{SU}} := \begin{pmatrix} \frac{\pi_{\mathrm{p}}^2}{\pi_{\mathrm{p}}^2 + \pi_{\mathrm{n}}^2} & \frac{\pi_{\mathrm{n}}^2}{\pi_{\mathrm{p}}^2 + \pi_{\mathrm{n}}^2} \\ \pi_{\mathrm{p}} & \pi_{\mathrm{n}} \end{pmatrix}, \tag{51}
$$

*which satisfies $\bar{P} = M_{\mathrm{SU}}B$, characterizes the data-generating distributions $\bar{P}$.*

Further, $M_{\mathrm{SU}}$ can be obtained by assigning $\gamma_1 = \frac{\pi_{\mathrm{n}}^2}{\pi_{\mathrm{p}}^2 + \pi_{\mathrm{n}}^2}$ and $\gamma_2 = \pi_{\mathrm{p}}$ in $M_{\mathrm{UU}}$ (49), and hence, we obtain the reduction path

$$
M_{\mathrm{corr}} \to M_{\mathrm{UU}} \to M_{\mathrm{SU}}.
$$

### 4.1.4 Pairwise Comparison (Pcomp) Learning (Feng et al., 2021)

In SU learning, we formulate the pointwise data-generating distributions $\mathcal{P}_{\tilde{\mathrm{S}}}$ and $\mathcal{P}_{\mathrm{U}}$; likewise, we use the following pointwise distributions of $\mathcal{P}_{\mathrm{PC}}$ (16) to formulate Pcomp learning:

$$
\begin{aligned}
\mathcal{P}_{\mathrm{Sup}} &:= \int_{x' \in \mathcal{X}} \mathcal{P}_{\mathrm{PC}} \, \mathrm{d}x' = \frac{\pi_{\mathrm{p}} \mathcal{P}_{X|Y=\mathrm{p}} + \pi_{\mathrm{n}}^2 \mathcal{P}_{X|Y=\mathrm{n}}}{\pi_{\mathrm{p}} + \pi_{\mathrm{n}}^2}, \\
\mathcal{P}_{\mathrm{Inf}} &:= \int_{x \in \mathcal{X}} \mathcal{P}_{\mathrm{PC}} \, \mathrm{d}x = \frac{\pi_{\mathrm{p}}^2 \mathcal{P}_{X'|Y=\mathrm{p}} + \pi_{\mathrm{n}} \mathcal{P}_{X'|Y=\mathrm{n}}}{\pi_{\mathrm{p}}^2 + \pi_{\mathrm{n}}}.
\end{aligned}
$$

**Lemma 6.** *Let B (46) be the base distributions and*

$$\bar{P} := \begin{pmatrix} \mathcal{P}_{\mathrm{Sup}} \\ \mathcal{P}_{\mathrm{Inf}} \end{pmatrix}.$$

*Then, the contamination matrix*

$$M_{\mathrm{Pcomp}} := \begin{pmatrix} \frac{\pi_{\mathrm{p}}}{\pi_{\mathrm{p}}+\pi_{\mathrm{n}}^2} & \frac{\pi_{\mathrm{n}}^2}{\pi_{\mathrm{p}}+\pi_{\mathrm{n}}^2} \\ \frac{\pi_{\mathrm{p}}^2}{\pi_{\mathrm{p}}^2+\pi_{\mathrm{n}}} & \frac{\pi_{\mathrm{n}}}{\pi_{\mathrm{p}}^2+\pi_{\mathrm{n}}} \end{pmatrix}, \tag{52}$$

*which satisfies $\bar{P} = M_{\mathrm{Pcomp}}B$, characterizes the data-generating distributions $\bar{P}$.*

Further, $M_{\mathrm{Pcomp}}$ can be obtained by assigning $\gamma_1 = \frac{\pi_{\mathrm{n}}^2}{\pi_{\mathrm{p}}+\pi_{\mathrm{n}}^2}$ and $\gamma_2 = \frac{\pi_{\mathrm{p}}^2}{\pi_{\mathrm{p}}^2+\pi_{\mathrm{n}}}$ in $M_{\mathrm{UU}}$ (49), and hence, we obtain the reduction path

$$M_{\mathrm{corr}} \to M_{\mathrm{UU}} \to M_{\mathrm{Pcomp}}.$$

### 4.1.5 Similar-dissimilar-unlabeled (SDU) Learning (Shimada et al., 2021)

Dissimilar-unlabeled (DU) learning and similar-dissimilar (SD) learning are two critical components of SDU learning. Hence, we present the matrix formulations of $M_{\mathrm{DU}}$ and $M_{\mathrm{SD}}$. Similar to the strategy taken in Sections 4.1.3 and 4.1.4, we use the following pointwise distribution

$$\mathcal{P}_{\tilde{\mathrm{D}}} = \int_{x'} \mathcal{P}_{\mathrm{D}}\, \mathrm{d}x' = \frac{\mathcal{P}_{x|Y=\mathrm{p}} + \mathcal{P}_{x|Y=\mathrm{n}}}{2}$$

(Shimada et al., 2021, (36) in Appendix A.1) in the following formulations.

We formulate the contamination matrix of DU learning via the following lemma.

**Lemma 7.** *Let B (46) be the base distributions and*

$$\bar{P} = \begin{pmatrix} \mathcal{P}_{\tilde{\mathrm{D}}} \\ \mathcal{P}_{\mathrm{U}} \end{pmatrix}.$$

*Then, the contamination matrix*

$$M_{\mathrm{DU}} = \begin{pmatrix} 1/2 & 1/2 \\ \pi_{\mathrm{p}} & \pi_{\mathrm{n}} \end{pmatrix}, \tag{53}$$

*which satisfies $\bar{P} = M_{\mathrm{DU}}B$, characterizes the data-generating distributions $\bar{P}$.*

Furthermore, since $M_{\mathrm{UU}}$ (49) reduces to $M_{\mathrm{DU}}$ by assigning $\gamma_1 = 1/2$ and $\gamma_2 = \pi_{\mathrm{p}}$, we have the reduction path

$$M_{\mathrm{corr}} \to M_{\mathrm{UU}} \to M_{\mathrm{DU}}.$$

The next lemma formulates the contamination matrix of SD learning.

**Lemma 8.** *Let B (46) be the base distributions and*

$$\bar{P} = \begin{pmatrix} \mathcal{P}_{\tilde{\mathrm{S}}} \\ \mathcal{P}_{\tilde{\mathrm{D}}} \end{pmatrix}.$$

*Then, the contamination matrix*

$$M_{\mathrm{SD}} = \begin{pmatrix} \frac{\pi_{\mathrm{p}}^2}{\pi_{\mathrm{p}}^2+\pi_{\mathrm{n}}^2} & \frac{\pi_{\mathrm{n}}^2}{\pi_{\mathrm{p}}^2+\pi_{\mathrm{n}}^2} \\[2mm] 1/2 & 1/2 \end{pmatrix} \tag{54}$$

*which satisfies $\bar{P} = M_{\mathrm{SD}}B$, characterizes the data-generating distributions $\bar{P}$.*

Moreover, because $M_{\mathrm{UU}}$ (49) reduces to $M_{\mathrm{SD}}$ via $\gamma_1 = \frac{\pi_{\mathrm{n}}^2}{\pi_{\mathrm{p}}^2+\pi_{\mathrm{n}}^2}$ and $\gamma_2 = 1/2$, we obtain the reduction path

$$M_{\mathrm{corr}} \to M_{\mathrm{UU}} \to M_{\mathrm{SD}}.$$

### 4.1.6 Similarity-Confidence (Sconf) Learning (Cao et al., 2021b)

Recall from the Sconf setting (18) that $(x, x')$ is a pair of data sampled i.i.d. from $\mathcal{P}_{X,X'} := \mathcal{P}_X \mathcal{P}_{X'}$. On seeing $\mathcal{P}_X$, one might wonder if it is sufficient to express the data-generating distribution simply as $\mathcal{P}_X = \mathcal{P}_{Y=\mathrm{p},X} + \mathcal{P}_{Y=\mathrm{n},X}$. This approach, however correct, does not consider all available information in the Sconf setting. Similar to $M_{\mathrm{UU}}$ that uses parameters $\gamma_1$ and $\gamma_2$ to characterize the data-generating process in UU learning, we use the following lemma that includes the confidence $r(x, x') := \mathcal{P}_{y=y'|x,x'}$ to characterize Sconf learning. Let us abbreviate $r(X, X')$ as $r$, $\mathcal{P}_{X|Y=\mathrm{p}}$ as $\mathcal{P}_{X|\mathrm{p}}$, and $\mathcal{P}_{X'|Y=\mathrm{n}}$ as $\mathcal{P}_{X'|\mathrm{n}}$.

**Lemma 9.** *Assume $\pi_{\mathrm{p}} \neq 1/2$. Let $B$ (46) be the base distributions and*

$$\bar{P} := \begin{pmatrix} \mathcal{P}_X \mathcal{P}_{X'} \\ \mathcal{P}_X \mathcal{P}_{X'} \end{pmatrix}.$$

*Then, the contamination matrix*

$$M_{\mathrm{Sconf}} := \begin{pmatrix} \frac{\pi_{\mathrm{p}}\left(\pi_{\mathrm{p}}^2\mathcal{P}_{X'|\mathrm{p}}-\pi_{\mathrm{n}}^2\mathcal{P}_{X'|\mathrm{n}}\right)}{r-\pi_{\mathrm{n}}} & \frac{\pi_{\mathrm{p}}\left(\pi_{\mathrm{n}}^2\mathcal{P}_{X'|\mathrm{n}}-\pi_{\mathrm{n}}^2\mathcal{P}_{X'|\mathrm{p}}\right)}{r-\pi_{\mathrm{n}}} \\[3mm] \frac{\pi_{\mathrm{n}}\left(\pi_{\mathrm{p}}^2\mathcal{P}_{X'|\mathrm{n}}-\pi_{\mathrm{p}}^2\mathcal{P}_{X'|\mathrm{p}}\right)}{\pi_{\mathrm{p}}-r} & \frac{\pi_{\mathrm{n}}\left(\pi_{\mathrm{p}}^2\mathcal{P}_{X'|\mathrm{p}}-\pi_{\mathrm{n}}^2\mathcal{P}_{X'|\mathrm{n}}\right)}{\pi_{\mathrm{p}}-r} \end{pmatrix} \tag{55}$$

*which satisfies $\bar{P} = M_{\mathrm{Sconf}}B$, characterizes the data-generating distributions $\bar{P}$.*

*Proof.* Note that once

$$\left(\frac{r-\pi_{\mathrm{n}}}{\pi_{\mathrm{p}}}\right)\mathcal{P}_X\mathcal{P}_{X'} = \left(\pi_{\mathrm{p}}^2\mathcal{P}_{X'|\mathrm{p}} - \pi_{\mathrm{n}}^2\mathcal{P}_{X'|\mathrm{n}}\right)\mathcal{P}_{X|\mathrm{p}} + \left(\pi_{\mathrm{n}}^2\mathcal{P}_{X'|\mathrm{n}} - \pi_{\mathrm{n}}^2\mathcal{P}_{X'|\mathrm{p}}\right)\mathcal{P}_{X|\mathrm{n}} \tag{56}$$

and

$$\left(\frac{\pi_{\mathrm{p}}-r}{\pi_{\mathrm{n}}}\right)\mathcal{P}_X\mathcal{P}_{X'} = \left(\pi_{\mathrm{p}}^2\mathcal{P}_{X'|\mathrm{n}} - \pi_{\mathrm{p}}^2\mathcal{P}_{X'|\mathrm{p}}\right)\mathcal{P}_{X|\mathrm{p}} + \left(\pi_{\mathrm{p}}^2\mathcal{P}_{X'|\mathrm{p}} - \pi_{\mathrm{n}}^2\mathcal{P}_{X'|\mathrm{n}}\right)\mathcal{P}_{X|\mathrm{n}}, \tag{57}$$

is obtained, reorganizing the terms gives

$$\begin{pmatrix} \mathcal{P}_X\mathcal{P}_{X'} \\ \mathcal{P}_X\mathcal{P}_{X'} \end{pmatrix} = \begin{pmatrix} \frac{\pi_{\mathrm{p}}\left(\pi_{\mathrm{p}}^2\mathcal{P}_{X'|\mathrm{p}}-\pi_{\mathrm{n}}^2\mathcal{P}_{X'|\mathrm{n}}\right)}{r-\pi_{\mathrm{n}}} & \frac{\pi_{\mathrm{p}}\left(\pi_{\mathrm{n}}^2\mathcal{P}_{X'|\mathrm{n}}-\pi_{\mathrm{n}}^2\mathcal{P}_{X'|\mathrm{p}}\right)}{r-\pi_{\mathrm{n}}} \\[3mm] \frac{\pi_{\mathrm{n}}\left(\pi_{\mathrm{p}}^2\mathcal{P}_{X'|\mathrm{n}}-\pi_{\mathrm{p}}^2\mathcal{P}_{X'|\mathrm{p}}\right)}{\pi_{\mathrm{p}}-r} & \frac{\pi_{\mathrm{n}}\left(\pi_{\mathrm{p}}^2\mathcal{P}_{X'|\mathrm{p}}-\pi_{\mathrm{n}}^2\mathcal{P}_{X'|\mathrm{n}}\right)}{\pi_{\mathrm{p}}-r} \end{pmatrix} \begin{pmatrix} \mathcal{P}_{X|\mathrm{p}} \\ \mathcal{P}_{X|\mathrm{n}} \end{pmatrix} \tag{58}$$

and finishes the proof.

Therefore, we will focus on proving (56) and (57). According to (2) of Cao et al. (2021b), the confidence $r(X, X')$, measuring how likely $X$ and $X'$ share the same label, is shown to be

$$r = r(X, X') = \frac{\pi_{\mathrm{p}}^2\mathcal{P}_{X|\mathrm{p}}\mathcal{P}_{X'|\mathrm{p}} + \pi_{\mathrm{n}}^2\mathcal{P}_{X|\mathrm{n}}\mathcal{P}_{X'|\mathrm{n}}}{\mathcal{P}_X\mathcal{P}_{X'}}.$$

It implies

$$r\mathcal{P}_X\mathcal{P}_{X'} = \pi_\mathrm{p}^2 \mathcal{P}_{X|\mathrm{p}}\mathcal{P}_{X'|\mathrm{p}} + \pi_\mathrm{n}^2 \mathcal{P}_{X|\mathrm{n}}\mathcal{P}_{X'|\mathrm{n}}$$

and

$$(1-r)\mathcal{P}_X\mathcal{P}_{X'} = \pi_\mathrm{p}\pi_\mathrm{n} \left( \mathcal{P}_{X|\mathrm{p}}\mathcal{P}_{X'|\mathrm{n}} + \mathcal{P}_{X|\mathrm{n}}\mathcal{P}_{X'|\mathrm{p}} \right).$$

If $\pi_\mathrm{p} \neq 1/2$, $\pi_\mathrm{p} - r \neq 0$ and $r - \pi_\mathrm{n} \neq 0$. As a result, (56) is achieved as follows

$$
\begin{aligned}
\left( \frac{r - \pi_\mathrm{n}}{\pi_\mathrm{p}} \right) \mathcal{P}_X\mathcal{P}_{X'} &= \left( r - \frac{\pi_\mathrm{n}}{\pi_\mathrm{p}}(1-r) \right) \mathcal{P}_X\mathcal{P}_{X'} \\
&= \pi_\mathrm{p}^2 \mathcal{P}_{X|\mathrm{p}}\mathcal{P}_{X'|\mathrm{p}} + \pi_\mathrm{n}^2 \mathcal{P}_{X|\mathrm{n}}\mathcal{P}_{X'|\mathrm{n}} - \frac{\pi_\mathrm{n}}{\pi_\mathrm{p}}\pi_\mathrm{p}\pi_\mathrm{n} \left( \mathcal{P}_{X|\mathrm{p}}\mathcal{P}_{X'|\mathrm{n}} + \mathcal{P}_{X|\mathrm{n}}\mathcal{P}_{X'|\mathrm{p}} \right) \\
&= \pi_\mathrm{p}^2 \mathcal{P}_{X|\mathrm{p}}\mathcal{P}_{X'|\mathrm{p}} - \pi_\mathrm{n}^2 \mathcal{P}_{X|\mathrm{p}}\mathcal{P}_{X'|\mathrm{n}} + \pi_\mathrm{n}^2 \mathcal{P}_{X|\mathrm{n}}\mathcal{P}_{X'|\mathrm{n}} - \pi_\mathrm{n}^2 \mathcal{P}_{X|\mathrm{n}}\mathcal{P}_{X'|\mathrm{p}}.
\end{aligned}
$$

Also, (57) is achieved by having

$$
\begin{aligned}
\left( \frac{\pi_\mathrm{p} - r}{\pi_\mathrm{n}} \right) \mathcal{P}_X\mathcal{P}_{X'} &= \left( \frac{\pi_\mathrm{p}}{\pi_\mathrm{n}}(1-r) - r \right) \mathcal{P}_X\mathcal{P}_{X'} \\
&= \frac{\pi_\mathrm{p}}{\pi_\mathrm{n}}\pi_\mathrm{p}\pi_\mathrm{n} \left( \mathcal{P}_{X|\mathrm{p}}\mathcal{P}_{X'|\mathrm{n}} + \mathcal{P}_{X|\mathrm{n}}\mathcal{P}_{X'|\mathrm{p}} \right) - \pi_\mathrm{p}^2 \mathcal{P}_{X|\mathrm{p}}\mathcal{P}_{X'|\mathrm{p}} - \pi_\mathrm{n}^2 \mathcal{P}_{X|\mathrm{n}}\mathcal{P}_{X'|\mathrm{n}} \\
&= \pi_\mathrm{p}^2 \mathcal{P}_{X|\mathrm{p}}\mathcal{P}_{X'|\mathrm{n}} - \pi_\mathrm{p}^2 \mathcal{P}_{X|\mathrm{p}}\mathcal{P}_{X'|\mathrm{p}} + \pi_\mathrm{p}^2 \mathcal{P}_{X|\mathrm{n}}\mathcal{P}_{X'|\mathrm{p}} - \pi_\mathrm{n}^2 \mathcal{P}_{X|\mathrm{n}}\mathcal{P}_{X'|\mathrm{n}}.
\end{aligned}
$$

$\square$

The equality (58) implies that the inner product of the first row (resp. the second row) of $M_{\mathrm{Sconf}}$ and $B$ represents a way (resp. another way) of obtaining $\mathcal{P}_X\mathcal{P}_{X'}$. Although one might suspect that it is redundant to formulate $\mathcal{P}_X\mathcal{P}_{X'}$ twice, we show in Section 5.1.6 this expression is crucial to rewrite the classification risk via the proposed framework. Furthermore, comparing $\bar{P} = M_{\mathrm{Sconf}}B$ (58) with $\bar{P} = M_{\mathrm{corr}}B$ (35), we have the reduction path

$$M_{\mathrm{corr}} \rightarrow M_{\mathrm{Sconf}}.$$

Note that $M_{\mathrm{Sconf}}$ does not fit the intuition of mutual contamination perfectly; we list Sconf learning in this subsection as all settings share the same base distributions $B$ (46).

## 4.2 CCN Scenarios

The formulation component (35) also applies to the CCN model. Unlike MCD contaminating class-conditionals (distributions of $X$), CCN corrupts class probability functions (labeling distributions). Next, we show how to formulate CCN via (35) and extend the formulation to characterize diverse weakly supervised settings.

In binary classification, CCN (Natarajan et al., 2013; 2017) corrupts the labels by flipping the positive (resp. negative) labels with probability $\mathcal{P}_{\bar{Y}=\mathrm{n}|Y=\mathrm{p},X}$ (resp. $\mathcal{P}_{\bar{Y}=\mathrm{p}|Y=\mathrm{n},X}$). Specifically,

$$
\begin{aligned}
\mathcal{P}_{\bar{Y}=\mathrm{p}|X} &:= \mathcal{P}_{\bar{Y}=\mathrm{p}|Y=\mathrm{p},X}\ \mathcal{P}_{Y=\mathrm{p}|X} + \mathcal{P}_{\bar{Y}=\mathrm{p}|Y=\mathrm{n},X}\ \mathcal{P}_{Y=\mathrm{n}|X}, \\
\mathcal{P}_{\bar{Y}=\mathrm{n}|X} &:= \mathcal{P}_{\bar{Y}=\mathrm{n}|Y=\mathrm{p},X}\ \mathcal{P}_{Y=\mathrm{p}|X} + \mathcal{P}_{\bar{Y}=\mathrm{n}|Y=\mathrm{n},X}\ \mathcal{P}_{Y=\mathrm{n}|X}
\end{aligned}
\tag{59}
$$

define the contaminated class probability functions. Taking the contamination targets $\mathcal{P}_{Y=\mathrm{p}|X}$ and $\mathcal{P}_{Y=\mathrm{n}|X}$ as the base distributions

$$B := \begin{pmatrix} \mathcal{P}_{Y=\mathrm{p}|X} \\ \mathcal{P}_{Y=\mathrm{n}|X} \end{pmatrix}$$

and denoting the label-generating distributions $\mathcal{P}_{\bar{Y}=\mathrm{p}|X}$ and $\mathcal{P}_{\bar{Y}=\mathrm{n}|X}$ as

$$\bar{P} := \begin{pmatrix} \mathcal{P}_{\bar{Y}=\mathrm{p}|X} \\ \mathcal{P}_{\bar{Y}=\mathrm{n}|X} \end{pmatrix},$$

we express (59) in the matrix form as follows

$$\begin{pmatrix} \mathcal{P}_{\bar{Y}=\mathrm{p}|X} \\ \mathcal{P}_{\bar{Y}=\mathrm{n}|X} \end{pmatrix} = \begin{pmatrix} \mathcal{P}_{\bar{Y}=\mathrm{p}|Y=\mathrm{p},X} & \mathcal{P}_{\bar{Y}=\mathrm{p}|Y=\mathrm{n},X} \\ \mathcal{P}_{\bar{Y}=\mathrm{n}|Y=\mathrm{p},X} & \mathcal{P}_{\bar{Y}=\mathrm{n}|Y=\mathrm{n},X} \end{pmatrix} \begin{pmatrix} \mathcal{P}_{Y=\mathrm{p}|X} \\ \mathcal{P}_{Y=\mathrm{n}|X} \end{pmatrix}.$$

Comparing with the abstract form $\bar{P} = M_{\mathrm{corr}}B$ (35), we see that

$$M_{\mathrm{CCN}} := \begin{pmatrix} \mathcal{P}_{\bar{Y}=\mathrm{p}|Y=\mathrm{p},X} & \mathcal{P}_{\bar{Y}=\mathrm{p}|Y=\mathrm{n},X} \\ \mathcal{P}_{\bar{Y}=\mathrm{n}|Y=\mathrm{p},X} & \mathcal{P}_{\bar{Y}=\mathrm{n}|Y=\mathrm{n},X} \end{pmatrix} \tag{60}$$

instantiates the contamination matrix $M_{\mathrm{corr}}$ in the CCN setting.

### 4.2.1 Generalized CCN

The concept of contaminating a *single* label can be extended to generating a *compound* label in the multiclass classification setting. Let $2^{\mathcal{Y}}$ be the power set of the label space $\mathcal{Y} = [K]$. Define $\mathcal{S} := 2^{\mathcal{Y}} \setminus \{\emptyset, \mathcal{Y}\}$ as the observable space of compound labels [3]. Since a compound label $S \in \mathcal{S}$ consists of an arbitrary number of class indices, one can view the probability of observing $S$ for a given $X$ is governed by several class probabilities $\mathcal{P}_{Y=k|X}$. Therefore, generalizing the CCN formulation (59), we define the label-generating process of a compound label $S$ as

$$\mathcal{P}_{S|X} = \sum_{k=1}^{K} \mathcal{P}_{S|Y=k,X} \mathcal{P}_{Y=k|X},$$

where the role of $\mathcal{P}_{S|Y,X}$ is the probability of converting a single label $Y$ to a compound label $S \in \mathcal{S}$. Moreover, in CCN, the distribution $\mathcal{P}_X$ is not contaminated. Thus, by multiplying $\mathcal{P}_X$ on both sides, we obtain the data-generating distribution

$$\mathcal{P}_{S,X} = \sum_{k=1}^{K} \mathcal{P}_{S|Y=k,X} \mathcal{P}_{Y=k,X}, \tag{61}$$

Viewing $\mathcal{P}_{S|Y,X}$ as a contamination probability, we arrange $\mathcal{P}_{S=s|Y=k,X}$ into a matrix in the following lemma to formulate the contamination matrix for our generalized CCN (gCCN) setting.

**Lemma 10.** *Let the elements in $\mathcal{S}$ be $\{s_1, s_2, \ldots s_{|\mathcal{S}|}\}$. For the gCCN setting, denote the data-generating distributions as*

$$\bar{P} := \begin{pmatrix} \mathcal{P}_{S=s_1,X} \\ \vdots \\ \mathcal{P}_{S=s_{|\mathcal{S}|},X} \end{pmatrix} \tag{62}$$

*and the base distributions as*

$$B := \begin{pmatrix} \mathcal{P}_{Y=1,X} \\ \vdots \\ \mathcal{P}_{Y=K,X} \end{pmatrix} = P. \tag{63}$$

---

[3]Removing $\emptyset$ and $\mathcal{Y}$ is that the empty set does not contain any label information and $\mathcal{Y}$ is a trivial case.

*Define*

$$
M_{\mathrm{gCCN}} := \begin{pmatrix}
\mathcal{P}_{S=s_1|Y=1,X} & \mathcal{P}_{S=s_1|Y=2,X} & \cdots & \mathcal{P}_{S=s_1|Y=K,X} \\
\mathcal{P}_{S=s_2|Y=1,X} & \mathcal{P}_{S=s_2|Y=2,X} & \cdots & \mathcal{P}_{S=s_2|Y=K,X} \\
\vdots & \vdots & \ddots & \vdots \\
\mathcal{P}_{S=s_{|\mathcal{S}|}|Y=1,X} & \mathcal{P}_{S=s_{|\mathcal{S}|}|Y=2,X} & \cdots & \mathcal{P}_{S=s_{|\mathcal{S}|}|Y=K,X}
\end{pmatrix}.
\tag{64}
$$

*Then,* $\bar{P} = M_{\mathrm{gCCN}}B$.

The lemma implies that $M_{\mathrm{gCCN}}$ is the contamination matrix characterizing $\bar{P}$ of the gCCN setting. Also, note that $\bar{P} = M_{\mathrm{gCCN}}B$ is essentially the matrix form of (61). Moreover, $M_{\mathrm{gCCN}}$ generalizes $M_{\mathrm{CCN}}$ (60) by extending the label spaces: Both the clean label $Y$ and the contaminated label $\bar{Y}$ belong to $\{\mathrm{p}, \mathrm{n}\}$ in CCN, while in the gCCN setting, the clean label $Y \in \{1, \cdots, K\}$ and the compound label $S \in \{s_1, \cdots, s_{|\mathcal{S}|}\}$.

*Proof.* For each $j \in [|\mathcal{S}|]$, we have

$$
\left(M_{\mathrm{gCCN}}B\right)_j = \sum_{k=1}^{K} \mathcal{P}_{S=s_j|Y=k,X} \mathcal{P}_{Y=k,X} = \sum_{k=1}^{K} \mathcal{P}_{S=s_j,Y=k,X} = \mathcal{P}_{S=s_j,X} = \bar{P}_j.
$$

$\square$

Comparing $\bar{P} = M_{\mathrm{gCCN}}B$ with the formulation framework $\bar{P} = M_{\mathrm{corr}}B$ (35), we have the reduction path

$$
M_{\mathrm{corr}} \to M_{\mathrm{CCN}} \rightsquigarrow M_{\mathrm{gCCN}}.
$$

Similar to $M_{\mathrm{UU}}$ (49), which induces multiple contamination matrices as special cases of the MCD model, $M_{\mathrm{gCCN}}$ also derives several contamination matrices formulating partial- or complementary-label settings, as we will show in the rest of this subsection.

### 4.2.2 Proper Partial-Label (PPL) Learning (Wu et al., 2023)

For a given example $(y, x)$ and a compound label $s \in \mathcal{S}$, we call $s$ a partial-label of $x$ if $y \in s$. Statistically speaking, we assume $\mathcal{P}_{Y \in S|S,X} = 1$. Formally, according to Definition 1 of Wu et al. (2023), if the contamination probability can be defined as

$$
\mathcal{P}_{S|Y,X} := C(S,X)\mathbb{I}\left[Y \in S\right],
\tag{65}
$$

via a function $C : \mathcal{S} \times \mathcal{X} \to \mathbb{R}$, we call such a partial-label scenario proper.

Since the discussion above only involves specifying $\mathcal{P}_{S|Y,X}$, we replace the entries of $M_{\mathrm{gCCN}}$ (64) according to (65) to construct $M_{\mathrm{PPL}}$:

$$
\begin{pmatrix}
C(s_1, X)\mathbb{I}\left[Y = 1 \in s_1\right] & C(s_1, X)\mathbb{I}\left[Y = 2 \in s_1\right] & \cdots & C(s_1, X)\mathbb{I}\left[Y = K \in s_1\right] \\
C(s_2, X)\mathbb{I}\left[Y = 1 \in s_2\right] & C(s_2, X)\mathbb{I}\left[Y = 2 \in s_2\right] & \cdots & C(s_2, X)\mathbb{I}\left[Y = K \in s_2\right] \\
\vdots & \vdots & \ddots & \vdots \\
C(s_{|\mathcal{S}|}, X)\mathbb{I}\left[Y = 1 \in s_{|\mathcal{S}|}\right] & C(s_{|\mathcal{S}|}, X)\mathbb{I}\left[Y = 2 \in s_{|\mathcal{S}|}\right] & \cdots & C(s_{|\mathcal{S}|}, X)\mathbb{I}\left[Y = K \in s_{|\mathcal{S}|}\right]
\end{pmatrix}.
\tag{66}
$$

The following lemma justifies $M_{\mathrm{PPL}}$ as the contamination matrix for PPL learning.

**Lemma 11.** *Let the elements in $\mathcal{S}$ be $\{s_1, s_2, \ldots s_{|\mathcal{S}|}\}$. For each $j \in [|\mathcal{S}|]$, let the $j$-th entry of $\bar{P}$ be*

$$
\bar{P}_j = \mathcal{P}_{S=s_j,X} := C(S = s_j, X) \sum_{k \in s_j} \mathcal{P}_{Y=k,X},
$$

which denotes the data-generating distribution of $(s_j, X)$. Assume the base distributions $B$ and the contamination matrix $M_{\mathrm{PPL}}$ are given by (63) and (66), respectively. Then, $M_{\mathrm{PPL}}$ satisfies $\bar{P} = M_{\mathrm{PPL}}B$ and characterizes PPL learning (26).

The entry replacement that converts (64) to (66) through (65) gives the reduction path

$$M_{\mathrm{corr}} \to M_{\mathrm{gCCN}} \to M_{\mathrm{PPL}}.$$

### 4.2.3 Provably Consistent Partial-Label (PCPL) Learning (Feng et al., 2020b)

In PCPL, the probability of each partial-label is assumed to be sampled uniformly from all feasible partial-labels. Since there are $2^{K-1}-1$ feasible partial-labels for every $y$, the label-converting probability $\mathcal{P}_{S=s|Y=y,X}$ is $\frac{1}{2^{K-1}-1}$ if $y \in s^4$. It corresponds to assign $C(S,X) = \frac{1}{2^{K-1}-1}$ in (65). Hence, we obtain

$$C(S,X)\mathbb{I}\left[Y \in S\right] := \frac{1}{2^{K-1}-1}\mathbb{I}\left[Y \in S\right], \tag{67}$$

which reduces the label-converting process of PPL to that of PCPL and recovers (5) of Feng et al. (2020b).

Then, replacing entries in (66) via (67), we obtain the contamination matrix of PCPL learning

$$M_{\mathrm{PCPL}} := \frac{1}{2^{K-1}-1}\begin{pmatrix} \mathbb{I}\left[Y=1 \in s_1\right] & \mathbb{I}\left[Y=2 \in s_1\right] & \cdots & \mathbb{I}\left[Y=K \in s_1\right] \\ \mathbb{I}\left[Y=1 \in s_2\right] & \mathbb{I}\left[Y=2 \in s_2\right] & \cdots & \mathbb{I}\left[Y=K \in s_2\right] \\ \vdots & \vdots & \ddots & \vdots \\ \mathbb{I}\left[Y=1 \in s_{|\mathcal{S}|}\right] & \mathbb{I}\left[Y=2 \in s_{|\mathcal{S}|}\right] & \cdots & \mathbb{I}\left[Y=K \in s_{|\mathcal{S}|}\right] \end{pmatrix} \tag{68}$$

and the reduction path

$$M_{\mathrm{corr}} \to M_{\mathrm{gCCN}} \to M_{\mathrm{PPL}} \to M_{\mathrm{PCPL}}.$$

$M_{\mathrm{PCPL}}$ characterizing the data-generating process of PCPL is justified by the following lemma, whose proof follows the same steps as that for Lemma 11.

**Lemma 12.** *Let the elements in $\mathcal{S}$ be $\left\{s_1, s_2, \dots s_{|\mathcal{S}|}\right\}$. For each $j \in [|\mathcal{S}|]$, let the $j$-th entry of $\bar{P}$ be*

$$\bar{P}_j = \mathcal{P}_{S=s_j,X} := \frac{1}{2^{K-1}-1}\sum_{k \in s_j}\mathcal{P}_{Y=k,X},$$

*which denotes the data-generating distribution of $(s_j, X)$. Assume the base distributions $B$ and the contamination matrix $M_{\mathrm{PCPL}}$ are given by (63) and (68), respectively. Then, $M_{\mathrm{PCPL}}$ satisfies $\bar{P} = M_{\mathrm{PCPL}}B$ and characterizes PCPL learning (24).*

### 4.2.4 Multi-Complementary-Label (MCL) Learning (Feng et al., 2020a)

Recall the discussions in Sections 2.2.9 and 2.2.10 that a complementary-label contains the exclusion information of a true label. That is, for a data example $(y, x)$, we call a set of class indices $\bar{s} \in \mathcal{S} = 2^{\mathcal{Y}} \setminus \{\emptyset, \mathcal{Y}\}$ an MCL of $x$ if $\bar{s}$ does not contain $y$.

Denote $\mathcal{S} := \{\bar{s}_1, \bar{s}_2, \dots \bar{s}_N\}$. The equivalence

$$\sum_{d=1}^{K-1}\mathcal{P}_{|\bar{S}|=d} \cdot \frac{1}{\binom{K-1}{|\bar{S}|}}\sum_{Y \notin \bar{S}}\mathcal{P}_{Y,X}\mathbb{I}\left[|\bar{S}|=d\right] = \begin{cases} \sum_{d=1}^{K-1}\mathcal{P}_{|\bar{S}|=d} \cdot \frac{1}{\binom{K-1}{|\bar{S}|}}\sum_{Y \notin \bar{S}}\mathcal{P}_{Y,X}, & \text{if } |\bar{S}| = d, \\ 0, & \text{otherwise} \end{cases}$$

---

[4]There are $2^{\mathcal{Y}\setminus\{y\}}\setminus\{\mathcal{Y}\setminus\{y\}\} = 2^{K-1}-1$ combinations whose union with $\{y\}$ are partial-labels of $y$.

allows us to define the data-generating distribution of MCL (22) as

$$\bar{P} := \begin{pmatrix} \mathcal{P}_{\bar{S}=\bar{s}_1, X} \\ \vdots \\ \mathcal{P}_{\bar{S}=\bar{s}_N, X} \end{pmatrix}, \tag{69}$$

where for each $j \in [N]$,

$$\mathcal{P}_{\bar{S}=\bar{s}_j, X} := \sum_{d=1}^{K-1} \frac{\mathcal{P}_{|\bar{s}_j|=d}}{\binom{K-1}{|\bar{s}_j|}} \sum_{Y \notin \bar{s}_j} \mathcal{P}_{Y,X} \mathbb{I}\left[|\bar{s}_j| = d\right]. \tag{70}$$

Let $(y, x)$ be fixed. The data-generating process of MCL proposed by Feng et al. (2020a) is that one first samples a size $d$ with probability $\mathcal{P}_{|\bar{S}|=d}$, and then samples a $\bar{s}$ uniformly at random from $\{\bar{s}_{d,1}, \bar{s}_{d,2}, \ldots, \bar{s}_{d,N_d}\} \subset \mathcal{S}$, where $\bar{s}_{d,\cdot}$ means a set of size $d$ excluding $y$ and $N_d$ is the total number of those sets. Note that $N_d = \binom{K-1}{d}$ since we remove $y$ from $\mathcal{Y}$ and then choose a set of size $d$ to form a $\bar{s}_{d,\cdot}$. Furthermore, we need a more complicated lower index system to distinguish $\{\bar{s}_{d,1}, \bar{s}_{d,2}, \ldots, \bar{s}_{d,N_d}\}$ from $\mathcal{S} = \{\bar{s}_1, \bar{s}_2, \ldots \bar{s}_N\}$ since $d$ ranges from 1 to $K-1$ and $\sum_{d=1}^{K-1} N_d = \sum_{d=1}^{K-1} \binom{K-1}{d} = 2^K - 2 = |\mathcal{S}|$. According to this mechanism, we construct $M_{\mathrm{MCL}}$:

$$\begin{pmatrix} \frac{\mathcal{P}_{|\bar{S}|=|\bar{s}_1|}}{\binom{K-1}{|\bar{s}_1|}} \mathbb{I}\left[Y = 1 \notin \bar{s}_1\right] & \frac{\mathcal{P}_{|\bar{S}|=|\bar{s}_1|}}{\binom{K-1}{|\bar{s}_1|}} \mathbb{I}\left[Y = 2 \notin \bar{s}_1\right] & \cdots & \frac{\mathcal{P}_{|\bar{S}|=|\bar{s}_1|}}{\binom{K-1}{|\bar{s}_1|}} \mathbb{I}\left[Y = K \notin \bar{s}_1\right] \\ \frac{\mathcal{P}_{|\bar{S}|=|\bar{s}_2|}}{\binom{K-1}{|\bar{s}_2|}} \mathbb{I}\left[Y = 1 \notin \bar{s}_2\right] & \frac{\mathcal{P}_{|\bar{S}|=|\bar{s}_2|}}{\binom{K-1}{|\bar{s}_2|}} \mathbb{I}\left[Y = 2 \notin \bar{s}_2\right] & \cdots & \frac{\mathcal{P}_{|\bar{S}|=|\bar{s}_2|}}{\binom{K-1}{|\bar{s}_2|}} \mathbb{I}\left[Y = K \notin \bar{s}_2\right] \\ \vdots & \vdots & \ddots & \vdots \\ \frac{\mathcal{P}_{|\bar{S}|=|\bar{s}_N|}}{\binom{K-1}{|\bar{s}_N|}} \mathbb{I}\left[Y = 1 \notin \bar{s}_N\right] & \frac{\mathcal{P}_{|\bar{S}|=|\bar{s}_N|}}{\binom{K-1}{|\bar{s}_N|}} \mathbb{I}\left[Y = 2 \notin \bar{s}_N\right] & \cdots & \frac{\mathcal{P}_{|\bar{S}|=|\bar{s}_N|}}{\binom{K-1}{|\bar{s}_N|}} \mathbb{I}\left[Y = K \notin \bar{s}_N\right] \end{pmatrix}. \tag{71}$$

The following lemma justifies $M_{\mathrm{MCL}}$ as the contamination matrix for MCL learning.

**Lemma 13.** *Suppose the base distributions $B$, the contamination matrix $M_{\mathrm{MCL}}$, and the data-generating distributions $\bar{P}$ are given by (63), (71), and (70), respectively. Then, $M_{\mathrm{MCL}}$ satisfies $\bar{P} = M_{\mathrm{MCL}} B$ and characterizes MCL (22).*

At the first sight, $M_{\mathrm{MCL}}$ (71) does not resemble $M_{\mathrm{PCPL}}$ (68) or $M_{\mathrm{PPL}}$ (66). The subtle connection can be established via a relation between partial-label and complementary-label. Recall

$$2^{\mathcal{Y}} \setminus \{\emptyset, \mathcal{Y}\} = \mathcal{S} = \{\bar{s}_1, \bar{s}_2, \ldots \bar{s}_N\},$$

where $\bar{s}_j$ is a MCL. From the partial-label perspective, we can establish the following set equality relationship:

$$2^{\mathcal{Y}} \setminus \{\emptyset, \mathcal{Y}\} = \mathcal{S} = \{\bar{s}_1, \bar{s}_2, \ldots \bar{s}_N\} = \{s_1 := \mathcal{Y} \setminus \bar{s}_1, s_2 := \mathcal{Y} \setminus \bar{s}_2, \ldots, s_N := \mathcal{Y} \setminus \bar{s}_N\}. \tag{72}$$

This is because for every $\bar{s} \in \mathcal{S}$, there is a $s \in \mathcal{S}$ such that $s := \mathcal{Y} \setminus \bar{s}$. The intuition behind (72) is that if $\bar{s}$ is an MCL of $x$, then $s := \mathcal{Y} \setminus \bar{s}$ must be a partial-label of $x$. Therefore, we can also use the set of partial-labels $\{s_1, s_2, \ldots, s_N\}$ to denote $\mathcal{S}$. The following lemma exploits this relation to show that $M_{\mathrm{MCL}}$ is indeed a special case of $M_{\mathrm{PPL}}$.

**Lemma 14.** *Assign each $(s, k)$ entry of $M_{\mathrm{PPL}}$ (66) with*

$$C(s, X) \mathbb{I}\left[Y = k \in s\right] := \frac{\mathcal{P}_{|S|=|s|}}{\binom{K-1}{|s|-1}} \mathbb{I}\left[Y = k \in s\right]. \tag{73}$$

*Then, the resulting matrix*

$$M'_{\mathrm{MCL}} = \begin{pmatrix} \frac{\mathcal{P}_{|S|=|s_1|}}{\binom{K-1}{|s_1|-1}}\mathbb{I}\left[Y=1\in s_1\right] & \frac{\mathcal{P}_{|S|=|s_1|}}{\binom{K-1}{|s_1|-1}}\mathbb{I}\left[Y=2\in s_1\right] & \cdots & \frac{\mathcal{P}_{|S|=|s_1|}}{\binom{K-1}{|s_1|-1}}\mathbb{I}\left[Y=K\in s_1\right] \\ \frac{\mathcal{P}_{|S|=|s_2|}}{\binom{K-1}{|s_2|-1}}\mathbb{I}\left[Y=1\in s_2\right] & \frac{\mathcal{P}_{|S|=|s_2|}}{\binom{K-1}{|s_2|-1}}\mathbb{I}\left[Y=2\in s_2\right] & \cdots & \frac{\mathcal{P}_{|S|=|s_2|}}{\binom{K-1}{|s_2|-1}}\mathbb{I}\left[Y=K\in s_2\right] \\ \vdots & \vdots & \ddots & \vdots \\ \frac{\mathcal{P}_{|S|=|s_N|}}{\binom{K-1}{|s_N|-1}}\mathbb{I}\left[Y=1\in s_N\right] & \frac{\mathcal{P}_{|S|=|s_N|}}{\binom{K-1}{|s_N|-1}}\mathbb{I}\left[Y=2\in s_N\right] & \cdots & \frac{\mathcal{P}_{|S|=|s_N|}}{\binom{K-1}{|s_N|-1}}\mathbb{I}\left[Y=K\in s_N\right] \end{pmatrix} \tag{74}$$

*is equivalent to $M_{\mathrm{MCL}}$ (71) under the relationship (72).*

*Proof.* Note that for every $j \in [N]$, $s_j = \mathcal{Y}\backslash\bar{s}_j$. This implies $\mathcal{P}_{|\bar{S}|=|\bar{s}_j|} = \mathcal{P}_{|S|=|s_j|}$, $\binom{K-1}{|s_j|-1} = \binom{K-1}{|\bar{s}_j|}$, and $\mathbb{I}\left[Y \in s_j\right] = \mathbb{I}\left[Y \notin \bar{s}_j\right]$ hold for every $j \in [N]$. Therefore, for each $(j, k)$ entry in $M'_{\mathrm{MCL}}$ and $M_{\mathrm{MCL}}$ (71), we have

$$\frac{\mathcal{P}_{|S|=|s_j|}}{\binom{K-1}{|s_j|-1}}\mathbb{I}\left[Y=k\in s_j\right] = \frac{\mathcal{P}_{|\bar{S}|=|\bar{s}_j|}}{\binom{K-1}{|\bar{s}_j|}}\mathbb{I}\left[Y=k\notin \bar{s}_j\right]. \tag{75}$$

$\square$

The assignment rule (73) implies the reduction path

$$M_{\mathrm{corr}} \to M_{\mathrm{gCCN}} \to M_{\mathrm{PPL}} \to M_{\mathrm{MCL}}.$$

Comparing (73) with (67) of PCPL, we see that MCL and PCPL can be viewed as different ways of composing $\mathcal{P}_{Y,X}$ to generate a partial-label, with weights $\frac{\mathcal{P}_{|S|=|s|}}{\binom{K-1}{|s|-1}}$ and $\frac{1}{2^{K-1}-1}$, respectively.

### 4.2.5 Complementary-Label (CL) Learning (Ishida et al., 2019)

As a special case of MCL (Section 2.2.10), we can construct the contamination matrix $M_{\mathrm{CL}}$ from $M_{\mathrm{MCL}}$. The set of all CLs is composed of MCL with size 1: $\{1\},\ldots,\{K\}$. Therefore, we assign values in $M_{\mathrm{MCL}}$ (71) as follows. For each $\bar{s} \in \mathcal{S}$, $\mathcal{P}_{|\bar{S}|=|\bar{s}|} = 1$ if $|\bar{s}| = 1$ and $\mathcal{P}_{|\bar{S}|=|\bar{s}|} = 0$ if $|\bar{s}| > 1$. Dropping all-zero rows, we obtain from (71) the contamination matrix of CL learning

$$\begin{aligned} M_{\mathrm{CL}} &:= \begin{pmatrix} \frac{1}{K-1}\mathbb{I}\left[Y=1\notin \{1\}\right] & \frac{1}{K-1}\mathbb{I}\left[Y=2\notin \{1\}\right] & \cdots & \frac{1}{K-1}\mathbb{I}\left[Y=K\notin \{1\}\right] \\ \frac{1}{K-1}\mathbb{I}\left[Y=1\notin \{2\}\right] & \frac{1}{K-1}\mathbb{I}\left[Y=2\notin \{2\}\right] & \cdots & \frac{1}{K-1}\mathbb{I}\left[Y=K\notin \{2\}\right] \\ \vdots & \vdots & \ddots & \vdots \\ \frac{1}{K-1}\mathbb{I}\left[Y=1\notin \{K\}\right] & \frac{1}{K-1}\mathbb{I}\left[Y=2\notin \{K\}\right] & \cdots & \frac{1}{K-1}\mathbb{I}\left[Y=K\notin \{K\}\right] \end{pmatrix} \\ &= \frac{1}{K-1}\begin{pmatrix} 0 & 1 & \cdots & 1 \\ 1 & 0 & \cdots & 1 \\ \vdots & \vdots & \ddots & \vdots \\ 1 & 1 & \cdots & 0 \end{pmatrix} \end{aligned} \tag{76}$$

and the reduction path

$$M_{\mathrm{corr}} \to M_{\mathrm{gCCN}} \to M_{\mathrm{PPL}} \to M_{\mathrm{MCL}} \to M_{\mathrm{CL}}.$$

Furthermore, it is easy to verify that given $B$ (63), for any $j \in [K]$,

$$\left(M_{\mathrm{CL}}B\right)_j = \sum_{Y\neq j}\frac{1}{K-1}\mathcal{P}_{Y,X} = \mathcal{P}_{\bar{S}=j,X},$$

which corresponds to formulation (20). Hence, we have the following.

**Lemma 15.** $M_{\mathrm{CL}}$ *(76) is the contamination matrix characterizing the data-generating distribution* $\mathcal{P}_{\bar{S},X}$ *(20) of CL learning.*

### 4.3 Confidence-based Scenarios

At first sight, there seems to be no connection between "contamination" and single-class classification (Cao et al., 2021a). However, the following derivation

$$\frac{\mathcal{P}_{Y=y_{\mathrm{s}}|X}}{\mathcal{P}_{Y=j|X}} \cdot \mathcal{P}_{Y=j,X} = \frac{\mathcal{P}_{Y=y_{\mathrm{s}}|X}}{\mathcal{P}_{Y=j|X}} \cdot \mathcal{P}_{Y=j|X}\,\mathcal{P}_X = \mathcal{P}_{Y=y_{\mathrm{s}},X} \tag{77}$$

reveals a way to *contaminate* a clean joint probability $\mathcal{P}_{Y=j,X}$ to the joint probability $\mathcal{P}_{Y=y_{\mathrm{s}},X}$ of a designated class $y_{\mathrm{s}}$ via confidence weighting $\frac{\mathcal{P}_{Y=y_{\mathrm{s}}|X}}{\mathcal{P}_{Y=j|X}}$. As we will see in the rest of this subsection, the confidence weights are the key elements in formulating the contamination matrices for the confidence-based WSL settings.

#### 4.3.1 Subset Confidence (Sub-Conf) Learning (Cao et al., 2021a)

Let $\mathcal{Y}_{\mathrm{s}} \subset [K]$ be a subset of classes. Viewing $\mathcal{Y}_{\mathrm{s}}$ as a "superclass", such that every instance $x$ of $(y,x)$ will be labeled $\mathcal{Y}_{\mathrm{s}}$ if $y \in \mathcal{Y}_{\mathrm{s}}$, we can define its class prior as $\mathcal{P}_{Y\in\mathcal{Y}_{\mathrm{s}}} = \pi_{\mathcal{Y}_{\mathrm{s}}} := \sum_{y\in\mathcal{Y}_{\mathrm{s}}} \pi_y$ and its class probability function as $\mathcal{P}_{Y\in\mathcal{Y}_{\mathrm{s}}|X} := \sum_{y\in\mathcal{Y}_{\mathrm{s}}} \mathcal{P}_{Y=y|X}$. Substituting the designated class $y_{\mathrm{s}}$ in (77) with the superclass $\mathcal{Y}_{\mathrm{s}}$,

$$\frac{\mathcal{P}_{Y\in\mathcal{Y}_{\mathrm{s}}|X}}{\mathcal{P}_{Y=j|X}} \cdot \mathcal{P}_{Y=j,X} = \frac{\mathcal{P}_{Y\in\mathcal{Y}_{\mathrm{s}}|X}}{\mathcal{P}_{Y=j|X}} \cdot \mathcal{P}_{Y=j|X}\,\mathcal{P}_X = \mathcal{P}_{Y\in\mathcal{Y}_{\mathrm{s}},X} \tag{78}$$

shows that no matter what joint distribution $\mathcal{P}_{Y=j,X}$ to begin with, the confidence weight $\frac{\mathcal{P}_{Y\in\mathcal{Y}_{\mathrm{s}}|X}}{\mathcal{P}_{Y=j|X}}$ twists that joint distribution so that every observed data appears to be sampled from the same superclass distribution $\mathcal{P}_{\mathcal{Y}_{\mathrm{s}},X}$. The following lemma leverages the observation to specify the contamination matrix $M_{\mathrm{Sub}}$ characterizing Sub-Conf learning.

**Lemma 16.** *Denote the base distributions as*

$$B := \begin{pmatrix} \mathcal{P}_{Y=1,X} \\ \vdots \\ \mathcal{P}_{Y=K,X} \end{pmatrix} = P \tag{79}$$

*and the data-generating distributions as*

$$\bar{P} := \begin{pmatrix} \mathcal{P}_{Y\in\mathcal{Y}_{\mathrm{s}},X} \\ \vdots \\ \mathcal{P}_{Y\in\mathcal{Y}_{\mathrm{s}},X} \end{pmatrix}.$$

*Inserting the confidence weights into the identity matrix, we define the contamination matrix*

$$M_{\mathrm{Sub}} := \begin{pmatrix} \frac{\mathcal{P}_{Y\in\mathcal{Y}_{\mathrm{s}}|X}}{\mathcal{P}_{Y=1|X}} & \cdots & 0 \\ \vdots & \ddots & \vdots \\ 0 & \cdots & \frac{\mathcal{P}_{Y\in\mathcal{Y}_{\mathrm{s}}|X}}{\mathcal{P}_{Y=K|X}} \end{pmatrix}. \tag{80}$$

*Then, $\bar{P} = M_{\mathrm{Sub}}B$, and $M_{\mathrm{Sub}}$ characterizes the data-generating process of Sub-Conf learning (30).*

*Proof.* For each $j \in [K]$, $\left(M_{\mathrm{Sub}}B\right)_j = \mathcal{P}_{Y\in\mathcal{Y}_{\mathrm{s}},X}$ follows from (78). Thus, $\bar{P} = M_{\mathrm{Sub}}B$. It further implies all observed instances are labeled with the same superclass $\mathcal{Y}_{\mathrm{s}}$, meaning we can drop the observed labels, and the observed examples $\{x_i\}_{i=1}^n$ is equivalent to a set of i.i.d. samples from $\mathcal{P}_{X|Y\in\mathcal{Y}_{\mathrm{s}}}$ (30). $\qquad\square$

Comparing $\bar{P} = M_{\mathrm{Sub}}B$ with the formulation framework $\bar{P} = M_{\mathrm{corr}}B$ (35), we observe that in Sub-Conf learning, $M_{\mathrm{corr}}$ is realized as $M_{\mathrm{Sub}}$:

$$M_{\mathrm{corr}} \to M_{\mathrm{Sub}}.$$

### 4.3.2 Single-Class Confidence (SC-Conf) Learning (Cao et al., 2021a)

We compare the formulation of SC-Conf (28) with Sub-Conf (30) and observe that SC-Conf is a special case of Sub-Conf when $\mathcal{Y}_{\mathrm{s}} = \{y_{\mathrm{s}}\}$ being a singleton. Thus, we straightforwardly obtain the matrix formulation of SC-Conf from Lemma 16 be replacing $\mathcal{Y}_{\mathrm{s}}$ in (80) with $y_{\mathrm{s}}$:

**Lemma 17.** *Let the base distributions $B$ be defined by (79) and the data-generating distributions be defined by*

$$\bar{P} := \begin{pmatrix} \mathcal{P}_{Y=y_{\mathrm{s}},X} \\ \vdots \\ \mathcal{P}_{Y=y_{\mathrm{s}},X} \end{pmatrix}.$$

*Define the contamination matrix*

$$M_{\mathrm{SC}} := \begin{pmatrix} \frac{\mathcal{P}_{Y=y_{\mathrm{s}}|X}}{\mathcal{P}_{Y=1|X}} & \cdots & 0 \\ \vdots & \ddots & \vdots \\ 0 & \cdots & \frac{\mathcal{P}_{Y=y_{\mathrm{s}}|X}}{\mathcal{P}_{Y=K|X}} \end{pmatrix} \tag{81}$$

*by substituting $\mathcal{Y}_{\mathrm{s}}$ in (80) with $y_{\mathrm{s}}$. Then, $\bar{P} = M_{\mathrm{SC}}B$ and $M_{\mathrm{SC}}$ characterizes the data-generating process of SC-Conf learning (28).*

Since SC-Conf is a special case of Sub-Conf, we have the reduction path

$$M_{\mathrm{corr}} \to M_{\mathrm{Sub}} \to M_{\mathrm{SC}}.$$

### 4.3.3 Positive-confidence (Pconf) Learning (Ishida et al., 2018)

Comparing (6) with (28), we see that Pconf is a special case of SC-Conf when $K = 2$ and $y_{\mathrm{s}} = \mathrm{p}$ since $r_{\mathrm{n}}(X) = 1 - r_{\mathrm{p}}(X)$. A further modification to (81) we obtain the contamination matrix $M_{\mathrm{Pconf}}$ characterizing Pconf learning.

**Lemma 18.** *Let $B := \begin{pmatrix} \mathcal{P}_{Y=\mathrm{p},X} \\ \mathcal{P}_{Y=\mathrm{n},X} \end{pmatrix} = P$ and $\bar{P} := \begin{pmatrix} \mathcal{P}_{Y=\mathrm{p},X} \\ \mathcal{P}_{Y=\mathrm{p},X} \end{pmatrix}$. Define*

$$M_{\mathrm{Pconf}} := \begin{pmatrix} \frac{\mathcal{P}_{Y=\mathrm{p}|X}}{\mathcal{P}_{Y=\mathrm{p}|X}} & 0 \\ 0 & \frac{\mathcal{P}_{Y=\mathrm{p}|X}}{\mathcal{P}_{Y=\mathrm{n}|X}} \end{pmatrix}. \tag{82}$$

*Then, $\bar{P} = M_{\mathrm{Pconf}}B$, and $M_{\mathrm{Pconf}}$ characterizes the data-generating process of Pconf learning (6).*

The entry replacement that converts (81) to (82) implies the reduction path

$$M_{\mathrm{corr}} \to M_{\mathrm{Sub}} \to M_{\mathrm{SC}} \to M_{\mathrm{Pconf}}.$$

### 4.3.4 Soft-Label Learning (Ishida et al., 2023)

The difference between the soft-label and the previous confidence-based settings (Sub-Conf, SC-Conf, and Pconf) is how $x$ is sampled. The sample distributions condition on the label information in the previous settings, while that in soft-label is $\mathcal{P}_X$. Replacing the confidence weight $\frac{\mathcal{P}_{Y=y_s|X}}{\mathcal{P}_{Y=j|X}}$ in (77) with $\frac{1}{\mathcal{P}_{Y=j|X}}$,

$$\frac{1}{\mathcal{P}_{Y=j|X}} \cdot \mathcal{P}_{Y=j,X} = \mathcal{P}_X$$

explains how to convert $\mathcal{P}_{Y=j,X}$ to $\mathcal{P}_X$. Therefore, filling the $j$-th diagonal entry of the identity matrix with $\frac{1}{\mathcal{P}_{Y=j|X}}$, we obtain the contamination matrix $M_{\text{Soft}}$ for soft-label learning:

**Lemma 19.** *Let the base distributions $B$ be defined by (79). Denote the data-generating distribution as*

$$\bar{P} := \begin{pmatrix} \mathcal{P}_X \\ \vdots \\ \mathcal{P}_X \end{pmatrix}. \tag{83}$$

*Define the contamination matrix*

$$M_{\text{Soft}} := \begin{pmatrix} \frac{1}{\mathcal{P}_{Y=1|X}} & \cdots & 0 \\ \vdots & \ddots & \vdots \\ 0 & \cdots & \frac{1}{\mathcal{P}_{Y=K|X}} \end{pmatrix}. \tag{84}$$

*Then, $\bar{P} = M_{\text{Soft}}B$, and $M_{\text{Soft}}$ characterizes the data-generating process in (33).*

Unlike SC-Conf and Pconf, which are special cases of Sub-Conf with $\mathcal{Y}_s$ taking only one label, the generation process of a soft-label can be viewed as assigning $\mathcal{Y}_s := [K]$. Considering the entire label space results in $\mathcal{P}_{Y\in[K]|X} = 1$; it coincides with the meaning of $\mathcal{P}_X$ that samples $x$ regardless of the labels. Although technically the soft-label setting is not a special case of Sub-Conf (recalling the $\mathcal{Y}_s \subset [K]$ assumption from Section 4.3.1), $M_{\text{Soft}}$ (84) is reduced from $M_{\text{Sub}}$ (80) by realizing $\mathcal{P}_{Y\in\mathcal{Y}_s|X}$ as $\mathcal{P}_{Y\in[K]|X} = 1$. Therefore, we obtain the following reduction path

$$M_{\text{corr}} \to M_{\text{Sub}} \to M_{\text{Soft}}.$$

## 5 Risk Rewrite via Decontamination

We have demonstrated the capability of the proposed formulation component (35) in the last section. This section shows how the proposed framework provides a unified methodology for solving the risk rewrite problem. Specifically, given each contamination matrix described in Section 4, we show how to construct the corrected losses (38) to perform the risk rewrite via (39). We then recover each rewrite to the corresponding form reported in the literature to justify its feasibility. Because this paper focuses on a unified methodology for rewriting the classification risk instead of the designs of practical training objectives, we assume the required parameters are given or can be estimated accurately from the observed data.

Similar to the previous section, we only provide proofs that appear for the first time to avoid repeating similar proofs. For auxiliary lemmas and results whose proofs are similar to the previous ones, we refer to the omitted proofs in Appendix D. In particular, the omitted proofs in Section 5.1 can be found in Appendix D.1, those in Section 5.2 can be found in Appendix D.2, and those in Section 5.3 can be found in Appendix D.3.

### 5.1 MCD Scenarios

We apply the framework to conduct the risk rewrites for WSLs formulated in Section 4.1 and summarized in Table 7. A general approach is to show that the inversion method discussed in Proposition 1 provides the decontamination matrix $M_{\text{corr}}^\dagger$ required in (38).

### 5.1.1 Unlabeled-Unlabeled (UU) Learning

We justify the proposed framework for UU learning via the following steps.

**Step 1: Corrected Loss Design and Risk Rewrite.**
The three milestones in the proposed framework are (1) finding the contamination matrix $M_{\mathrm{corr}}$ that characterizes the data-generating process (35) of a weakly supervised scenario, (2) finding the decontamination matrix $M_{\mathrm{corr}}^{\dagger}$ that compensates for the contamination effect (37), which is then used in (3) the construction of corrected losses (38) for the risk rewrite (39).

Section 4.1.1 has reached the first milestone (35) as (48) of the form $\bar{P} = M_{\mathrm{UU}}B$ finds

$$M_{\mathrm{UU}} = \begin{pmatrix} 1 - \gamma_1 & \gamma_1 \\ \gamma_2 & 1 - \gamma_2 \end{pmatrix}$$

that connects the data-generating distributions $\bar{P} = \begin{pmatrix} \mathcal{P}_{\mathrm{U}_1} \\ \mathcal{P}_{\mathrm{U}_2} \end{pmatrix}$ and the base distributions $B = \begin{pmatrix} \mathcal{P}_{X|Y=\mathrm{p}} \\ \mathcal{P}_{X|Y=\mathrm{n}} \end{pmatrix}$.

Note that $B$ is not the risk-defining distributions $P = \begin{pmatrix} \mathcal{P}_{Y=\mathrm{p},X} \\ \mathcal{P}_{Y=\mathrm{n},X} \end{pmatrix}$, we need an additional step before reaching

the second milestone. To further link $\bar{P}$ with $P$, we still need a $M_{\mathrm{trsf}}$ that satisfies $B = M_{\mathrm{trsf}}P$. Introducing the prior matrix $\Pi = \begin{pmatrix} \pi_{\mathrm{p}} & 0 \\ 0 & \pi_{\mathrm{n}} \end{pmatrix}$, we see that choosing $M_{\mathrm{trsf}} := \Pi^{-1}$ fulfills the need:

$$M_{\mathrm{trsf}}P = \begin{pmatrix} \pi_{\mathrm{p}}^{-1} & 0 \\ 0 & \pi_{\mathrm{n}}^{-1} \end{pmatrix} \begin{pmatrix} \mathcal{P}_{Y=\mathrm{p},X} \\ \mathcal{P}_{Y=\mathrm{n},X} \end{pmatrix} = \begin{pmatrix} \frac{\mathcal{P}_{Y=\mathrm{p},X}}{\mathcal{P}_{Y=\mathrm{p}}} \\ \frac{\mathcal{P}_{Y=\mathrm{n},X}}{\mathcal{P}_{Y=\mathrm{n}}} \end{pmatrix} = \begin{pmatrix} \mathcal{P}_{X|Y=\mathrm{p}} \\ \mathcal{P}_{X|Y=\mathrm{n}} \end{pmatrix} = B.$$

Hence, we can instantiate $\bar{P} = M_{\mathrm{corr}}M_{\mathrm{trsf}}P$ (36) as

$$\begin{pmatrix} \mathcal{P}_{\mathrm{U}_1} \\ \mathcal{P}_{\mathrm{U}_2} \end{pmatrix} = M_{\mathrm{UU}}\Pi^{-1} \begin{pmatrix} \mathcal{P}_{Y=\mathrm{p},X} \\ \mathcal{P}_{Y=\mathrm{n},X} \end{pmatrix} \tag{85}$$

in UU learning.

Next, we use Proposition 1 to derive the decontamination matrix $M_{\mathrm{corr}}^{\dagger}$ to reach the second milestone (37).

**Corollary 20.** *Assume $M_{\mathrm{UU}}$ in (85) is invertible. Then, defining the decontamination matrix for UU learning as*

$$M_{\mathrm{UU}}^{\dagger} := \Pi M_{\mathrm{UU}}^{-1}$$

*gives rise to $M_{\mathrm{UU}}^{\dagger}\bar{P} = P$.*

*Proof.* Suggested by Proposition 1, the inverse matrix $\Pi M_{\mathrm{UU}}^{-1}$ cancels out the contamination brought by $M_{\mathrm{UU}}\Pi^{-1}$ in (85). Assigning $M_{\mathrm{UU}}^{\dagger} = \Pi M_{\mathrm{UU}}^{-1}$ and repeating the proof of Proposition 1, we have

$$M_{\mathrm{UU}}^{\dagger}\bar{P} = \Pi M_{\mathrm{UU}}^{-1}\bar{P} = \Pi M_{\mathrm{UU}}^{-1}M_{\mathrm{UU}}\Pi^{-1}P = P$$

that completes the proof. $\qquad\square$

Now we will move on to the third milestone. With $M_{\mathrm{UU}}^{\dagger}$ in hand, we devise the corrected losses $\bar{L}$ to achieve the risk rewrite for UU learning. We denote the corrected loss at the $\bar{k}$-th entry of $\bar{L}$ as $\bar{\ell}_{\bar{k}} := \ell_{\bar{Y}=\bar{k}}(g(X))$, where $\bar{k} \in \bar{\mathcal{Y}}$ is a class of the observed data[5]. In UU learning, $\bar{\mathcal{Y}} = \{\mathrm{U}_1, \mathrm{U}_2\}$. The following theorem proves rewrite (9) in Section 2.2.3.

---

[5]The definition of the corrected loss $\bar{\ell}_{\bar{k}}$ is in contrast to the original loss $\ell_k := \ell_{Y=k}(g(X))$.

**Theorem 21.** *Let $\gamma_1, \gamma_2 > 0$ and $\gamma_1 + \gamma_2 \neq 1$. Then, $M_{\mathrm{UU}}^{\dagger}$ defined in Corollary 20 is feasible. Moreover, the vector of corrected losses suggested by (38)*

$$\left( \bar{\ell}_{\mathrm{U}_1} \quad \bar{\ell}_{\mathrm{U}_2} \right) = \bar{L}^{\top} := L^{\top} M_{\mathrm{UU}}^{\dagger}$$

*with*

$$
\begin{aligned}
\bar{\ell}_{\mathrm{U}_1} &= \frac{(1 - \gamma_2)\pi_{\mathrm{p}}}{1 - \gamma_1 - \gamma_2} \ell_{\mathrm{p}} + \frac{-\gamma_2 \pi_{\mathrm{n}}}{1 - \gamma_1 - \gamma_2} \ell_{\mathrm{n}}, \\
\bar{\ell}_{\mathrm{U}_2} &= \frac{-\gamma_1 \pi_{\mathrm{p}}}{1 - \gamma_1 - \gamma_2} \ell_{\mathrm{p}} + \frac{(1 - \gamma_1)\pi_{\mathrm{n}}}{1 - \gamma_1 - \gamma_2} \ell_{\mathrm{n}}
\end{aligned}
\tag{86}
$$

*achieves the following risk rewrite:*

$$R(g) = \mathbb{E}_{\mathrm{U}_1}\left[ \bar{\ell}_{\mathrm{U}_1} \right] + \mathbb{E}_{\mathrm{U}_2}\left[ \bar{\ell}_{\mathrm{U}_2} \right]. \tag{87}$$

*Proof.* Since $\gamma_1 + \gamma_2 \neq 1$,

$$
M_{\mathrm{UU}}^{-1} = \begin{pmatrix} 1 - \gamma_1 & \gamma_1 \\ \gamma_2 & 1 - \gamma_2 \end{pmatrix}^{-1} = \begin{pmatrix} \frac{1 - \gamma_2}{1 - \gamma_1 - \gamma_2} & \frac{-\gamma_1}{1 - \gamma_1 - \gamma_2} \\ \frac{-\gamma_2}{1 - \gamma_1 - \gamma_2} & \frac{1 - \gamma_1}{1 - \gamma_1 - \gamma_2} \end{pmatrix}
$$

exists. Thus, it is feasible for us to define $M_{\mathrm{UU}}^{\dagger} := \Pi M_{\mathrm{UU}}^{-1}$ according to Corollary 20. Following (38), we construct

$$\bar{L}^{\top} := L^{\top} M_{\mathrm{UU}}^{\dagger}$$

and obtain

$$
\begin{aligned}
\left( \bar{\ell}_{\mathrm{U}_1} \quad \bar{\ell}_{\mathrm{U}_2} \right) &= L^{\top} \Pi M_{\mathrm{UU}}^{-1} \\
&= \left( \ell_{\mathrm{p}} \quad \ell_{\mathrm{n}} \right) \begin{pmatrix} \pi_{\mathrm{p}} & 0 \\ 0 & \pi_{\mathrm{n}} \end{pmatrix} \begin{pmatrix} \frac{1 - \gamma_2}{1 - \gamma_1 - \gamma_2} & \frac{-\gamma_1}{1 - \gamma_1 - \gamma_2} \\ \frac{-\gamma_2}{1 - \gamma_1 - \gamma_2} & \frac{1 - \gamma_1}{1 - \gamma_1 - \gamma_2} \end{pmatrix} \\
&= \left( \ell_{\mathrm{p}} \quad \ell_{\mathrm{n}} \right) \begin{pmatrix} \frac{(1 - \gamma_2)\pi_{\mathrm{p}}}{1 - \gamma_1 - \gamma_2} & \frac{-\gamma_1 \pi_{\mathrm{p}}}{1 - \gamma_1 - \gamma_2} \\ \frac{-\gamma_2 \pi_{\mathrm{n}}}{1 - \gamma_1 - \gamma_2} & \frac{(1 - \gamma_1)\pi_{\mathrm{n}}}{1 - \gamma_1 - \gamma_2} \end{pmatrix}
\end{aligned}
\tag{88}
$$

that gives (86).

Next, with the critical component $\bar{L}^{\top}$ in hand, applying (39), we obtain

$$
\begin{aligned}
R(g) &= \int_{\mathcal{X}} \bar{L}^{\top} \bar{P} \, \mathrm{d}x \\
&= \int_{\mathcal{X}} \left( \mathcal{P}_{\mathrm{U}_1} \bar{\ell}_{\mathrm{U}_1} + \mathcal{P}_{\mathrm{U}_2} \bar{\ell}_{\mathrm{U}_2} \right) \mathrm{d}x \\
&= \mathbb{E}_{\mathrm{U}_1}\left[ \bar{\ell}_{\mathrm{U}_1} \right] + \mathbb{E}_{\mathrm{U}_2}\left[ \bar{\ell}_{\mathrm{U}_2} \right],
\end{aligned}
\tag{89}
$$

where the first equality holds since according to Corollary 20,

$$\bar{L}^{\top} \bar{P} = L^{\top} M_{\mathrm{UU}}^{\dagger} \bar{P} = L^{\top} P.$$

$\square$

In (88), we do not need to specify the instance in $\ell_{\mathrm{p}}$ and $\ell_{\mathrm{n}}$ to be $x^{\mathrm{u}_1}$ or $x^{\mathrm{u}_2}$ since the equality holds for any instance $x$. We only need to distinguish $x^{\mathrm{u}_1}$ from $x^{\mathrm{u}_2}$ when the corrected losses multiply the data distributions. In particular, the detailed form of rewrite (89) using (8) is

$$
\begin{aligned}
R(g) &= \mathbb{E}_{\mathrm{U}_1}\left[\bar{\ell}_{\mathrm{U}_1}\right] + \mathbb{E}_{\mathrm{U}_2}\left[\bar{\ell}_{\mathrm{U}_2}\right] \\
&= \mathbb{E}_{x^{\mathrm{u}_1}\sim\mathcal{P}_{\mathrm{U}_1}}\left[\frac{(1-\gamma_2)\pi_{\mathrm{p}}}{1-\gamma_1-\gamma_2}\ell_{\mathrm{p}}(X^{\mathrm{u}_1}) + \frac{-\gamma_2\pi_{\mathrm{n}}}{1-\gamma_1-\gamma_2}\ell_{\mathrm{n}}(X^{\mathrm{u}_1})\right] \\
&\quad + \mathbb{E}_{x^{\mathrm{u}_2}\sim\mathcal{P}_{\mathrm{U}_2}}\left[\frac{-\gamma_1\pi_{\mathrm{p}}}{1-\gamma_1-\gamma_2}\ell_{\mathrm{p}}(X^{\mathrm{u}_2}) + \frac{(1-\gamma_1)\pi_{\mathrm{n}}}{1-\gamma_1-\gamma_2}\ell_{\mathrm{n}}(X^{\mathrm{u}_2})\right].
\end{aligned}
$$

The freedom from specifying $x$ in (88) eliminates the notational burden of distinguishing $\ell_Y(X^{\mathrm{u}_1})$ from $\ell_Y(X^{\mathrm{u}_2})$, allowing us to exploit the advantage of matrix multiplication while constructing the corrected losses. The freedom also enables separated treatments for the data distributions (e.g., formulating $\bar{P} = M_{\mathrm{UU}}\Pi^{-1}P$) and the corrected losses (e.g., devising $\bar{L}^\top = L^\top M_{\mathrm{UU}}^\dagger$).

**Step 2: Recovering the previous result(s).**
Lastly, we verify the feasibility of our rewrite by showing that our rewrite corresponds to an existing result. By parameter substitution, we replace $\gamma_1$ with $1-\theta$, $\gamma_2$ with $\theta'$, $\pi_{\mathrm{n}}$ with $1-\pi_{\mathrm{p}}$, $\ell_{\mathrm{p}}$ with $\ell(g(X))$, and $\ell_{\mathrm{n}}$ with $\ell(-g(X))$. Then, (86) becomes

$$
\begin{aligned}
\frac{(1-\theta')\pi_{\mathrm{p}}}{\theta-\theta'}\ell(g(X)) + \frac{-\theta'(1-\pi_{\mathrm{p}})}{\theta-\theta'}\ell(-g(X)) &= \bar{\ell}_+(g(X)), \\
\frac{\theta(1-\pi_{\mathrm{p}})}{\theta-\theta'}\ell(-g(X)) + \frac{-(1-\theta)\pi_{\mathrm{p}}}{\theta-\theta'}\ell(g(X)) &= \bar{\ell}_-(-g(X)),
\end{aligned}
$$

recovering the corrected loss functions (8) and the constants reported in Theorem 4 of Lu et al. (2019).

### 5.1.2 Positive-Unlabeled (PU) Learning

Recall that all WSLs discussed in Section 4.1 share the same base distributions $B$ (46). Further, as shown in Table 7, the contamination matrix of every WSL scenario beneath UU learning except $M_{\mathrm{Sconf}}$ is a child of $M_{\mathrm{UU}}$ on the reduction graph. It means $\bar{P} = M_{\mathrm{UU}}\Pi^{-1}P$ (85) is a general form for every child scenario in Table 7 (with different realizations of $\gamma_1$ and $\gamma_2$). Hence, we can reuse Theorem 21 to conduct the risk rewrite for every child scenario on the reduction graph. PU learning is the first of such examples.

**Step 1: Corrected Loss Design and Risk Rewrite.**
By the following corollary, we prove the rewrite (5) in Section 2.2.1.

**Corollary 22.** *For PU learning, the classification risk can be rewritten as*

$$
R(g) = \mathbb{E}_{\mathrm{P}}\left[\bar{\ell}_{\mathrm{P}}\right] + \mathbb{E}_{\mathrm{U}}\left[\bar{\ell}_{\mathrm{U}}\right], \tag{90}
$$

*where*

$$
\begin{aligned}
\bar{\ell}_{\mathrm{P}} &= \pi_{\mathrm{p}}\ell_{\mathrm{p}} - \pi_{\mathrm{p}}\ell_{\mathrm{n}}, \\
\bar{\ell}_{\mathrm{U}} &= \ell_{\mathrm{n}}.
\end{aligned}
\tag{91}
$$

**Step 2: Recovering the previous result(s).**
Since $\mathcal{P}_{\mathrm{P}}$ is $\mathcal{P}_{X|Y=\mathrm{p}}$ and $\mathcal{P}_{\mathrm{U}}$ is $\mathcal{P}_X$, we swap the notations to obtain

$$
\begin{aligned}
R(g) &= \mathbb{E}_{\mathrm{P}}\left[\bar{\ell}_{\mathrm{P}}\right] + \mathbb{E}_{\mathrm{U}}\left[\bar{\ell}_{\mathrm{U}}\right] \\
&= \mathbb{E}_{\mathrm{P}}\left[\pi_{\mathrm{p}}\ell_{\mathrm{p}} - \pi_{\mathrm{p}}\ell_{\mathrm{n}}\right] + \mathbb{E}_{\mathrm{U}}\left[\ell_{\mathrm{n}}\right] \\
&= \pi_{\mathrm{p}}\mathbb{E}_{X|Y=\mathrm{p}}\left[\ell_{\mathrm{p}}\right] - \pi_{\mathrm{p}}\mathbb{E}_{X|Y=\mathrm{p}}\left[\ell_{\mathrm{n}}\right] + \mathbb{E}_X\left[\ell_{\mathrm{n}}\right]
\end{aligned}
$$

from (90), which corresponds to the risk estimators (2) in Kiryo et al. (2017) and (3) in du Plessis et al. (2015).

Moreover, with an additional symmetric assumption of $\ell_{\mathrm{p}} + \ell_{\mathrm{n}} = 1$, one further obtains

$$
\begin{aligned}
R(g) &= \pi_{\mathrm{p}} \mathbb{E}_{X|Y=\mathrm{p}}\left[\ell_{\mathrm{p}}\right] - \pi_{\mathrm{p}} \mathbb{E}_{X|Y=\mathrm{p}}\left[\ell_{\mathrm{n}}\right] + \mathbb{E}_X\left[\ell_{\mathrm{n}}\right] \\
&= \pi_{\mathrm{p}} \mathbb{E}_{X|Y=\mathrm{p}}\left[\ell_{\mathrm{p}}\right] - \pi_{\mathrm{p}} \mathbb{E}_{X|Y=\mathrm{p}}\left[1 - \ell_{\mathrm{p}}\right] + \mathbb{E}_X\left[\ell_{\mathrm{n}}\right] \\
&= \pi_{\mathrm{p}} \mathbb{E}_{X|Y=\mathrm{p}}\left[\ell_{\mathrm{p}}\right] - \pi_{\mathrm{p}} \mathbb{E}_{X|Y=\mathrm{p}}\left[1\right] + \pi_{\mathrm{p}} \mathbb{E}_{X|Y=\mathrm{p}}\left[\ell_{\mathrm{p}}\right] + \mathbb{E}_X\left[\ell_{\mathrm{n}}\right] \\
&= 2\pi_{\mathrm{p}} \mathbb{E}_{X|Y=\mathrm{p}}\left[\ell_{\mathrm{p}}\right] - \pi_{\mathrm{p}} + \mathbb{E}_X\left[\ell_{\mathrm{n}}\right].
\end{aligned}
$$

This expression recovers several risk rewrites such as (4) of Kiryo et al. (2017), (3) of Niu et al. (2016), (2) of du Plessis et al. (2015)[6], and (3) of du Plessis et al. (2014).

### 5.1.3  Similar-Unlabeled (SU) Learning

According to Table 7, $M_{\mathrm{SU}}$ is a child of $M_{\mathrm{UU}}$ on the reduction graph. Thus, we can follow the same steps illustrated in Section 5.1.2 to justify the proposed framework.

**Step 1: Corrected Loss Design and Risk Rewrite.**
The following corollary combines (86) and (87) to conduct the risk rewrite.

**Corollary 23.** *Assume $\pi_{\mathrm{p}} \neq 1/2$. For SU learning, the classification risk can be rewritten as*

$$
R(g) = \mathbb{E}_{\tilde{\mathrm{S}}}\left[\bar{\ell}_{\tilde{\mathrm{S}}}\right] + \mathbb{E}_{\mathrm{U}}\left[\bar{\ell}_{\mathrm{U}}\right],
$$

*where*

$$
\begin{aligned}
\bar{\ell}_{\tilde{\mathrm{S}}} &= \frac{\pi_{\mathrm{p}}^2 + \pi_{\mathrm{n}}^2}{2\pi_{\mathrm{p}} - 1}\ell_{\mathrm{p}} - \frac{\pi_{\mathrm{p}}^2 + \pi_{\mathrm{n}}^2}{2\pi_{\mathrm{p}} - 1}\ell_{\mathrm{n}}, \\
\bar{\ell}_{\mathrm{U}} &= -\frac{\pi_{\mathrm{n}}}{2\pi_{\mathrm{p}} - 1}\ell_{\mathrm{p}} + \frac{\pi_{\mathrm{p}}}{2\pi_{\mathrm{p}} - 1}\ell_{\mathrm{n}}.
\end{aligned}
\tag{92}
$$

**Step 2: Recovering the previous result(s).**
To recover Theorem 1 of Bao et al. (2018), we first need to restore $\mathbb{E}_{\mathrm{S}}\left[\cdot\right]$ from $\mathbb{E}_{\tilde{\mathrm{S}}}\left[\cdot\right]$ in Corollary 23. The following lemma provides a means for us to do so.

**Lemma 24.** *Given B (46) and following the SU learning notations, we have*

$$
M'_{\mathrm{SU}} B = \begin{pmatrix} \mathcal{P}_{\tilde{\mathrm{S}}} \\ \mathcal{P}_{\mathrm{U}} \end{pmatrix} = \bar{P},
$$

*where*

$$
M'_{\mathrm{SU}} := \begin{pmatrix} \frac{\pi_{\mathrm{p}}^2 \int_{x' \in \mathcal{X}} \mathcal{P}_{x'|Y=\mathrm{p}} \mathrm{d}x'}{\pi_{\mathrm{p}}^2 + \pi_{\mathrm{n}}^2} & \frac{\pi_{\mathrm{n}}^2 \int_{x' \in \mathcal{X}} \mathcal{P}_{x'|Y=\mathrm{n}} \mathrm{d}x'}{\pi_{\mathrm{p}}^2 + \pi_{\mathrm{n}}^2} \\ \pi_{\mathrm{p}} & \pi_{\mathrm{n}} \end{pmatrix}.
$$

*Proof.* Since $\int_{x' \in \mathcal{X}} \mathcal{P}_{x'|Y=\mathrm{p}} \mathrm{d}x' = 1$ and $\int_{x' \in \mathcal{X}} \mathcal{P}_{x'|Y=\mathrm{n}} \mathrm{d}x' = 1$, we have $M'_{\mathrm{SU}} = M_{\mathrm{SU}}$, and hence $M'_{\mathrm{SU}} B = M_{\mathrm{SU}} B = \begin{pmatrix} \mathcal{P}_{\tilde{\mathrm{S}}} \\ \mathcal{P}_{\mathrm{U}} \end{pmatrix}$. The last equality follows from Lemma 5. $\qquad\square$

---

[6]As the 0-1 loss is symmetric.

Lemma 24 allows us to slightly revise the derivation (89) as follows:

$$
\begin{aligned}
R(g) &= \int_{x \in \mathcal{X}} \bar{L}^\top \bar{P} \, \mathrm{d}x = \int_{x \in \mathcal{X}} \bar{L}^\top M'_{\mathrm{SU}} B \, \mathrm{d}x \\
&= \int_{x \in \mathcal{X}} \begin{pmatrix} \bar{\ell}_{\tilde{\mathrm{S}}} & \bar{\ell}_{\mathrm{U}} \end{pmatrix} \begin{pmatrix} \frac{\pi_{\mathrm{p}}^2 \int_{x' \in \mathcal{X}} \mathcal{P}_{x'|Y=\mathrm{p}} \mathrm{d}x'}{\pi_{\mathrm{p}}^2 + \pi_{\mathrm{n}}^2} & \frac{\pi_{\mathrm{n}}^2 \int_{x' \in \mathcal{X}} \mathcal{P}_{x'|Y=\mathrm{n}} \mathrm{d}x'}{\pi_{\mathrm{p}}^2 + \pi_{\mathrm{n}}^2} \\ \pi_{\mathrm{p}} & \pi_{\mathrm{n}} \end{pmatrix} \begin{pmatrix} \mathcal{P}_{x|Y=\mathrm{p}} \\ \mathcal{P}_{x|Y=\mathrm{n}} \end{pmatrix} \mathrm{d}x \\
&\overset{\text{(a)}}{=} \int_{x \in \mathcal{X}} \int_{x' \in \mathcal{X}} \mathcal{P}_{\mathrm{S}} \bar{\ell}_{\tilde{\mathrm{S}}} \, \mathrm{d}x' \mathrm{d}x + \int_{x \in \mathcal{X}} \mathcal{P}_{\mathrm{U}} \bar{\ell}_{\mathrm{U}} \, \mathrm{d}x \\
&= \mathbb{E}_{\mathrm{S}} \left[ \bar{\ell}_{\tilde{\mathrm{S}}} \right] + \mathbb{E}_{\mathrm{U}} \left[ \bar{\ell}_{\mathrm{U}} \right],
\end{aligned}
$$

where equality (a) follows from the SU formulation (10).

Then, denoting

$$
\begin{aligned}
\mathcal{L}(X) &:= \frac{1}{\pi_{\mathrm{p}} - \pi_{\mathrm{n}}} \ell_{\mathrm{p}}(X) - \frac{1}{\pi_{\mathrm{p}} - \pi_{\mathrm{n}}} \ell_{\mathrm{n}}(X), & (93) \\
\mathcal{L}_{-}(X) &:= -\frac{\pi_{\mathrm{n}}}{\pi_{\mathrm{p}} - \pi_{\mathrm{n}}} \ell_{\mathrm{p}}(X) + \frac{\pi_{\mathrm{p}}}{\pi_{\mathrm{p}} - \pi_{\mathrm{n}}} \ell_{\mathrm{n}}(X) & (94)
\end{aligned}
$$

and continuing with (92), we obtain

$$
\begin{aligned}
\mathbb{E}_{\mathrm{S}} \left[ \bar{\ell}_{\tilde{\mathrm{S}}} \right] &= \left( \pi_{\mathrm{p}}^2 + \pi_{\mathrm{n}}^2 \right) \mathbb{E}_{\mathrm{S}} \left[ \frac{1}{2\pi_{\mathrm{p}} - 1} \left( \ell_{\mathrm{p}} - \ell_{\mathrm{n}} \right) \right] \\
&= \left( \pi_{\mathrm{p}}^2 + \pi_{\mathrm{n}}^2 \right) \mathbb{E}_{\mathrm{S}} \left[ \mathcal{L}(X) \right] \\
&\overset{\text{(b)}}{=} \left( \pi_{\mathrm{p}}^2 + \pi_{\mathrm{n}}^2 \right) \mathbb{E}_{\mathrm{S}} \left[ \frac{\mathcal{L}(X) + \mathcal{L}(X')}{2} \right]
\end{aligned}
$$

and

$$
\begin{aligned}
\mathbb{E}_{\mathrm{U}} \left[ \bar{\ell}_{\mathrm{U}} \right] &= \mathbb{E}_{\mathrm{U}} \left[ -\frac{\pi_{\mathrm{n}}}{2\pi_{\mathrm{p}} - 1} \ell_{\mathrm{p}} + \frac{\pi_{\mathrm{p}}}{2\pi_{\mathrm{p}} - 1} \ell_{\mathrm{n}} \right] \\
&= \mathbb{E}_{\mathrm{U}} \left[ \mathcal{L}_{-}(f(X)) \right] & (95)
\end{aligned}
$$

that prove rewrite (11) in Section 2.2.4 and recover Theorem 1 of Bao et al. (2018) by matching notations[7]. The following lemma justifies equality (b).

**Lemma 25.** *Let* $(x, x') \sim \mathcal{P}_{\mathrm{S}}$ *defined by (10). Then,* $\mathbb{E}_{\mathrm{S}} \left[ \frac{\mathcal{L}(X)}{2} \right] = \mathbb{E}_{\mathrm{S}} \left[ \frac{\mathcal{L}(X')}{2} \right].$

The derivation demonstrates the flexibility of the proposed framework in which a slight modification of $M_{\mathrm{SU}}$ recovers the pairwise distribution $\mathcal{P}_{\mathrm{S}}$ required for $\mathbb{E}_{\mathrm{S}} [\cdot]$. Moreover, the technique developed here significantly reduces the proof length in Appendix B of Bao et al. (2018). Later in Section 5.1.5, we apply the same trick to recover Theorem 1 of Shimada et al. (2021) for SDU learning.

We remark that the result recovered in this paper is merely Theorem 1 of Bao et al. (2018) but not the last expression in (5) of Bao et al. (2018), which later was implemented as the objective (10) for optimization. It is because, pointed out by Negishi (2023), the additional assumption $\mathcal{P}_{\mathrm{S}}(x, x') = \mathcal{P}_{\tilde{\mathrm{S}}}(x) \mathcal{P}_{\tilde{\mathrm{S}}}(x')$ required for achieving (5) of Bao et al. (2018) is impractical. We note that the remedy proposed by Negishi (2023) can be analyzed by the proposed framework, but we omit it due to the amount of overlap with the analyses in Sections 5.1.1 and 5.1.3.

---

[7]The matching to the notations of Bao et al. (2018) is as follows: $\pi_{\mathrm{p}}$ is $\pi_{+}$, $\pi_{\mathrm{n}}$ is $\pi_{-}$, $\pi_{\mathrm{p}}^2 + \pi_{\mathrm{n}}^2$ is $\pi_{\mathrm{S}}$, $\mathcal{P}_{\mathrm{S}}$ is $p_{\mathrm{S}}$, $\mathcal{P}_{\mathrm{U}}$ is $p$, $\ell_{\mathrm{p}}$ is $\ell(f(X), +1)$, $\ell_{\mathrm{n}}$ is $\ell(f(X), -1)$, $\mathcal{L}(X)$ by definition is $\frac{1}{2\pi_{+} - 1} \left( \ell(f(X), +1) - \ell(f(X), -1) \right)$, and $\mathcal{L}_{-}(f(X))$ by definition is $-\frac{\pi_{-}}{2\pi_{+} - 1} \ell(f(X), +1) + \frac{\pi_{+}}{2\pi_{+} - 1} \ell(f(X), -1)$.

### 5.1.4 Pairwise Comparison (Pcomp) Learning

We follow the steps illustrated in Section 5.1.2 to justify the proposed framework since, by Table 7, $M_{\text{Pcomp}}$ is reduced from $M_{\text{UU}}$.

**Step 1: Corrected Loss Design and Risk Rewrite.**
The following corollary combines (86) and (87) to achieve rewrite (17) in Section 2.2.7.

**Corollary 26.** *For Pcomp learning, the classification risk can be rewritten as*

$$R(g) = \mathbb{E}_{\text{Sup}} \left[ \bar{\ell}_{\text{Sup}} \right] + \mathbb{E}_{\text{Inf}} \left[ \bar{\ell}_{\text{Inf}} \right],$$

*where*

$$
\begin{aligned}
\bar{\ell}_{\text{Sup}} &= \ell_{\text{p}} - \pi_{\text{p}} \ell_{\text{n}}, \\
\bar{\ell}_{\text{Inf}} &= -\pi_{\text{n}} \ell_{\text{p}} + \ell_{\text{n}}.
\end{aligned}
\tag{96}
$$

**Step 2: Recovering the previous result(s).**
It is straightforward to recover Theorem 3 of Feng et al. (2021) by matching notations[8]. Since $x$ is a variable and can be substituted by $x'$, we express Corollary 26 as

$$R(g) = \mathbb{E}_{x \sim \mathcal{P}_{\text{Sup}}} \left[ \ell_{\text{p}}(x) - \pi_{\text{p}} \ell_{\text{n}}(x) \right] + \mathbb{E}_{x' \sim \mathcal{P}_{\text{Inf}}} \left[ \ell_{\text{n}}(x') - \pi_{\text{n}} \ell_{\text{p}}(x') \right], \tag{97}$$

recovering (5) of Feng et al. (2021).

### 5.1.5 Similar-dissimilar-unlabeled (SDU) Learning

We justify the applicability of the proposed framework for DU and SD separately. Firstly, we start with DU learning, which is similar to SU learning in the sense that pairwise information is provided. From Lemmas 5 and 7, we see that the pairwise distributions are treated similarly. Thus, following the same steps in Section 5.1.3, we conduct the risk rewrite for DU learning.

**Step 1: Corrected Loss Design and Risk Rewrite for DU Learning.**
The following corollary is a variant of Corollary 23.

**Corollary 27.** *Assume $\pi_{\text{p}} \neq 1/2$. For DU learning, the classification risk can be rewritten as*

$$R(g) = \mathbb{E}_{\tilde{\text{D}}} \left[ \bar{\ell}_{\tilde{\text{D}}} \right] + \mathbb{E}_{\text{U}} \left[ \bar{\ell}_{\text{U}} \right],$$

*where*

$$
\begin{aligned}
\bar{\ell}_{\tilde{\text{D}}} &= 2\pi_{\text{p}}\pi_{\text{n}} \left( \frac{1}{\pi_{\text{n}} - \pi_{\text{p}}} \ell_{\text{p}} - \frac{1}{\pi_{\text{n}} - \pi_{\text{p}}} \ell_{\text{n}} \right), \\
\bar{\ell}_{\text{U}} &= -\frac{\pi_{\text{p}}}{\pi_{\text{n}} - \pi_{\text{p}}} \ell_{\text{p}} + \frac{\pi_{\text{n}}}{\pi_{\text{n}} - \pi_{\text{p}}} \ell_{\text{n}}.
\end{aligned}
\tag{98}
$$

**Step 2: Recovering the previous result(s) for DU Learning.**
We reuse the trick in Lemma 24 for restoring the pairwise distribution $\mathcal{P}_{\text{S}}$ to restore $\mathcal{P}_{\text{D}}$ needed here, allowing us to recover the rewrite (15) in Theorem 1 of Shimada et al. (2021) and the first result in Theorem 7.3 of Sugiyama et al. (2022). The derivation resembles that of SU learning. We start with the next lemma, adapted from Lemma 24.

**Lemma 28.** *Given B (46) and following the DU learning notations, we have*

$$M'_{\text{DU}} B = \begin{pmatrix} \mathcal{P}_{\tilde{\text{D}}} \\ \mathcal{P}_{\text{U}} \end{pmatrix} = \bar{P},$$

---

[8]The matching is as follows: $\mathcal{P}_{\text{Sup}}$ is $\tilde{p}_{+}(x)$, $\mathcal{P}_{\text{Inf}}$ is $\tilde{p}_{-}(x)$, $\ell_{\text{p}}$ is $\ell(f(x), +1)$, and $\ell_{\text{n}}$ is $\ell(f(x), -1)$.

*where*

$$M'_{\mathrm{DU}} := \begin{pmatrix} \dfrac{\int_{x' \in \mathcal{X}} \mathcal{P}_{x'|Y=n} \mathrm{d}x'}{2} & \dfrac{\int_{x' \in \mathcal{X}} \mathcal{P}_{x'|Y=p} \mathrm{d}x'}{2} \\ \pi_{\mathrm{p}} & \pi_{\mathrm{n}} \end{pmatrix}.$$

We apply Lemma 28 to slightly revise the derivation of (89) as follows:

$$
\begin{aligned}
R(g) &= \int_{x \in \mathcal{X}} \bar{L}^\top \bar{P} \,\mathrm{d}x = \int_{x \in \mathcal{X}} \bar{L}^\top M'_{\mathrm{DU}} B \,\mathrm{d}x \\
&= \int_{x \in \mathcal{X}} \begin{pmatrix} \bar{\ell}_{\tilde{\mathrm{D}}} & \bar{\ell}_{\mathrm{U}} \end{pmatrix} \begin{pmatrix} \dfrac{\int_{x' \in \mathcal{X}} \mathcal{P}_{x'|Y=n} \mathrm{d}x'}{2} & \dfrac{\int_{x' \in \mathcal{X}} \mathcal{P}_{x'|Y=p} \mathrm{d}x'}{2} \\ \pi_{\mathrm{p}} & \pi_{\mathrm{n}} \end{pmatrix} \begin{pmatrix} \mathcal{P}_{x|Y=p} \\ \mathcal{P}_{x|Y=n} \end{pmatrix} \mathrm{d}x \\
&= \int_{x \in \mathcal{X}} \int_{x' \in \mathcal{X}} \mathcal{P}_{\mathrm{D}} \, \bar{\ell}_{\tilde{\mathrm{D}}} \,\mathrm{d}x' \mathrm{d}x + \int_{x \in \mathcal{X}} \mathcal{P}_{\mathrm{U}} \, \bar{\ell}_{\mathrm{U}} \,\mathrm{d}x \\
&= \mathbb{E}_{\mathrm{D}} \left[ \bar{\ell}_{\tilde{\mathrm{D}}} \right] + \mathbb{E}_{\mathrm{U}} \left[ \bar{\ell}_{\mathrm{U}} \right],
\end{aligned}
$$

where the second to last equality follows from the DU formulation (12). Denoting

$$\mathcal{L}_+(X) := \frac{\pi_{\mathrm{p}}}{\pi_{\mathrm{p}} - \pi_{\mathrm{n}}} \ell_{\mathrm{p}}(X) - \frac{\pi_{\mathrm{n}}}{\pi_{\mathrm{p}} - \pi_{\mathrm{n}}} \ell_{\mathrm{n}}(X), \tag{99}$$

recalling $\mathcal{L}(X)$ from (93), and continuing with (98), we have

$$
\begin{aligned}
\mathbb{E}_{\mathrm{D}} \left[ \bar{\ell}_{\tilde{\mathrm{D}}} \right] &= 2\pi_{\mathrm{p}} \pi_{\mathrm{n}} \mathbb{E}_{\mathrm{D}} \left[ \frac{1}{\pi_{\mathrm{n}} - \pi_{\mathrm{p}}} \ell_{\mathrm{p}} - \frac{1}{\pi_{\mathrm{n}} - \pi_{\mathrm{p}}} \ell_{\mathrm{n}} \right] \\
&= 2\pi_{\mathrm{p}} \pi_{\mathrm{n}} \mathbb{E}_{\mathrm{D}} \left[ -\mathcal{L}(X) \right] \\
&\overset{(a)}{=} 2\pi_{\mathrm{p}} \pi_{\mathrm{n}} \mathbb{E}_{\mathrm{D}} \left[ -\frac{\mathcal{L}(X) + \mathcal{L}(X')}{2} \right]
\end{aligned}
$$

and

$$
\begin{aligned}
\mathbb{E}_{\mathrm{U}} \left[ \bar{\ell}_{\mathrm{U}} \right] &= \mathbb{E}_{\mathrm{U}} \left[ -\frac{\pi_{\mathrm{p}}}{\pi_{\mathrm{n}} - \pi_{\mathrm{p}}} \ell_{\mathrm{p}} + \frac{\pi_{\mathrm{n}}}{\pi_{\mathrm{n}} - \pi_{\mathrm{p}}} \ell_{\mathrm{n}} \right] \\
&= \mathbb{E}_{\mathrm{U}} \left[ \mathcal{L}_+(X) \right] \tag{100}
\end{aligned}
$$

that prove rewrite (13) in Section 2.2.5. By matching notations, we recover (15) in Theorem 1 of Shimada et al. (2021) [9] . Equality (a) follows from the next lemma.

**Lemma 29.** *Let* $(x, x') \sim \mathcal{P}_{\mathrm{D}}$ *defined in (12). Then,* $\mathbb{E}_{\mathrm{D}} \left[ \frac{\mathcal{L}(X)}{2} \right] = \mathbb{E}_{\mathrm{D}} \left[ \frac{\mathcal{L}(X')}{2} \right].$

Secondly, we consider the rewrite of SD learning. To do so, we apply the knowledge acquired from SU and DU learning (Corollaries 23 and 27).

**Step 1: Corrected Loss Design and Risk Rewrite for SD Learning.**
We provide another variant of Corollary 23 to conduct the risk rewrite.

**Corollary 30.** *Assume* $\pi_{\mathrm{p}} \neq 1/2$. *For SD learning, the classification risk can be rewritten as*

$$R(g) = \mathbb{E}_{\tilde{\mathrm{S}}} \left[ \bar{\ell}_{\tilde{\mathrm{S}}} \right] + \mathbb{E}_{\tilde{\mathrm{D}}} \left[ \bar{\ell}_{\tilde{\mathrm{D}}} \right],$$

*where*

$$
\begin{aligned}
\bar{\ell}_{\tilde{\mathrm{S}}} &= \left( \pi_{\mathrm{p}}^2 + \pi_{\mathrm{n}}^2 \right) \left( \frac{\pi_{\mathrm{p}}}{\pi_{\mathrm{p}} - \pi_{\mathrm{n}}} \ell_{\mathrm{p}} - \frac{\pi_{\mathrm{n}}}{\pi_{\mathrm{p}} - \pi_{\mathrm{n}}} \ell_{\mathrm{n}} \right), \\
\bar{\ell}_{\tilde{\mathrm{D}}} &= 2\pi_{\mathrm{p}} \pi_{\mathrm{n}} \left( -\frac{\pi_{\mathrm{n}}}{\pi_{\mathrm{p}} - \pi_{\mathrm{n}}} \ell_{\mathrm{p}} + \frac{\pi_{\mathrm{p}}}{\pi_{\mathrm{p}} - \pi_{\mathrm{n}}} \ell_{\mathrm{n}} \right). \tag{101}
\end{aligned}
$$

---

[9] The matching to the notations of Shimada et al. (2021) is as follows: $\pi_{\mathrm{p}}$ is $\pi_+$, $\pi_{\mathrm{n}}$ is $\pi_-$, $\pi_{\mathrm{p}}^2 + \pi_{\mathrm{n}}^2$ is $\pi_{\mathrm{S}}$, $2\pi_{\mathrm{p}} \pi_{\mathrm{n}}$ is $\pi_{\mathrm{D}}$, $\mathcal{P}_{\mathrm{S}}$ is $p_{\mathrm{S}}(x, x')$, $\mathcal{P}_{\mathrm{D}}$ is $p_{\mathrm{D}}(x, x')$, $\mathcal{P}_{\mathrm{U}}$ is $p_{\mathrm{U}}(x)$, $\ell_{\mathrm{p}}$ is $\ell(f(X), +1)$, $\ell_{\mathrm{n}}$ is $\ell(f(X), -1)$, $\mathcal{L}(X)$ is $\tilde{\mathcal{L}}(f(X))$, $\mathcal{L}_+(X)$ is $\mathcal{L}(f(X), +1)$, and $\mathcal{L}_-(X)$ is $\mathcal{L}(f(X), -1)$.

**Step 2: Recovering the previous result(s) for SD Learning.**

We apply the same strategy as in Lemma 28 to obtain the needed $\mathcal{P}_S$ and $\mathcal{P}_D$. We begin with the next lemma, adapted from Lemma 24, to recover (16) in Theorem 1 of Shimada et al. (2021) and the second result in Theorem 7.3 of Sugiyama et al. (2022).

**Lemma 31.** *Given B (46) and following the SD learning notations, we have*

$$M'_{\mathrm{SD}}B = \begin{pmatrix} \mathcal{P}_{\tilde{\mathrm{S}}} \\ \mathcal{P}_{\tilde{\mathrm{D}}} \end{pmatrix} = \bar{P},$$

*where*

$$M'_{\mathrm{SD}} := \begin{pmatrix} \dfrac{\pi_{\mathrm{p}}^2 \int_{x' \in \mathcal{X}} \mathcal{P}_{x'|Y=\mathrm{p}} \mathrm{d}x'}{\pi_{\mathrm{p}}^2 + \pi_{\mathrm{n}}^2} & \dfrac{\pi_{\mathrm{n}}^2 \int_{x' \in \mathcal{X}} \mathcal{P}_{x'|Y=\mathrm{n}} \mathrm{d}x'}{\pi_{\mathrm{p}}^2 + \pi_{\mathrm{n}}^2} \\[2mm] \dfrac{\int_{x' \in \mathcal{X}} \mathcal{P}_{x'|Y=\mathrm{n}} \mathrm{d}x'}{2} & \dfrac{\int_{x' \in \mathcal{X}} \mathcal{P}_{x'|Y=\mathrm{p}} \mathrm{d}x'}{2} \end{pmatrix}.$$

We apply Lemma 31 to slightly revise the derivation of (89) as follows:

$$
\begin{aligned}
R(g) &= \int_{x \in \mathcal{X}} \bar{L}^\top \bar{P} \,\mathrm{d}x = \int_{x \in \mathcal{X}} \bar{L}^\top M'_{\mathrm{SD}} B \,\mathrm{d}x \\
&= \int_{x \in \mathcal{X}} \begin{pmatrix} \bar{\ell}_{\tilde{\mathrm{S}}} & \bar{\ell}_{\tilde{\mathrm{D}}} \end{pmatrix} \begin{pmatrix} \dfrac{\pi_{\mathrm{p}}^2 \int_{x' \in \mathcal{X}} \mathcal{P}_{x'|Y=\mathrm{p}} \mathrm{d}x'}{\pi_{\mathrm{p}}^2 + \pi_{\mathrm{n}}^2} & \dfrac{\pi_{\mathrm{n}}^2 \int_{x' \in \mathcal{X}} \mathcal{P}_{x'|Y=\mathrm{n}} \mathrm{d}x'}{\pi_{\mathrm{p}}^2 + \pi_{\mathrm{n}}^2} \\[2mm] \dfrac{\int_{x' \in \mathcal{X}} \mathcal{P}_{x'|Y=\mathrm{n}} \mathrm{d}x'}{2} & \dfrac{\int_{x' \in \mathcal{X}} \mathcal{P}_{x'|Y=\mathrm{p}} \mathrm{d}x'}{2} \end{pmatrix} \begin{pmatrix} \mathcal{P}_{x|Y=\mathrm{p}} \\ \mathcal{P}_{x|Y=\mathrm{n}} \end{pmatrix} \mathrm{d}x \\
&= \int_{x \in \mathcal{X}} \int_{x' \in \mathcal{X}} \mathcal{P}_S \, \bar{\ell}_{\tilde{\mathrm{S}}} \,\mathrm{d}x' \mathrm{d}x + \int_{x \in \mathcal{X}} \int_{x' \in \mathcal{X}'} \mathcal{P}_D \, \bar{\ell}_{\tilde{\mathrm{D}}} \,\mathrm{d}x' \mathrm{d}x \\
&= \mathbb{E}_S \left[ \bar{\ell}_{\tilde{\mathrm{S}}} \right] + \mathbb{E}_D \left[ \bar{\ell}_{\tilde{\mathrm{D}}} \right],
\end{aligned}
$$

where the second to last equality follows from the SD formulation (14). Recalling $\mathcal{L}_+(X)$ (99) and $\mathcal{L}_-(X)$ (94) and continuing with (101),

$$
\begin{aligned}
\mathbb{E}_S \left[ \bar{\ell}_{\tilde{\mathrm{S}}} \right] &= \left( \pi_{\mathrm{p}}^2 + \pi_{\mathrm{n}}^2 \right) \mathbb{E}_S \left[ \frac{\pi_{\mathrm{p}}}{\pi_{\mathrm{p}} - \pi_{\mathrm{n}}} \ell_{\mathrm{p}} - \frac{\pi_{\mathrm{n}}}{\pi_{\mathrm{p}} - \pi_{\mathrm{n}}} \ell_{\mathrm{n}} \right] \\
&= \left( \pi_{\mathrm{p}}^2 + \pi_{\mathrm{n}}^2 \right) \mathbb{E}_S \left[ \mathcal{L}_+(X) \right] \\
&\overset{\text{(b)}}{=} \left( \pi_{\mathrm{p}}^2 + \pi_{\mathrm{n}}^2 \right) \mathbb{E}_S \left[ \frac{\mathcal{L}_+(X) + \mathcal{L}_+(X')}{2} \right]
\end{aligned}
$$

and

$$
\begin{aligned}
\mathbb{E}_D \left[ \bar{\ell}_{\tilde{\mathrm{D}}} \right] &= 2\pi_{\mathrm{p}} \pi_{\mathrm{n}} \mathbb{E}_D \left[ -\frac{\pi_{\mathrm{n}}}{\pi_{\mathrm{p}} - \pi_{\mathrm{n}}} \ell_{\mathrm{p}} + \frac{\pi_{\mathrm{p}}}{\pi_{\mathrm{p}} - \pi_{\mathrm{n}}} \ell_{\mathrm{n}} \right] \\
&= 2\pi_{\mathrm{p}} \pi_{\mathrm{n}} \mathbb{E}_D \left[ \mathcal{L}_-(X) \right] \\
&\overset{\text{(c)}}{=} 2\pi_{\mathrm{p}} \pi_{\mathrm{n}} \mathbb{E}_D \left[ \frac{\mathcal{L}_-(X) + \mathcal{L}_-(X')}{2} \right]
\end{aligned}
$$

prove rewrite (15) in Section 2.2.6. We also recover (16) in Theorem 1 of Shimada et al. (2021) via matching notations. The required matches can be found in the paragraph before Lemma 29. The equality (b) holds by applying Lemma 25 with $\mathcal{L}(X)$ replaced by $\mathcal{L}_+(X)$, and (c) follows from Lemma 29 with $\mathcal{L}(X)$ replaced by $\mathcal{L}_-(X)$.

An intriguing observation worth mentioning is that the losses $\mathcal{L}_+(X)$ and $\mathcal{L}_-(X)$ applied to decontaminate the unlabeled data in SU and DU learning ((95) and (100)) are now used to decontaminate the similar and the dissimilar data in SD learning, respectively. One can also quickly draw the same conclusion from Table 4. Knowing the reason behind this observation would help to transfer one corrected loss developed in one scenario to another weakly supervised scenario.

### 5.1.6 Similarity-Confidence (Sconf) Learning

Since $M_{\text{Sconf}}$ (55) is not a child of $M_{\text{UU}}$ (49) on the reduction graph, a direct application of Theorem 21 is infeasible. Nevertheless, we demonstrate how our framework is applied to rewrite the classification risk for Sconf learning. We make a small adjustment to the framework that instead of showing $\bar{L}^\top \bar{P} = L^\top M^\dagger \bar{P} = L^\top P$, we show that for loss vector $\bar{L}$ with a certain property,

$$\int_{x' \in \mathcal{X}} \bar{L}^\top \bar{P} \, \mathrm{d}x' = \bar{L}^\top \tilde{M}_{\text{Sconf}} P. \tag{102}$$

The idea behind this approach is to accommodate $x'$ sampled from $\mathcal{P}_{X'}$ (18). Suppose, informally, we have the equation above. Then, the right-hand side of (102) will produce $L^\top P$ if we can compute a decontamination matrix $\tilde{M}_{\text{Sconf}}^\dagger$ satisfying $\tilde{M}_{\text{Sconf}}^\dagger \tilde{M}_{\text{Sconf}} = I$ and assign $\bar{L}^\top := L^\top \tilde{M}_{\text{Sconf}}^\dagger$. Lastly, integrating over $x$ on both sides, we obtain the key equation

$$\int_{x \in \mathcal{X}} \int_{x' \in \mathcal{X}} \bar{L}^\top \bar{P} \, \mathrm{d}x' \mathrm{d}x = \int_{x \in \mathcal{X}} L^\top P \, \mathrm{d}x$$

for risk rewrite.

**Step 1: Corrected Loss Design and Risk Rewrite.**
Let us follow the notations in Section 4.1.6. We begin with two technical lemmas and leave their proofs to Appendix D.1. The first technical lemma shows how to achieve (102).

**Lemma 32.** *Assume the formulation $\bar{P} = M_{\text{Sconf}} B$ (58) is given. Suppose a vector of corrected losses $\bar{L}^\top$ of the form $\left( \tilde{\ell}_1(x) \quad \tilde{\ell}_2(x) \right)$ is independent of $x'$. Then, we have*

$$\int_{x' \in \mathcal{X}} \bar{L}^\top \bar{P} \, \mathrm{d}x' = \bar{L}^\top \tilde{M}_{\text{Sconf}} P, \tag{103}$$

*where*

$$\tilde{M}_{\text{Sconf}} = \begin{pmatrix} \int_{x'} \frac{\pi_{\text{p}}^2 \mathcal{P}_{x'|\text{p}} - \pi_{\text{n}}^2 \mathcal{P}_{x'|\text{n}}}{r - \pi_{\text{n}}} \mathrm{d}x' & \int_{x'} \frac{\pi_{\text{n}}^2 \mathcal{P}_{x'|\text{n}} - \pi_{\text{n}}^2 \mathcal{P}_{x'|\text{p}}}{r - \pi_{\text{n}}} \mathrm{d}x' \\ \int_{x'} \frac{\pi_{\text{p}}^2 \mathcal{P}_{x'|\text{n}} - \pi_{\text{p}}^2 \mathcal{P}_{x'|\text{p}}}{\pi_{\text{p}} - r} \mathrm{d}x' & \int_{x'} \frac{\pi_{\text{p}}^2 \mathcal{P}_{x'|\text{p}} - \pi_{\text{n}}^2 \mathcal{P}_{x'|\text{n}}}{\pi_{\text{p}} - r} \mathrm{d}x' \end{pmatrix}.$$

The second technical lemma computes the decontamination matrix.

**Lemma 33.** *Let*

$$\tilde{M}_{\text{Sconf}}^\dagger := \begin{pmatrix} \frac{r - \pi_{\text{n}}}{\pi_{\text{p}} - \pi_{\text{n}}} & 0 \\ 0 & \frac{\pi_{\text{p}} - r}{\pi_{\text{p}} - \pi_{\text{n}}} \end{pmatrix}.$$

*Then,*

$$\tilde{M}_{\text{Sconf}}^\dagger \tilde{M}_{\text{Sconf}} = I.$$

Next, we follow the sketch above to instantiate the corrected losses as

$$\bar{L}^\top := L^\top \tilde{M}_{\text{Sconf}}^\dagger = \left( \frac{r - \pi_{\text{n}}}{\pi_{\text{p}} - \pi_{\text{n}}} \ell_{\text{p}}(X) \quad \frac{\pi_{\text{p}} - r}{\pi_{\text{p}} - \pi_{\text{n}}} \ell_{\text{n}}(X) \right).$$

Putting $\tilde{M}_{\text{Sconf}}^\dagger$, $\bar{L}$, and (103) together, we have the following rewrite.

**Theorem 34.** *Assume $\pi_{\text{p}} \neq 1/2$. The classification risk of Sconf learning can be expressed by*

$$R(g) = \mathbb{E}_{X, X'} \left[ \frac{r - \pi_{\text{n}}}{\pi_{\text{p}} - \pi_{\text{n}}} \ell_{\text{p}}(X) + \frac{\pi_{\text{p}} - r}{\pi_{\text{p}} - \pi_{\text{n}}} \ell_{\text{n}}(X) \right]. \tag{104}$$

*Proof.* Integrating both sides of (103) over $x$ and applying Lemma 33, we obtain

$$\int_{x\in\mathcal{X}}\int_{x'\in\mathcal{X}}\bar{L}^\top\bar{P}\,\mathrm{d}x'\mathrm{d}x = \int_{x\in\mathcal{X}}\bar{L}^\top\tilde{M}_{\mathrm{Sconf}}P\,\mathrm{d}x$$
$$= \int_{x\in\mathcal{X}}L^\top\tilde{M}_{\mathrm{Sconf}}^{-1}\tilde{M}_{\mathrm{Sconf}}P\,\mathrm{d}x = R(g).$$

On the other hand, substituting $\bar{L}$ with $\left(\frac{r-\pi_{\mathrm{n}}}{\pi_{\mathrm{p}}-\pi_{\mathrm{n}}}\ell_{\mathrm{p}}(X)\quad\frac{\pi_{\mathrm{p}}-r}{\pi_{\mathrm{p}}-\pi_{\mathrm{n}}}\ell_{\mathrm{n}}(X)\right)$ and $\bar{P}$ with $\begin{pmatrix}\mathcal{P}_X\mathcal{P}_{X'}\\\mathcal{P}_X\mathcal{P}_{X'}\end{pmatrix}$,

$$\int_{x\in\mathcal{X}}\int_{x'\in\mathcal{X}}\bar{L}^\top\bar{P}\,\mathrm{d}x'\mathrm{d}x = \int_{x\in\mathcal{X}}\int_{x'\in\mathcal{X}}\mathcal{P}_x\mathcal{P}_{x'}\left(\frac{r-\pi_{\mathrm{n}}}{\pi_{\mathrm{p}}-\pi_{\mathrm{n}}}\ell_{\mathrm{p}}(x)+\frac{\pi_{\mathrm{p}}-r}{\pi_{\mathrm{p}}-\pi_{\mathrm{n}}}\ell_{\mathrm{n}}(x)\right)\,\mathrm{d}x'\mathrm{d}x$$
$$= \mathbb{E}_{X,X'}\left[\frac{r-\pi_{\mathrm{n}}}{\pi_{\mathrm{p}}-\pi_{\mathrm{n}}}\ell_{\mathrm{p}}(X)+\frac{\pi_{\mathrm{p}}-r}{\pi_{\mathrm{p}}-\pi_{\mathrm{n}}}\ell_{\mathrm{n}}(X)\right]$$

completes the proof of the theorem. □

**Step 2: Recovering the previous result(s).**
From the above derivation, we have achieved the first half of the rewrite in (19). Notice that (56) can be rephrased as

$$\left(\frac{r-\pi_{\mathrm{n}}}{\pi_{\mathrm{p}}}\right)\mathcal{P}_X\mathcal{P}_{X'} = \left(\pi_{\mathrm{p}}^2\mathcal{P}_{X|\mathrm{p}}-\pi_{\mathrm{n}}^2\mathcal{P}_{X|\mathrm{n}}\right)\mathcal{P}_{X'|\mathrm{p}}+\left(\pi_{\mathrm{n}}^2\mathcal{P}_{X|\mathrm{n}}-\pi_{\mathrm{n}}^2\mathcal{P}_{X|\mathrm{p}}\right)\mathcal{P}_{X'|\mathrm{n}}$$

and that (57) can be rephrased as

$$\left(\frac{\pi_{\mathrm{p}}-r}{\pi_{\mathrm{n}}}\right)\mathcal{P}_X\mathcal{P}_{X'} = \left(\pi_{\mathrm{p}}^2\mathcal{P}_{X|\mathrm{n}}-\pi_{\mathrm{p}}^2\mathcal{P}_{X|\mathrm{p}}\right)\mathcal{P}_{X'|\mathrm{p}}+\left(\pi_{\mathrm{p}}^2\mathcal{P}_{X|\mathrm{p}}-\pi_{\mathrm{n}}^2\mathcal{P}_{X|\mathrm{n}}\right)\mathcal{P}_{X'|\mathrm{n}}.$$

Thus, when $\pi_{\mathrm{p}}\neq 1/2$, we can repeat the proof steps in Lemma 9 to rephrase (58) as

$$\begin{pmatrix}\mathcal{P}_X\mathcal{P}_{X'}\\\mathcal{P}_X\mathcal{P}_{X'}\end{pmatrix} = \begin{pmatrix}\frac{\pi_{\mathrm{p}}\left(\pi_{\mathrm{p}}^2\mathcal{P}_{X|\mathrm{p}}-\pi_{\mathrm{n}}^2\mathcal{P}_{X|\mathrm{n}}\right)}{r-\pi_{\mathrm{n}}} & \frac{\pi_{\mathrm{p}}\left(\pi_{\mathrm{n}}^2\mathcal{P}_{X|\mathrm{n}}-\pi_{\mathrm{n}}^2\mathcal{P}_{X|\mathrm{p}}\right)}{r-\pi_{\mathrm{n}}}\\\frac{\pi_{\mathrm{n}}\left(\pi_{\mathrm{p}}^2\mathcal{P}_{X|\mathrm{n}}-\pi_{\mathrm{p}}^2\mathcal{P}_{X|\mathrm{p}}\right)}{\pi_{\mathrm{p}}-r} & \frac{\pi_{\mathrm{n}}\left(\pi_{\mathrm{p}}^2\mathcal{P}_{X|\mathrm{p}}-\pi_{\mathrm{n}}^2\mathcal{P}_{X|\mathrm{n}}\right)}{\pi_{\mathrm{p}}-r}\end{pmatrix}\begin{pmatrix}\mathcal{P}_{X'|\mathrm{p}}\\\mathcal{P}_{X'|\mathrm{n}}\end{pmatrix}.$$

Comparing the equation above with $\bar{P}=M_{\mathrm{Sconf}}B$, we see that it is still feasible to formulate $\bar{P}$ with $X$ and $X'$ in $M_{\mathrm{Sconf}}$ and $B$ of (58) swapped. Then, repeating the same argument in **Step 1** with $x$ and $x'$ swapped, we obtain

$$R(g) = \mathbb{E}_{X',X}\left[\frac{r-\pi_{\mathrm{n}}}{\pi_{\mathrm{p}}-\pi_{\mathrm{n}}}\ell_{\mathrm{p}}(X')+\frac{\pi_{\mathrm{p}}-r}{\pi_{\mathrm{p}}-\pi_{\mathrm{n}}}\ell_{\mathrm{n}}(X')\right]. \tag{105}$$

Therefore, the following combines (104) and (105) to obtain

$$R(g) = \frac{1}{2}(R(g)+R(g))$$
$$= \frac{1}{2}\mathbb{E}_{X,X'}\left[\frac{r-\pi_{\mathrm{n}}}{\pi_{\mathrm{p}}-\pi_{\mathrm{n}}}\ell_{\mathrm{p}}(X)+\frac{\pi_{\mathrm{p}}-r}{\pi_{\mathrm{p}}-\pi_{\mathrm{n}}}\ell_{\mathrm{n}}(X)\right]+\frac{1}{2}\mathbb{E}_{X,X'}\left[\frac{r-\pi_{\mathrm{n}}}{\pi_{\mathrm{p}}-\pi_{\mathrm{n}}}\ell_{\mathrm{p}}(X')+\frac{\pi_{\mathrm{p}}-r}{\pi_{\mathrm{p}}-\pi_{\mathrm{n}}}\ell_{\mathrm{n}}(X')\right]$$
$$= \mathbb{E}_{X,X'}\left[\frac{r-\pi_{\mathrm{n}}}{\pi_{\mathrm{p}}-\pi_{\mathrm{n}}}\frac{\ell_{\mathrm{p}}(X)+\ell_{\mathrm{p}}(X')}{2}+\frac{\pi_{\mathrm{p}}-r}{\pi_{\mathrm{p}}-\pi_{\mathrm{n}}}\frac{\ell_{\mathrm{n}}(X)+\ell_{\mathrm{n}}(X')}{2}\right]$$

that recovers rewrite (19) in Section 2.2.8. By matching notations, we recover Theorem 3 of Cao et al. (2021b) [10].

---

[10]The matching to the notations of Cao et al. (2021b) is as follows: $\pi_{\mathrm{p}}$ is $\pi_+$, $\pi_{\mathrm{n}}$ is $\pi_-$, $r$ is $s$, $\ell_{\mathrm{p}}(X)$ is $\ell(g(X),+1)$, and $\ell_{\mathrm{n}}(X)$ is $\ell(g(X),-1)$.

### 5.2 CCN Scenarios

The proposed framework is now applied to conduct the risk rewrites for WSLs discussed in Section 4.2 and summarized in Table 8. Counterintuitively, we demonstrate that finding an inverse matrix (e.g., Proposition 1) is not the only way to solve the risk rewrite problem. Introduced in Proposition 2, the new technique exploited in this subsection, marginal chain, calculates the decontamination matrix for (37) via applying the conditional probability formula twice during a chain of matrix multiplications.

#### 5.2.1 Generalized CCN

We justify the proposed framework for generalized CCN learning via the following steps. Derived equations will be applied to solve the risk rewrite problem for WSLs discussed in Section 4.2.

**Step 1: Corrected Loss Design.**
Let us follow the notations in Section 4.2.1. Same as what we have illustrated in the beginning of Section 5.1.1, the proposed framework achieves three milestones to rewrite the risk. We apply Lemma 10 to achieve the first milestone, $\bar{P} = M_{\mathrm{gCCN}} P$. This is done by noting that for generalized CCN, $\bar{P} = M_{\mathrm{gCCN}} B$ and $B = P$ are given by Lemma 10 (i.e., do not need to handle $M_{\mathrm{trsf}}$ discussed in Section 3.2 since $M_{\mathrm{trsf}}$ is the identity matrix when $B = P$).

The second milestone is to find $M_{\mathrm{gCCN}}^{\dagger}$ to achieve $M_{\mathrm{gCCN}}^{\dagger} \bar{P} = P$. Since $M_{\mathrm{gCCN}}$ (64) is identical to $M$ (40), a direct application of Proposition 2 gives the decontamination matrix

$$
M_{\mathrm{gCCN}}^{\dagger} := \begin{pmatrix}
\mathcal{P}_{Y=1|S=s_1,X} & \mathcal{P}_{Y=1|S=s_2,X} & \cdots & \mathcal{P}_{Y=1|S=s_{|\mathcal{S}|},X} \\
\mathcal{P}_{Y=2|S=s_1,X} & \mathcal{P}_{Y=2|S=s_2,X} & \cdots & \mathcal{P}_{Y=2|S=s_{|\mathcal{S}|},X} \\
\vdots & \vdots & \ddots & \vdots \\
\mathcal{P}_{Y=K|S=s_1,X} & \mathcal{P}_{Y=K|S=s_2,X} & \cdots & \mathcal{P}_{Y=K|S=s_{|\mathcal{S}|},X}
\end{pmatrix}
\tag{106}
$$

that satisfies the $M_{\mathrm{gCCN}}^{\dagger} \bar{P} = P$ requirement.

The final milestone is achieved by instantiating the corrected loss (38) as $\bar{L}^{\top} := L^{\top} M_{\mathrm{gCCN}}^{\dagger}$. We denote the $k$-th entry of $L$ is $\ell_{Y=k}$ with $k \in [K]$ and the $j$-th entry of $\bar{L}$ is $\bar{\ell}_{S=s_j}$ with $j \in [|\mathcal{S}|]$.

Despite Proposition 2's simplicity, the construction of $M_{\mathrm{gCCN}}^{\dagger}$ is somewhat surprising. $M_{\mathrm{gCCN}}^{\dagger}$, to our best knowledge, contributes to a first loss correction result relaxing the invertibility constraint. Unlike $M_{\mathrm{UU}}^{\dagger}$ (Corollary 20), which needs to compute an inverse matrix, one can construct $M_{\mathrm{gCCN}}^{\dagger}$ by calculating each entry $\mathcal{P}_{Y|S,X}$ in (106), to which, we point out a systematic way in Section 5.2.2.

**Step 2: Classification Risk Rewrite.**
With $\bar{L}$ in hand, the following theorem provides an intermediate form of risk rewrite.

**Theorem 35.** *Let $\bar{P}$ and $P$ are given by (62) and (63), respectively. Denote $\bar{L}^{\top} := L M_{\mathrm{gCCN}}^{\dagger}$. Then, $\bar{L}^{\top} \bar{P} = L^{\top} P$ and*

$$
R(g) = \int_{\mathcal{X}} L^{\top} P \mathrm{d}x = \int_{\mathcal{X}} \bar{L}^{\top} \bar{P} \mathrm{d}x.
\tag{107}
$$

*Proof.* Since $M_{\mathrm{gCCN}}^{\dagger}$ is given by Proposition 2, $M_{\mathrm{gCCN}}^{\dagger} \bar{P} = P$. Thus, following the framework (39), we have $\bar{L}^{\top} \bar{P} = L^{\top} M_{\mathrm{gCCN}}^{\dagger} \bar{P} = L^{\top} P$ implying (107). □

Theorem 35 will be applied to derive the respective rewrites for WSLs discussed in Section 4.2 in the rest of this subsection. In particular, we explain how to realize $M_{\mathrm{gCCN}}^{\dagger}$ (106) for a given CCN scenario. Then, the risk rewrite (107) automatically carries over for the scenario considered, and the respective $\bar{L}$ specifies the corrected losses in the rewrite.

### 5.2.2 Proper Partial-Label (PPL) Learning

$M_{\text{gCCN}}^{\dagger}$ provides an abstraction for us to construct the corrected losses $\bar{L}$. Next, we focus on deriving the actual form of $\mathcal{P}_{Y|S,X}$ in $M_{\text{gCCN}}^{\dagger}$ (106) to explicitly express $\bar{\ell}_S$ for PPL.

**Step 1: Corrected Loss Design and Risk Rewrite.**
Let us follow the notations in Section 4.2.2. The following lemma specifies the form of $\mathcal{P}_{Y|S,X}$ to instantiate $M_{\text{gCCN}}^{\dagger}$.

**Lemma 36.** $M_{\text{PPL}}^{\dagger}$ *corresponds to realizing* $M_{\text{gCCN}}^{\dagger}$ *(106) with*

$$\mathcal{P}_{Y=i|S=s_j,X} := \frac{\mathcal{P}_{Y=i|X}\mathbb{I}\left[Y=i\in s_j\right]}{\sum_{a\in s_j}\mathcal{P}_{Y=a|X}}. \tag{108}$$

*Proof.* Recall that the decontamination matrix of $M_{\text{gCCN}}$ (64) is $M_{\text{gCCN}}^{\dagger}$ (106) and $M_{\text{PPL}}$ is a reduction of $M_{\text{gCCN}}$ via $\mathcal{P}_{S|Y,X} = C(S,X)\mathbb{I}\left[Y\in S\right]$ (65). Thus, to find out the $(i,j)$ entry of $M_{\text{PPL}}^{\dagger}$, we need to find out the form of $\mathcal{P}_{Y=i|S=s_j,X}$ subject to (65).

Applying Theorem 1 of Wu et al. (2023) directly gives

$$\mathcal{P}_{Y=i|S=s_j,X} = \frac{\mathcal{P}_{Y=i|X}\mathbb{I}\left[Y=i\in s_j\right]}{\sum_{a\in s_j}\mathcal{P}_{Y=a|X}},$$

which completes the proof. For completeness, we provide a derivation as follows.

Note that $\mathcal{P}_{S|Y,X} = C(S,X)\mathbb{I}\left[Y\in S\right]$ (65) implies

$$
\begin{aligned}
\sum_{b\in\mathcal{Y}\setminus S}\mathcal{P}_{S,Y=b|X} &= \sum_{b\in\mathcal{Y}\setminus S}\mathcal{P}_{S|Y=b,X}\mathcal{P}_{Y=b|X} \\
&= \sum_{b\in\mathcal{Y}\setminus S}C(S,X)\mathbb{I}\left[b\in S\right]\mathcal{P}_{Y=b|X} \\
&= 0.
\end{aligned}
$$

Therefore, $\mathcal{P}_{S|X} = \sum_{a\in S}\mathcal{P}_{S,Y=a|X} + \sum_{b\in\mathcal{Y}\setminus S}\mathcal{P}_{S,Y=b|X} = \sum_{a\in S}\mathcal{P}_{S,Y=a|X}$. Utilizing this fact, we obtain

$$
\begin{aligned}
\mathcal{P}_{Y|S,X} = \frac{\mathcal{P}_{S,Y|X}}{\mathcal{P}_{S|X}} &= \frac{\mathcal{P}_{S|Y,X}\mathcal{P}_{Y|X}}{\sum_{a\in S}\mathcal{P}_{S|Y=a,X}\mathcal{P}_{Y=a|X}} \\
&= \frac{C(S,X)\mathbb{I}\left[Y\in S\right]\mathcal{P}_{Y|X}}{\sum_{a\in S}C(S,X)\mathbb{I}\left[Y=a\in S\right]\mathcal{P}_{Y=a|X}} \\
&= \frac{\mathcal{P}_{Y|X}\mathbb{I}\left[Y\in S\right]}{\sum_{a\in S}\mathcal{P}_{Y=a|X}}
\end{aligned}
$$

that finishes the proof for Theorem 1 of Wu et al. (2023). □

We have shown that $M_{\text{PPL}}^{\dagger}$ is derived from $M_{\text{gCCN}}^{\dagger}$. Thus, we can follow Theorem 35 to construct the corrected losses using (108) and obtain the risk rewrite (27) for PPL in Section 2.2.12.

**Corollary 37.** *Given $M_{\text{PPL}}^{\dagger}$ defined by (108), we denote the corrected losses $\bar{L}^{\top} := L^{\top}M_{\text{PPL}}^{\dagger}$. Then, for PPL learning, the classification risk can be rewritten as*

$$R(g) = \mathbb{E}_{S,X}\left[\bar{\ell}_S\right],$$

*where*

$$\bar{\ell}_S = \sum_{i\in S}\frac{\mathcal{P}_{Y=i|X}}{\sum_{a\in S}\mathcal{P}_{Y=a|X}}\ell_{Y=i}. \tag{109}$$

*Proof.* Given (108), the $j$-th entry of $\bar{L}^\top$ is of the form

$$\bar{\ell}_{S=s_j} = \left(L^\top M_{\text{PPL}}^\dagger\right)_j = \sum_{i=1}^K \frac{\mathcal{P}_{Y=i|X}\mathbb{I}\left[Y=i\in s_j\right]}{\sum_{a\in s_j}\mathcal{P}_{Y=a|X}}\ell_{Y=i}$$

$$= \sum_{i\in s_j}\frac{\mathcal{P}_{Y=i|X}}{\sum_{a\in s_j}\mathcal{P}_{Y=a|X}}\ell_{Y=i}.$$

Then, since $M_{\text{PPL}}^\dagger$ is a realization of $M_{\text{gCCN}}^\dagger$ according to Lemma 36, we continue (107) to express the risk as

$$R(g) = \int_{x\in\mathcal{X}}\bar{L}^\top\bar{P}\mathrm{d}x = \int_{x\in\mathcal{X}}\sum_{j=1}^{|\mathcal{S}|}\mathcal{P}_{S=s_j,x}\bar{\ell}_{S=s_j}\mathrm{d}x = \mathbb{E}_{S,X}\left[\bar{\ell}_S\right].$$

$\square$

**Step 2: Recovering the previous result(s).**
We finish this part by pointing out Corollary 37 recovers Theorem 3 of Wu et al. (2023).

### 5.2.3 Provably Consistent Partial-Label (PCPL) Learning

It is fairly straightforward to apply the proposed framework to rewrite the classification risk. However, it is more involved in recovering the existing result.

**Step 1: Corrected Loss Design and Risk Rewrite.**
The argument for obtaining the risk rewrite for PCPL is similar to that of PPL. From Section 4.2.3 we know that PCPL is a special case of PPL that only differs in the choice of $C(S,X)$. Since $C(S,X)$ is independent of (108), $M_{\text{PCPL}}^\dagger$ and $M_{\text{PPL}}^\dagger$ are identical. Hence, following the notations in Section 4.2.3 and repeating the proof of Corollary 37, we obtain the risk rewrite for PCPL:

**Corollary 38.** *The decontamination matrix $M_{\text{PCPL}}^\dagger$ for PCPL equals $M_{\text{PPL}}^\dagger$. If we define the corrected losses as $\bar{L}^\top := L^\top M_{\text{PCPL}}^\dagger$, the classification risk for PCPL learning can be rewritten as*

$$R(g) = \mathbb{E}_{S,X}\left[\bar{\ell}_S\right],$$

*where*

$$\bar{\ell}_S = \sum_{i\in S}\frac{\mathcal{P}_{Y=i|X}}{\sum_{a\in S}\mathcal{P}_{Y=a|X}}\ell_{Y=i}. \tag{110}$$

**Step 2: Recovering the previous result(s).**
In order to recover (8) of Feng et al. (2020b), we need to reorganize the sum in (110) by leveraging a unique property of a pair of partial-labels $(s,s')$ that complement each other. The following technical lemma states the required property, with proof deferred to Appendix D.2.

**Lemma 39.** *Let $(s,s')$ be a pair of partial-labels satisfying $s=\mathcal{Y}\backslash s'$. Then,*

$$\mathcal{P}_{S=s,X}\bar{\ell}_{S=s} + \mathcal{P}_{S=s',X}\bar{\ell}_{S=s'} = \mathcal{P}_{S=s,X}\sum_{i=1}^K\frac{\mathcal{P}_{Y=i|X}\ell_{Y=i}}{\sum_{a\in s}\mathcal{P}_{Y=a|X}}.$$

Denote $s_j' := \mathcal{Y}\backslash s_j$ for every $s_j\in\mathcal{S}$. Then, Lemma 39 implies

$$\sum_{j=1}^{|\mathcal{S}|}2\mathcal{P}_{S=s_j,X}\bar{\ell}_{S=s_j} = \sum_{j=1}^{|\mathcal{S}|}\left(\mathcal{P}_{S=s_j,X}\bar{\ell}_{S=s_j} + \mathcal{P}_{S=s_j',X}\bar{\ell}_{S=s_j'}\right)$$

$$= \sum_{j=1}^{|\mathcal{S}|}\mathcal{P}_{S=s_j,X}\sum_{i=1}^K\frac{\mathcal{P}_{Y=i|X}\ell_{Y=i}}{\sum_{a\in s_j}\mathcal{P}_{Y=a|X}}.$$

Hence, continuing from Corollary 38,

$$
\begin{aligned}
\mathbb{E}_{S,X}\left[\bar{\ell}_S\right] &= \int_{x\in\mathcal{X}} \sum_{j=1}^{|\mathcal{S}|} \mathcal{P}_{S=s_j,x}\bar{\ell}_{S=s_j}\mathrm{d}x \\
&= \frac{1}{2}\int_{x\in\mathcal{X}} \sum_{j=1}^{|\mathcal{S}|} \mathcal{P}_{S=s_j,x} \sum_{i=1}^{K} \frac{\mathcal{P}_{Y=i|x}\ell_{Y=i}}{\sum_{a\in s_j}\mathcal{P}_{Y=a|x}}\mathrm{d}x \\
&= \frac{1}{2}\mathbb{E}_{S,X}\left[\sum_{i=1}^{K} \frac{\mathcal{P}_{Y=i|X}}{\sum_{a\in S}\mathcal{P}_{Y=a|X}}\ell_{Y=i}\right]
\end{aligned}
$$

shows that the rewrite from the framework recovers (25) in Section 2.2.11. By matching notations, we also recover (8) of Feng et al. (2020b)[11].

### 5.2.4 Multi-Complementary-Label (MCL) Learning

**Step 1: Corrected Loss Design and Risk Rewrite.**
Let us follow the notations in Section 4.2.4. As discussed in Section 4.2.4, MCL is a special case of PPL. Thus, we can modify Lemma 36 based on the notations in Section 4.2.4 to construct the decontamination matrix $M_{\mathrm{MCL}}^{\dagger}$ for MCL. Then, following the same steps for proving Corollary 37, we instantiate $\bar{L}$ to conduct the risk rewrite for MCL:

**Corollary 40.** *The $(i,j)$ entry of the decontamination matrix $M_{\mathrm{MCL}}^{\dagger}$ is of the form*

$$
\mathcal{P}_{Y=i|\bar{S}=\bar{s}_j,X} = \frac{\mathcal{P}_{Y=i|X}\mathbb{I}\left[Y=i\notin\bar{s}_j\right]}{\sum_{a\notin\bar{s}_j}\mathcal{P}_{Y=a|X}}. \tag{111}
$$

*Define the corrected losses $\bar{L}^{\top} := L^{\top}M_{\mathrm{MCL}}^{\dagger}$. Then, for MCL learning, the classification risk can be rewritten as*

$$
R(g) = \mathbb{E}_{\bar{S},X}\left[\bar{\ell}_{\bar{S}}\right],
$$

*where*

$$
\bar{\ell}_{\bar{S}} = \sum_{i\notin\bar{S}} \frac{\mathcal{P}_{Y=i|X}}{\sum_{a\notin\bar{S}}\mathcal{P}_{Y=a|X}}\ell_{Y=i}. \tag{112}
$$

**Step 2: Recovering the previous result(s).**
Although legitimate, the risk rewrite (112) following the marginal chain approach appears different from Theorem 3 of Feng et al. (2020a), to which we resort to the inversion approach (Proposition 1) that finds another decontamination matrix, termed $M_{\mathrm{MCL}}^{-1}$, to recover. As a preparation step, we denote $N_d$ as the number of multi-complementary-labels with size $d$ and group rows of $M_{\mathrm{MCL}}$ (71) by the size of labels as follows.

$$
M_{\mathrm{MCL}} = \begin{pmatrix} \mathcal{P}_{|\bar{S}|=1}M_1 \\ \mathcal{P}_{|\bar{S}|=2}M_2 \\ \vdots \\ \mathcal{P}_{|\bar{S}|=K-1}M_{K-1} \end{pmatrix}, \tag{113}
$$

---

[11]The matching to the notations of Feng et al. (2020b) is as follows: $\mathcal{P}_{S,X}$ is $\tilde{p}(x,Y)$, $\mathcal{P}_{Y=i|X}$ is $p(y=i|x)$, and $\ell_{Y=i}$ is $\mathcal{L}(f(x),i)$.

where for $d \in [K-1]$, each block is of the form[12]

$$M_d = \frac{1}{\binom{K-1}{d}} \begin{pmatrix} \mathbb{I}[Y = 1 \notin \bar{s}_{d,1}] & \mathbb{I}[Y = 2 \notin \bar{s}_{d,1}] & \cdots & \mathbb{I}[Y = K \notin \bar{s}_{d,1}] \\ \mathbb{I}[Y = 1 \notin \bar{s}_{d,2}] & \mathbb{I}[Y = 2 \notin \bar{s}_{d,2}] & \cdots & \mathbb{I}[Y = K \notin \bar{s}_{d,2}] \\ \vdots & \vdots & \ddots & \vdots \\ \mathbb{I}[Y = 1 \notin \bar{s}_{d,N_d}] & \mathbb{I}[Y = 2 \notin \bar{s}_{d,N_d}] & \cdots & \mathbb{I}[Y = K \notin \bar{s}_{d,N_d}] \end{pmatrix}. \tag{114}$$

To maintain the equality $\bar{P} = M_{\mathrm{MCL}}P$ established in Lemma 13, we also rearrange $\bar{P}$ (69) as

$$\left( \mathcal{P}_{\bar{S}=\bar{s}_{1,1},X} \quad \cdots \mathcal{P}_{\bar{S}=\bar{s}_{1,N_1},X} \quad \cdots \mathcal{P}_{\bar{S}=\bar{s}_{K-1,1},X} \quad \cdots \mathcal{P}_{\bar{S}=\bar{s}_{K-1,N_{K-1}},X} \right)^{\top}. \tag{115}$$

As a sanity check, we see that for any $d' \in [K-1]$ and $j' \in [N_d]$,

$$\begin{aligned} \left( \mathcal{P}_{|\bar{S}|=d'} M_{d'} P \right)_{j'} &= \mathcal{P}_{|\bar{S}|=d'} \cdot \frac{1}{\binom{K-1}{d'}} \sum_Y \mathbb{I}[Y \notin \bar{s}_{d',j'}] \mathcal{P}_{Y,X} \\ &= \sum_{d=1}^{K-1} \mathcal{P}_{|\bar{s}_{d',j'}|=d} \cdot \frac{1}{\binom{K-1}{d'}} \sum_{Y \notin \bar{s}_{d',j'}} \mathcal{P}_{Y,X} \mathbb{I}[|\bar{s}_{d',j'}| = d] \\ &= \mathcal{P}_{\bar{S}=\bar{s}_{d',j'},X}. \end{aligned} \tag{116}$$

The next lemma is crucial for us to devise the decontamination matrix $M_{\mathrm{MCL}}^{-1}$ via the inversion approach. We defer its proof to the later part of this sub-subsection.

**Lemma 41.** *Let $i^\star \in \mathcal{Y}$ be fixed. Then, for every $d \in [K-1]$,*

$$\mathcal{P}_{Y=i^\star,X} = \sum_{j=1}^{N_d} \left( 1 - \frac{K-1}{d} \mathbb{I}[Y = i^\star \in \bar{S} = \bar{s}_{d,j}] \right) \mathcal{P}_{\bar{S}=\bar{s}_{d,j},X||\bar{S}|=d}.$$

*Moreover, the inverse matrix $M_d^{-1}$ of $M_d$ (114) is of the form*

$$\begin{pmatrix} 1 - \frac{K-1}{d}\mathbb{I}[Y = 1 \in \bar{s}_{d,1}] & 1 - \frac{K-1}{d}\mathbb{I}[Y = 1 \in \bar{s}_{d,2}] & \cdots & 1 - \frac{K-1}{d}\mathbb{I}[Y = 1 \in \bar{s}_{d,N_d}] \\ 1 - \frac{K-1}{d}\mathbb{I}[Y = 2 \in \bar{s}_{d,1}] & 1 - \frac{K-1}{d}\mathbb{I}[Y = 2 \in \bar{s}_{d,2}] & \cdots & 1 - \frac{K-1}{d}\mathbb{I}[Y = 2 \in \bar{s}_{d,N_d}] \\ \vdots & \vdots & \ddots & \vdots \\ 1 - \frac{K-1}{d}\mathbb{I}[Y = K \in \bar{s}_{d,1}] & 1 - \frac{K-1}{d}\mathbb{I}[Y = K \in \bar{s}_{d,2}] & \cdots & 1 - \frac{K-1}{d}\mathbb{I}[Y = K \in \bar{s}_{d,N_d}] \end{pmatrix}. \tag{117}$$

Applying the lemma, we construct

$$M_{\mathrm{MCL}}^{-1} := \begin{pmatrix} M_1^{-1} & M_2^{-1} & \cdots & M_{K-1}^{-1} \end{pmatrix} \tag{118}$$

and obtain $M_{\mathrm{MCL}}^{-1}\bar{P} = P$ since $\bar{P} = M_{\mathrm{MCL}}P$ (116) and

$$M_{\mathrm{MCL}}^{-1} M_{\mathrm{MCL}} = \sum_{d=1}^{K-1} M_d^{-1} \mathcal{P}_{|\bar{S}|=d} M_d = \sum_{d=1}^{K-1} \mathcal{P}_{|\bar{S}|=d} M_d^{-1} M_d = \sum_{d=1}^{K-1} \mathcal{P}_{|\bar{S}|=d} I = I.$$

We remark that $M_{\mathrm{MCL}}^{-1}$ plays the same role as $M_{\mathrm{MCL}}^{\dagger}$ realized by (111), as they both are decontamination matrices (designed to convert $\bar{P}$ back to $P$ and used to construct the corrected losses $\bar{L}$). Distinct symbols

---

[12]Comparing to (71) where we use one index to denote a total of $|\mathcal{S}|$ partial-labels, $M_d$ uses a pair of indices $d$ and $j$ to denote the $j$-th partial-label with size $d$. It is easy to verify that $\sum_{d=1}^{K-1} N_d = \sum_{d=1}^{K-1} \binom{K-1}{d} = 2^K - 2 = |\mathcal{S}|$.

are merely used to reflect the difference that $M_{\mathrm{MCL}}^{\dagger}$ results from the marginal chain method while $M_{\mathrm{MCL}}^{-1}$ comes from the inversion approach. Then, applying the framework (38), $\bar{L}^{\top} := L^{\top} M_{\mathrm{MCL}}^{-1}$ leads to

$$\bar{L}^{\top} \bar{P} = L^{\top} M_{\mathrm{MCL}}^{-1} \bar{P} = L^{\top} P.$$

With the corrected losses $\bar{L}$ in hand, the following theorem provides the risk rewrite (23) for MCL via the inversion approach and recovers Theorem 3 of Feng et al. (2020a)[13].

**Theorem 42.** *For MCL learning, the classification risk can be expressed as follows.*

$$R(g) = \mathbb{E}_{\bar{S}, X} \left[ \bar{\ell}_{\bar{S}} \right] = \sum_{d=1}^{K-1} \mathcal{P}_{|\bar{S}|=d} \mathbb{E}_{\bar{S}, X \| |\bar{S}|=d} \left[ \bar{\ell}_{\bar{S}} \right],$$

*where*

$$\bar{\ell}_{\bar{S}} = \sum_{i \notin \bar{S}} \ell_{Y=i} - \frac{K - 1 - |\bar{S}|}{|\bar{S}|} \sum_{\bar{s} \in \bar{S}} \ell_{Y=\bar{s}}.$$

*Proof.* We first establish

$$R(g) = \int_{\mathcal{X}} \bar{L}^{\top} \bar{P} \mathrm{d}x = \mathbb{E}_{\bar{S}, X} \left[ \bar{\ell}_{\bar{S}} \right]$$

since $\bar{L}^{\top} \bar{P} = L^{\top} P$, where $\bar{P}$ is specified in (115) and $\bar{L}^{\top} = L^{\top} M_{\mathrm{MCL}}^{-1}$ with the $\bar{S}$-th entry being $\bar{\ell}_{\bar{S}}$. Also, recall that $\mathcal{P}_{\bar{S}, X} = \sum_{d=1}^{K-1} \mathcal{P}_{|\bar{S}|=d} \mathcal{P}_{\bar{S}, X \| |\bar{S}|=d}$ in Section 4.2.4. Thus, decomposing the probability by the size of $\bar{S}$, we have

$$\mathbb{E}_{\bar{S}, X} \left[ \bar{\ell}_{\bar{S}} \right] = \sum_{d=1}^{K-1} \mathcal{P}_{|\bar{S}|=d} \mathbb{E}_{\bar{S}, X \| |\bar{S}|=d} \left[ \bar{\ell}_{\bar{S}} \right].$$

Lastly, $M_{\mathrm{MCL}}^{-1}$ (118) and $M_d^{-1}$ (117) imply, when $\bar{S} = \bar{s}_{d,j}$,

$$\bar{\ell}_{\bar{S}=\bar{s}_{d,j}} = \left( L^{\top} M_d^{-1} \right)_j = \sum_{i=1}^{K} \ell_{Y=i} \left( 1 - \frac{K-1}{d} \mathbb{I}\left[ Y = i \in \bar{s}_{d,j} \right] \right) = \sum_{i=1}^{K} \ell_{Y=i} - \frac{K-1}{d} \sum_{i \in \bar{s}_{d,j}} \ell_{Y=i}.$$

A simple reorganization and substituting $d$ with $|\bar{S}|$ shows

$$\bar{\ell}_{\bar{S}} = \sum_{i \notin \bar{S}} \ell_{Y=i} + \sum_{i \in \bar{S}} \ell_{Y=i} - \frac{K-1}{|\bar{S}|} \sum_{i \in \bar{S}} \ell_{Y=i} = \sum_{i \notin \bar{S}} \ell_{Y=i} - \frac{K - 1 - |\bar{S}|}{|\bar{S}|} \sum_{i \in \bar{S}} \ell_{Y=i}.$$

$\square$

Now we return to the postponed proof.

*Proof.* **of Lemma 41.** We start with identifying $M_d^{-1}$. Denote $\{ \bar{s}_{d,1}, \ldots, \bar{s}_{d,N_d} \}$, the set of multi-complementary-labels of size $d$, as $\bar{\mathcal{S}}_d$. Let us focus on the sized-$d$ data-generating distribution

$$\bar{P}_d = \begin{pmatrix} \mathcal{P}_{\bar{S}=\bar{s}_{d,1}, X \| |\bar{S}|=d} \\ \vdots \\ \mathcal{P}_{\bar{S}=\bar{s}_{d,N_d}, X \| |\bar{S}|=d} \end{pmatrix}.$$

---

[13]The matching to the notations of Feng et al. (2020a) is as follows: $\mathcal{P}_{\bar{S}, X \| |\bar{S}|=d}$ is $\bar{p}(x, \bar{Y} | s = d)$, $\mathcal{P}_{|\bar{S}|=d}$ is $p(s = d)$, and $\bar{\ell}_{\bar{S}}$ is $\bar{\mathcal{L}}_d(f(x), \bar{Y})$.

Note that $\bar{P}_d$ corresponds to extracting the entries from (115) that generate sized-$d$ data and then dividing them by $\mathcal{P}_{|\bar{S}|=d}$. Thus, $\bar{P} = M_{\text{MCL}}P$ in Lemma 13 implies $\bar{P}_d = M_d P$ and its $j$-th entry is expressed as

$$\mathcal{P}_{\bar{S}=\bar{s}_{d,j},X||\bar{S}|=d} = \frac{1}{\binom{K-1}{d}} \sum_{i=1}^{K} \mathbb{I}\left[Y = i \notin \bar{s}_{d,j}\right] \mathcal{P}_{Y=i,X}. \tag{119}$$

The equality hints to us that if one manages to collect certain multi-complementary-labels $\bar{s}'$ to form an equation resembling $\sum_{\bar{s}'} \mathcal{P}_{\bar{s}',X||\bar{s}'|=d} = c_3 \cdot \mathcal{P}_{Y=i,X}$ for some constant $c_3$, then a reciprocal operation $\frac{1}{c_3}$ recovers $\mathcal{P}_{Y=i,X}$ we need (recall we want to find $M_d^{-1}$ achieving $M_d^{-1}\bar{P}_d = P$). To achieve such a goal, we fix on class $i^\star$ and collect elements in $\bar{\mathcal{S}}_d$ that do not contain $i^\star$ to form $\mathcal{E}_d^{i^\star} := \left\{\bar{s}_{d,j}|\bar{s}_{d,j} \in \bar{\mathcal{S}}_d, i^\star \notin \bar{s}_{d,j}\right\}$ to connect $\mathcal{P}_{\bar{S},X||\bar{S}|=d}$ with $\mathcal{P}_{Y=i^\star,X}$ as follows. Summing (119) over all elements in $\mathcal{E}_d^{i^\star}$, we obtain

$$\begin{aligned}
\sum_{\bar{s}\in\mathcal{E}_d^{i^\star}} \mathcal{P}_{\bar{S}=\bar{s},X||\bar{S}|=d} &= \sum_{\bar{s}\in\mathcal{E}_d^{i^\star}} \frac{1}{\binom{K-1}{d}} \sum_{i=1}^{K} \mathbb{I}\left[Y = i \notin \bar{s}\right] \mathcal{P}_{Y=i,X} \\
&= \frac{1}{\binom{K-1}{d}}\left[\binom{K-2}{d} \sum_{\substack{i=1 \\ i\neq i^\star}}^{K} \mathcal{P}_{Y=i,X} + \binom{K-1}{d}\mathcal{P}_{Y=i^\star,X}\right].
\end{aligned}$$

The last equality holds since there are $\binom{K-2}{d}$ multi-complementary-labels $\bar{s} \in \bar{\mathcal{S}}_d$ such that $i \neq i^\star$ and neither of them is in $\bar{s}$ (i.e., $i \notin \bar{s}$ and $i^\star \notin \bar{s}$), and there are $\binom{K-1}{d}$ multi-complementary-labels $\bar{s} \in \bar{\mathcal{S}}_d$ such that $i = i^\star$ and $i$ is not in $\bar{s}$. Then, we regroup the sums by pulling $\binom{K-2}{d}\mathcal{P}_{Y=i^\star,X}$ out of $\binom{K-1}{d}\mathcal{P}_{Y=i^\star,X}$ to combine with $\binom{K-2}{d}\sum_{\substack{i=1 \\ i\neq i^\star}}^{K}\mathcal{P}_{Y=i,X}$. It leads to

$$\begin{aligned}
\sum_{\bar{s}\in\mathcal{E}_d^{i^\star}} \mathcal{P}_{\bar{S}=\bar{s},X||\bar{S}|=d} &= \frac{1}{\binom{K-1}{d}}\left[\binom{K-2}{d} \sum_{i=1}^{K} \mathcal{P}_{Y=i,X} + \binom{K-2}{d-1}\mathcal{P}_{Y=i^\star,X}\right] \\
&= \frac{K-1-d}{K-1}\mathcal{P}_X + \frac{d}{K-1}\mathcal{P}_{Y=i^\star,X}. \tag{120}
\end{aligned}$$

Denoting $\bar{\mathcal{S}}_d \backslash \mathcal{E}_d^{i^\star} = \{\bar{s}_{d,j}|\bar{s}_{d,j} \in \bar{\mathcal{S}}_d, i^\star \in \bar{s}_{d,j}\}$ as $\mathcal{I}_d^{i^\star}$ and rearranging terms in the above equation according to the reciprocal idea illustrated above, we have

$$\begin{aligned}
\mathcal{P}_{Y=i^\star,X} &= \frac{K-1}{d}\left(\sum_{\bar{s}\in\mathcal{E}_d^{i^\star}} \mathcal{P}_{\bar{S}=\bar{s},X||\bar{S}|=d} - \frac{K-1-d}{K-1}\mathcal{P}_X\right) \tag{121} \\
&\overset{(a)}{=} \frac{K-1}{d}\left(\mathcal{P}_X - \sum_{\bar{s}\in\mathcal{I}_d^{i^\star}} \mathcal{P}_{\bar{S}=\bar{s},X||\bar{S}|=d} - \frac{K-1-d}{K-1}\mathcal{P}_X\right) \\
&= \mathcal{P}_X - \frac{K-1}{d}\sum_{\bar{s}\in\mathcal{I}_d^{i^\star}} \mathcal{P}_{\bar{S}=\bar{s},X||\bar{S}|=d}.
\end{aligned}$$

Equality (a) holds since $|\bar{S}|$ and $X$ are independent (Feng et al., 2020a), which implies

$$\mathcal{P}_X = \mathcal{P}_{X||\bar{S}|=d} = \sum_{\bar{s}\in\bar{\mathcal{S}}_d} \mathcal{P}_{\bar{S}=\bar{s},X||\bar{S}|=d} = \sum_{\bar{s}\in\mathcal{E}_d^{i^\star}} \mathcal{P}_{\bar{S}=\bar{s},X||\bar{S}|=d} + \sum_{\bar{s}\in\mathcal{I}_d^{i^\star}} \mathcal{P}_{\bar{S}=\bar{s},X||\bar{S}|=d}.$$

Continuing the derivation, we have

$$\begin{aligned}
\mathcal{P}_{Y=i^\star,X} &= \sum_{j=1}^{N_d} \mathcal{P}_{\bar{S}=\bar{s}_{d,j},X||\bar{S}|=d} - \sum_{j=1}^{N_d} \frac{K-1}{d}\mathbb{I}\left[Y = i^\star \in \bar{s}_{d,j}\right]\mathcal{P}_{\bar{S}=\bar{s}_{d,j},X||\bar{S}|=d} \\
&= \sum_{j=1}^{N_d}\left(1 - \frac{K-1}{d}\mathbb{I}\left[Y = i^\star \in \bar{s}_{d,j}\right]\right)\mathcal{P}_{\bar{S}=\bar{s}_{d,j},X||\bar{S}|=d}, \tag{122}
\end{aligned}$$

proving the first part of the lemma.

The derivation of turning (120) to (121) is a reciprocal action. Thus, if we view $1 - \frac{K-1}{d}\mathbb{I}[Y = i^\star \in \bar{s}_{d,j}]$ as the $(i^\star, j)$ entry of some matrix $M'$, (122) can be interpreted as $P_{i^\star} = \left(M'\bar{P}_d\right)_{i^\star}$, suggesting $M'M_d = I$ since $\bar{P}_d = M_dP$. We formalize this intuition in the next lemma.

**Lemma 43.** *Let $M'$ be of the form (117), and recall $M_d$ is defined by (114). Then, $M'M_d = I$, meaning $M' = M_d^{-1}$.*

The above lemma finishes the proof of Lemma 41. $\qquad\qquad\square$

*Proof.* **of Lemma 43.** Let $d$ be fixed. Denoted by $A_{i,k}$, the $(i,k)$ entry of $M'M_d$, is the inner product of $i$-th row of $M'$ (117) and the $k$-th column of $M_d$ (114)

$$A_{i,k} = \sum_{j=1}^{N_d}\left(1 - \frac{K-1}{d}\mathbb{I}[Y = i \in \bar{s}_{d,j}]\right)\left(\frac{1}{\binom{K-1}{d}}\mathbb{I}[Y = k \notin \bar{s}_{d,j}]\right) = \sum_{j=1}^{N_d}c_{i,k}.$$

In the following, we will show that the calculation results in the identity matrix

$$A_{i,k} = \begin{cases} 1, & \text{if } i = k, \\ 0, & \text{if } i \neq k, \end{cases}$$

to complete the proof.

When $i \neq k$, we have 4 possible cases: (i) Both $i$ and $k$ are in $\bar{s}_{d,j}$, (ii) Both of them are not in $\bar{s}_{d,j}$, (iii) $i \in \bar{s}_{d,j}$ and $k \notin \bar{s}_{d,j}$, and (iv) $i \notin \bar{s}_{d,j}$ and $k \in \bar{s}_{d,j}$. For cases (i) and (iv), the coefficients $c_{i,k}$ are 0 since $\mathbb{I}[k \notin \bar{s}_{d,j}] = 0$ if $k \in \bar{s}_{d,j}$. For case (ii), the coefficient $c_{i,k}$ is $\frac{1}{\binom{K-1}{d}}$. The number of such $\bar{s}_{d,j}$ is $\binom{K-2}{d}$ since we are counting the ways of forming a set of size $d$ from $K-2$ elements. For case (iii), the coefficient $c_{i,k}$ is $\left(1 - \frac{K-1}{d}\right)\frac{1}{\binom{K-1}{d}}$. The number of such $\bar{s}_{d,j}$ is $\binom{K-2}{d-1}$ since we are counting the ways of forming a set of size $d-1$ from $k-2$ elements. Thus, if $i \neq k$,

$$\begin{aligned} A_{i,k} &= \frac{1}{\binom{K-1}{d}}\binom{K-2}{d} + \left(1 - \frac{K-1}{d}\right)\frac{1}{\binom{K-1}{d}}\binom{K-2}{d-1} \\ &= \frac{\binom{K-2}{d}}{\binom{K-1}{d}} + \frac{\binom{K-2}{d-1}}{\binom{K-1}{d}} - \frac{\frac{K-1}{d}\binom{K-2}{d-1}}{\binom{K-1}{d}} = 0 \end{aligned}$$

since

$$\binom{K-2}{d} + \binom{K-2}{d-1} = \binom{K-1}{d} = \frac{K-1}{d}\binom{K-2}{d-1}.$$

When $i = k$, we have 2 possible cases: (i) Both $i$ and $k$ are in $\bar{s}_{d,j}$, (ii) Both are not in $\bar{s}_{d,j}$. For case (i), the coefficient $c_{i,k}$ is 0. For case (ii), the coefficient $c_{i,k}$ is $\frac{1}{\binom{K-1}{d}}$, and the number of such $\bar{s}_{d,j}$ is $\binom{K-1}{d}$, as we want to form a set of size $d$ from $K-1$ candidates. Therefore, if $i = k$,

$$A_{i,k} = \frac{1}{\binom{K-1}{d}}\binom{K-1}{d} = 1.$$

$\qquad\qquad\square$

We want to elaborate more on the role of Theorem 1 of Wu et al. (2023) in the analyses discussed in Section 5.2. Firstly, as shown in the proof of Lemma 41, it aids the execution of the inversion approach (Proposition 1). The properness $C(S, X)\mathbb{I}[Y \in S]$ (65) can be instantiated to define the entries of $M_d$ (114),

which in turn establishes the key equation (120) enabling us to identify the entries of $M_d^{-1}$ (122). Composing $M_d^{-1}$, we obtain $M_{\mathrm{MCL}}^{-1}$, a crucial element for applying our framework (38).

Secondly, Theorem 1 of Wu et al. (2023) contributes to the marginal chain approach (Proposition 2) as well. The key equations (108) and (111) realised from Theorem 1 of Wu et al. (2023) provide the entries of $M_{\mathrm{PPL}}^{\dagger}$ (Lemma 36, Section 5.2.2), $M_{\mathrm{PCPL}}^{\dagger}$ (Section 5.2.3), and $M_{\mathrm{MCL}}^{\dagger}$ (Section 5.2.4) when applying (38). Therefore, the combined advantage of our framework and Theorem 1 of Wu et al. (2023) provides CCN scenarios unified analyses whose key steps can also be rationally interpreted. Moreover, as will be shown later, we compare the marginal chain and the inversion approaches via a CL example in Section 5.2.5. A CL example is the simplest way to convey the differences between the two methods without burying the essence in complicated derivations.

### 5.2.5 Complementary-Label (CL) Learning

**Step 1: Corrected Loss Design and Risk Rewrite.**
Note that the parameters chosen for the construction of $M_{\mathrm{CL}}$ (76) in Section 4.2.5 reduces $M_{\mathrm{MCL}}$ (113) to be $M_1$ of (114). That is, assigning $\mathcal{P}_{|\bar{S}|=d} = 1$ for $d = 1$, $\mathcal{P}_{|\bar{S}|=d} = 0$ for $d > 1$, and $\bar{s}_{1,j} = \{j\}$ for all $j \in [K]$ in (113), we have

$$M_{\mathrm{MCL}} \rightarrow M_1 = \frac{1}{K-1}\begin{pmatrix} 0 & 1 & \cdots & 1 \\ 1 & 0 & \cdots & 1 \\ \vdots & \vdots & \ddots & \vdots \\ 1 & 1 & \cdots & 0 \end{pmatrix} = M_{\mathrm{CL}}.$$

Hence, the proof steps of Theorem 42 carry over to CL learning. With a simple rearranging on

$$\begin{aligned} \bar{\ell}_{\bar{S}} &= \sum_{i \notin \bar{S}} \ell_{Y=i} - \frac{K-1-|\bar{S}|}{|\bar{S}|} \sum_{\bar{s} \in \bar{S}} \ell_{Y=\bar{s}} \\ &= \sum_{i=1}^{K} \ell_{Y=i} - \frac{K-1}{|\bar{S}|} \sum_{\bar{s} \in \bar{S}} \ell_{Y=\bar{s}} \end{aligned}$$

and assigning $|\bar{S}| = 1$, we arrive at (21):

**Corollary 44.** *For CL learning, the classification risk can be expressed as*

$$R(g) = \mathbb{E}_{\bar{S},X}\left[\bar{\ell}_{\bar{S}}\right] = \mathbb{E}_{\bar{S},X}\left[\sum_{i=1}^{K} \ell_{Y=i} - (K-1)\ell_{\bar{S}}\right].$$

**Step 2: Recovering the previous result(s).**
The rewrite above recovers Theorem 1 of Ishida et al. (2019) if we substitute $\bar{S}$ with $\bar{Y}$ and $\ell_{\bar{S}}$ with $\ell(\bar{Y}, g(X))$. Moreover, if we choose $d = 1$ and $\bar{s}_{1,j} = \{j\}$ for all $j \in [K]$, the decontamination matrix provided by (117) becomes

$$M_1^{-1} = \begin{pmatrix} -(K-2) & 1 & \cdots & 1 \\ 1 & -(K-2) & \cdots & 1 \\ \vdots & \vdots & \ddots & \vdots \\ 1 & 1 & \cdots & -(K-2) \end{pmatrix}, \tag{123}$$

which translates the corrected losses $\bar{L}^{\top} = L^{\top} M_1^{-1}$ as

$$L^{\top}\left(-(K-2)\mathbf{I}_K + \mathbf{1}\mathbf{1}^{\top}\right),$$

recovering (9) of Ishida et al. (2019).

**Comparing inversion with marginal chain via an example.**
We use a simple CL example to demonstrate the differences between the inversion (Proposition 1) and the marginal chain (Proposition 2) approaches and explain how the intuition of decontamination is implemented. Here, we focus on comparing how a decontamination matrix $M_{\mathrm{corr}}^{\dagger}$ achieves $M_{\mathrm{corr}}^{\dagger}\bar{P} = P$ (37) since when the equality is established, the downstream construction of the corrected losses and the risk rewrite follow the framework. For this example, let us choose $K = 4$ and simplify $\mathcal{P}_{Y=k,X}$ as $p_k$. Applying (76), the contamination process defining the data-generating distributions is expressed as

$$\bar{P} = M_{\mathrm{CL}}P = \frac{1}{3}\begin{pmatrix} 0 & 1 & 1 & 1 \\ 1 & 0 & 1 & 1 \\ 1 & 1 & 0 & 1 \\ 1 & 1 & 1 & 0 \end{pmatrix}\begin{pmatrix} p_1 \\ p_2 \\ p_3 \\ p_4 \end{pmatrix} = \begin{pmatrix} \frac{p_2+p_3+p_4}{3} \\ \frac{p_1+p_3+p_4}{3} \\ \frac{p_1+p_2+p_4}{3} \\ \frac{p_1+p_2+p_3}{3} \end{pmatrix}.$$

Equation (123), simplified from (117), provides the decontamination matrix from the inversion approach:

$$M_{\mathrm{CL}}^{-1} = \begin{pmatrix} -2 & 1 & 1 & 1 \\ 1 & -2 & 1 & 1 \\ 1 & 1 & -2 & 1 \\ 1 & 1 & 1 & -2 \end{pmatrix}.$$

Then, the inversion approach (Proposition 1) achieves the decontamination (37) by showing

$$
\begin{aligned}
M_{\mathrm{CL}}^{-1}\bar{P} &= \frac{1}{3}\begin{pmatrix} -2 & 1 & 1 & 1 \\ 1 & -2 & 1 & 1 \\ 1 & 1 & -2 & 1 \\ 1 & 1 & 1 & -2 \end{pmatrix}\begin{pmatrix} 0 & 1 & 1 & 1 \\ 1 & 0 & 1 & 1 \\ 1 & 1 & 0 & 1 \\ 1 & 1 & 1 & 0 \end{pmatrix}\begin{pmatrix} p_1 \\ p_2 \\ p_3 \\ p_4 \end{pmatrix} \\
&= \frac{1}{3}\begin{pmatrix} 3 & 0 & 0 & 0 \\ 0 & 3 & 0 & 0 \\ 0 & 0 & 3 & 0 \\ 0 & 0 & 0 & 3 \end{pmatrix}\begin{pmatrix} p_1 \\ p_2 \\ p_3 \\ p_4 \end{pmatrix} = \begin{pmatrix} p_1 \\ p_2 \\ p_3 \\ p_4 \end{pmatrix} = P.
\end{aligned}
\tag{124}
$$

On the other hand, equation (111) produces the decontamination matrix from the marginal chain approach:

$$M_{\mathrm{CL}}^{\dagger} = \begin{pmatrix} \frac{0\cdot p_1}{p_2+p_3+p_4} & \frac{p_1}{p_1+p_3+p_4} & \frac{p_1}{p_1+p_2+p_4} & \frac{p_1}{p_1+p_2+p_3} \\ \frac{p_2}{p_2+p_3+p_4} & \frac{0\cdot p_2}{p_1+p_3+p_4} & \frac{p_2}{p_1+p_2+p_4} & \frac{p_2}{p_1+p_2+p_3} \\ \frac{p_3}{p_2+p_3+p_4} & \frac{p_3}{p_1+p_3+p_4} & \frac{0\cdot p_3}{p_1+p_2+p_4} & \frac{p_3}{p_1+p_2+p_3} \\ \frac{p_4}{p_2+p_3+p_4} & \frac{p_4}{p_1+p_3+p_4} & \frac{p_4}{p_1+p_2+p_4} & \frac{0\cdot p_4}{p_1+p_2+p_3} \end{pmatrix}.$$

Then, the marginal chain approach (Proposition 2) achieves the decontamination (37) by showing

$$
\begin{aligned}
M_{\mathrm{CL}}^{\dagger}\bar{P} \ &= \ 
\begin{pmatrix}
\frac{0\cdot p_1}{p_2+p_3+p_4} & \frac{p_1}{p_1+p_3+p_4} & \frac{p_1}{p_1+p_2+p_4} & \frac{p_1}{p_1+p_2+p_3} \\[4pt]
\frac{p_2}{p_2+p_3+p_4} & \frac{0\cdot p_2}{p_1+p_3+p_4} & \frac{p_2}{p_1+p_2+p_4} & \frac{p_2}{p_1+p_2+p_3} \\[4pt]
\frac{p_3}{p_2+p_3+p_4} & \frac{p_3}{p_1+p_3+p_4} & \frac{0\cdot p_3}{p_1+p_2+p_4} & \frac{p_3}{p_1+p_2+p_3} \\[4pt]
\frac{p_4}{p_2+p_3+p_4} & \frac{p_4}{p_1+p_3+p_4} & \frac{p_4}{p_1+p_2+p_4} & \frac{0\cdot p_4}{p_1+p_2+p_3}
\end{pmatrix}
\begin{pmatrix}
\frac{p_2+p_3+p_4}{3} \\[4pt]
\frac{p_1+p_3+p_4}{3} \\[4pt]
\frac{p_1+p_2+p_4}{3} \\[4pt]
\frac{p_1+p_2+p_3}{3}
\end{pmatrix} \\[10pt]
&= \ 
\begin{pmatrix}
\frac{p_1+p_1+p_1}{3} \\[4pt]
\frac{p_2+p_2+p_2}{3} \\[4pt]
\frac{p_3+p_3+p_3}{3} \\[4pt]
\frac{p_4+p_4+p_4}{3}
\end{pmatrix}
= 
\begin{pmatrix}
p_1 \\ p_2 \\ p_3 \\ p_4
\end{pmatrix}
= P.
\end{aligned}
\tag{125}
$$

Comparing (124) and (125), we see that the intuition of decontamination is realized differently. The inversion approach (124) directly cancels out the effect of $M_{\mathrm{corr}}$ without relying on any property of $P$. In contrast, the marginal chain method (125) leverages the fact that $P$ is a probability vector and carries out a procedure similar to importance reweighting to resolve the contamination. Both methods have respective merits, and we hope the comparison will inspire new thoughts leveraging certain properties of $P$ for the corrected loss design and the study of decontamination.

### 5.3 Confidence-based Scenarios

The proposed framework is now applied to conduct the risk rewrites for WSLs discussed in Section 4.3 and summarized in Table 9.

#### 5.3.1 Subset Confidence (Sub-Conf) Learning

**Step 1: Corrected Loss Design and Risk Rewrite.**
Let us follow the notations in Section 4.3.1. Recall that Lemma 16 has reached the first milestone (36) by showing $\bar{P} = M_{\mathrm{Sub}}P$. To reach the second milestone (37), we apply Proposition 1 to construct the decontamination matrix $M_{\mathrm{Sub}}^{\dagger}$ to cancel out the contamination caused by $M_{\mathrm{Sub}}$ (80) as follows.

**Lemma 45.** *Assume $\mathcal{P}_{Y\in\mathcal{Y}_{\mathrm{s}}|X} > 0$ for all possible outcomes of $X$. Choosing*

$$
M_{\mathrm{Sub}}^{\dagger} := M_{\mathrm{Sub}}^{-1} = 
\begin{pmatrix}
\frac{\mathcal{P}_{Y=1|X}}{\mathcal{P}_{Y\in\mathcal{Y}_{\mathrm{s}}|X}} & \cdots & 0 \\[6pt]
\vdots & \ddots & \vdots \\[6pt]
0 & \cdots & \frac{\mathcal{P}_{Y=K|X}}{\mathcal{P}_{Y\in\mathcal{Y}_{\mathrm{s}}|X}}
\end{pmatrix},
\tag{126}
$$

*we have $M_{\mathrm{Sub}}^{\dagger}\bar{P} = P$, where $\bar{P}$ and $M_{\mathrm{Sub}}P$ are given by Lemma 16.*

*Proof.* The assumption $\mathcal{P}_{Y\in\mathcal{Y}_{\mathrm{s}}|X} > 0$ implies $M_{\mathrm{Sub}}$ is invertible. As suggested by Proposition 1, we define $M_{\mathrm{Sub}}^{\dagger} := M_{\mathrm{Sub}}^{-1}$. Then,

$$
M_{\mathrm{Sub}}^{\dagger}M_{\mathrm{Sub}} = 
\begin{pmatrix}
\frac{\mathcal{P}_{Y=1|X}}{\mathcal{P}_{Y\in\mathcal{Y}_{\mathrm{s}}|X}} & \cdots & 0 \\[6pt]
\vdots & \ddots & \vdots \\[6pt]
0 & \cdots & \frac{\mathcal{P}_{Y=K|X}}{\mathcal{P}_{Y\in\mathcal{Y}_{\mathrm{s}}|X}}
\end{pmatrix}
\begin{pmatrix}
\frac{\mathcal{P}_{Y\in\mathcal{Y}_{\mathrm{s}}|X}}{\mathcal{P}_{Y=1|X}} & \cdots & 0 \\[6pt]
\vdots & \ddots & \vdots \\[6pt]
0 & \cdots & \frac{\mathcal{P}_{Y\in\mathcal{Y}_{\mathrm{s}}|X}}{\mathcal{P}_{Y=K|X}}
\end{pmatrix}
= I
$$

implies $M_{\mathrm{Sub}}^{\dagger}\bar{P} = M_{\mathrm{Sub}}^{-1}M_{\mathrm{Sub}}P = P$ that proves the lemma. $\qquad\square$

With $M_{\text{Sub}}^{\dagger}$ in hand, the next theorem defines the corrected losses $\bar{L}$ and achieves the risk rewrite (31).

**Theorem 46.** *For Sub-Conf learning, the classification risk can be written as*

$$R(g) = \pi_{\mathcal{Y}_{\text{s}}} \mathbb{E}_{X|Y \in \mathcal{Y}_{\text{s}}} \left[ \sum_{i=1}^{K} \frac{r_i(X)}{r_{\mathcal{Y}_{\text{s}}}(X)} \ell_i \right].$$

*Proof.* Given Lemma 45, we can define $\bar{L}^{\top} := L^{\top} M_{\text{Sub}}^{\dagger}$ so that

$$\bar{L}_i^{\top} = \left( L^{\top} M_{\text{Sub}}^{\dagger} \right)_i = \frac{\mathcal{P}_{Y=i|X}}{\mathcal{P}_{Y \in \mathcal{Y}_{\text{s}}|X}} \ell_i$$

for each $i \in [K]$ and

$$\bar{L}^{\top} \bar{P} = L^{\top} M_{\text{Sub}}^{\dagger} \bar{P} = L^{\top} P.$$

Therefore, we can apply (39) to obtain (31) as follows.

$$
\begin{aligned}
R(g) &= \int_{x \in \mathcal{X}} L^{\top} P \mathrm{d}x = \int_{x \in \mathcal{X}} \bar{L}^{\top} \bar{P} \mathrm{d}x = \int_{x \in \mathcal{X}} \sum_{i=1}^{K} \frac{\mathcal{P}_{Y=i|X}}{\mathcal{P}_{Y \in \mathcal{Y}_{\text{s}}|X}} \ell_i \cdot \mathcal{P}_{Y \in \mathcal{Y}_{\text{s}}} \mathcal{P}_{X|Y \in \mathcal{Y}_{\text{s}}} \mathrm{d}x \\
&= \mathcal{P}_{Y \in \mathcal{Y}_{\text{s}}} \mathbb{E}_{X|Y \in \mathcal{Y}_{\text{s}}} \left[ \sum_{i=1}^{K} \frac{\mathcal{P}_{Y=i|X}}{\mathcal{P}_{Y \in \mathcal{Y}_{\text{s}}|X}} \ell_i \right] \\
&= \pi_{\mathcal{Y}_{\text{s}}} \mathbb{E}_{X|Y \in \mathcal{Y}_{\text{s}}} \left[ \sum_{i=1}^{K} \frac{r_i(X)}{r_{\mathcal{Y}_{\text{s}}}(X)} \ell_i \right].
\end{aligned}
$$

The last equality follows the notations in Section 2.2.14. $\qquad \square$

**Step 2: Recovering the previous result(s).**
Notation matching gives

$$R(g) = \pi_{\mathcal{Y}_{\text{s}}} \mathbb{E}_{p(x|y \in \mathcal{Y}_{\text{s}})} \left[ \sum_{y=1}^{K} \frac{r^y(x)}{r^{\mathcal{Y}_{\text{s}}}(x)} \ell(g(x), y) \right],$$

recovering Theorem 6 of Cao et al. (2021a)[14].

### 5.3.2 Single-Class Confidence (SC-Conf) Learning

**Step 1: Corrected Loss Design and Risk Rewrite.**
The SC-Conf derivation resembles that in Section 5.3.1 since $M_{\text{SC}}$ is a child of $M_{\text{Sub}}$ on the reduction graph. Thus, following the notations in Section 4.3.2, assuming $\mathcal{P}_{Y=y_{\text{s}}|X} > 0$ for all possible outcomes of $X$, and replacing the set $\mathcal{Y}_{\text{s}}$ in $M_{\text{Sub}}^{\dagger}$ (126) with a singleton $y_{\text{s}}$, we have

$$M_{\text{SC}}^{\dagger} := \begin{pmatrix} \frac{\mathcal{P}_{Y=1|X}}{\mathcal{P}_{Y=y_{\text{s}}|X}} & \cdots & 0 \\ \vdots & \ddots & \vdots \\ 0 & \cdots & \frac{\mathcal{P}_{Y=K|X}}{\mathcal{P}_{Y=y_{\text{s}}|X}} \end{pmatrix}$$

satisfying $M_{\text{SC}}^{\dagger} \bar{P} = P$. We also obtain $\bar{L}^{\top} = L^{\top} M_{\text{SC}}^{\dagger}$ and $\bar{L}^{\top} \bar{P} = L^{\top} P$ by inheriting the proof of Lemma 45. Then, a modification to Theorem 46 by replacing $\bar{L}_i^{\top} = \frac{\mathcal{P}_{Y=i|X}}{\mathcal{P}_{Y \in \mathcal{Y}_{\text{s}}|X}} \ell_i$ with

$$\bar{L}_i^{\top} = \left( L^{\top} M_{\text{SC}}^{\dagger} \right)_i = \frac{\mathcal{P}_{Y=i|X}}{\mathcal{P}_{Y=y_{\text{s}}|X}} \ell_i = \frac{r_i(X)}{r_{y_{\text{s}}}(X)} \ell_i$$

proves the risk rewrite (29) for SC-Conf learning:

---

[14]The matching is as follows: $\mathcal{P}_{X|Y \in \mathcal{Y}_{\text{s}}}$ is $p(x|y \in \mathcal{Y}_{\text{s}})$, $r_i(X)$ is $r^i(X)$, $r_{\mathcal{Y}_{\text{s}}}(X)$ is $r^{\mathcal{Y}_{\text{s}}}(X)$, and $\ell_i$ is $\ell(g(X), i)$.

**Corollary 47.** *For SC-Conf learning, the classification risk can be written as*

$$R(g) = \pi_{y_{\mathrm{s}}} \mathbb{E}_{X|Y=y_{\mathrm{s}}} \left[ \sum_{i=1}^{K} \frac{r_i(X)}{r_{y_{\mathrm{s}}}(X)} \ell_i \right].$$

**Step 2: Recovering the previous result(s).**
By matching notations, we obtain

$$R(g) = \pi_{y_{\mathrm{s}}} \mathbb{E}_{p(x|y_{\mathrm{s}})} \left[ \sum_{y=1}^{K} \frac{r^y(x)}{r^{y_{\mathrm{s}}}(x)} \ell(g(x), y) \right],$$

recovering Theorem 1 of Cao et al. (2021a)[15].

### 5.3.3  Positive-confidence (Pconf) Learning

**Step 1: Corrected Loss Design and Risk Rewrite.**
Let us follow the notations in Section 4.3.3. Recall that $M_{\mathrm{Pconf}}$ is a child of $M_{\mathrm{SC}}$ on the reduction graph with $K = 2$ and $y_{\mathrm{S}} = \mathrm{p}$. Thus, assuming $\mathcal{P}_{Y=\mathrm{p}|X} > 0$ for all possible outcomes of $X$ and replacing $K$ and $y_{\mathrm{s}}$ in Section 5.3.2 accordingly, we obtain the decontamination matrix

$$M_{\mathrm{Pconf}}^{\dagger} := \begin{pmatrix} \frac{\mathcal{P}_{Y=\mathrm{p}|X}}{\mathcal{P}_{Y=\mathrm{p}|X}} & 0 \\ 0 & \frac{\mathcal{P}_{Y=\mathrm{n}|X}}{\mathcal{P}_{Y=\mathrm{p}|X}} \end{pmatrix} = \begin{pmatrix} 1 & 0 \\ 0 & \frac{1-r(X)}{r(X)} \end{pmatrix}$$

and the rewrite (7) reviewed in Section 2.2.2.

**Corollary 48.** *For Pconf learning, the classification risk can be written as*

$$R(g) = \pi_{\mathrm{p}} \mathbb{E}_{\mathrm{P}} \left[ \ell_{\mathrm{p}} + \frac{1-r(X)}{r(X)} \ell_{\mathrm{n}} \right].$$

**Step 2: Recovering the previous result(s).**
By matching notations, we obtain

$$R(g) = \pi_{+} \mathbb{E}_{+} \left[ \ell(g(x)) + \frac{1-r(x)}{r(x)} \ell(-g(x)) \right],$$

recovering Theorem 1 of Ishida et al. (2018)[16].

### 5.3.4  Soft-Label Learning

**Step 1: Corrected Loss Design and Risk Rewrite.**
We follow the notations in Section 4.3.4. As discussed in Section 4.3.4, $M_{\mathrm{Soft}}$ is a special case of $M_{\mathrm{Sub}}$ when $\mathcal{Y}_{\mathrm{s}} := [K]$. Thus, reducing $M_{\mathrm{Sub}}^{\dagger}$ (126) by assigning $\mathcal{P}_{Y \in \mathcal{Y}_{\mathrm{s}}|X} = \mathcal{P}_{Y \in [K]|X} = 1$, we obtain the the decontamination matrix

$$M_{\mathrm{Soft}}^{\dagger} := \begin{pmatrix} \mathcal{P}_{Y=1|X} & \cdots & 0 \\ \vdots & \ddots & \vdots \\ 0 & \cdots & \mathcal{P}_{Y=K|X} \end{pmatrix}.$$

Then, we follow the same argument in Theorem 46 to achieve the rewrite (34) by the next corollary.

**Corollary 49.** *For soft-label learning, the classification risk can be written as*

$$R(g) = \mathbb{E}_X \left[ \sum_{i=1}^{K} \mathcal{P}_{Y=i|X} \ell_i \right] = \mathbb{E}_X \left[ \sum_{i=1}^{K} r_i(X) \ell_i \right].$$

---

[15]The matching is as follows: $\mathcal{P}_{X|Y=y_{\mathrm{s}}}$ is $p(x|y_{\mathrm{s}})$, $r_i(X)$ is $r^i(X)$, $r_{y_{\mathrm{s}}}(X)$ is $r^{y_{\mathrm{s}}}(X)$, and $\ell_i$ is $\ell(g(X), i)$.

[16]The matching is as follows: $\pi_{\mathrm{p}}$ is $\pi_+$, $\mathcal{P}_{X|Y=\mathrm{p}}$ is $p(x|y=+1)$, $\mathcal{P}_{Y=\mathrm{p}|X}$ is $r(x)$, $\mathcal{P}_{Y=\mathrm{n}|X}$ is $1 - r(x)$, $\ell_{\mathrm{p}}$ is $\ell(g(x))$, and $\ell_{\mathrm{n}}$ is $\ell(-g(x))$.

**Step 2: Recovering the previous result(s).**
Ishida et al. (2023) did not focus on the classification risk rewrite problem. We can modify Corollary 49 to provide a risk rewrite for binary soft-label learning mentioned by Ishida et al. (2023). Taking $K = 2$, we have

$$R(g) = \mathbb{E}_X \left[ \mathcal{P}_{Y=\mathrm{p}|X} \ell_\mathrm{p} + \mathcal{P}_{Y=\mathrm{n}|X} \ell_\mathrm{n} \right] = \mathbb{E}_X \left[ r(X) \ell_\mathrm{p} + (1 - r(X)) \ell_\mathrm{n} \right].$$

## 6 New Risk Rewrites

The proposed framework can be applied to derive risk rewrites for new scenarios. In this section, we show how the proposed framework accommodates a different performance metric and adapts to a noisy environment.

### 6.1 The Balanced Error Rate

In the previous sections, we chose the most common metric, the classification risk, to elaborate our framework. Here, we demonstrate that the framework adapts to another common but different performance metric. The balanced error rate (BER) is defined as

$$R_{\mathrm{BER}}(g) := \frac{\mathbb{E}_{X \sim \mathcal{P}_{X|Y=\mathrm{p}}} \left[ \ell_\mathrm{p}(g(X)) \right] + \mathbb{E}_{X \sim \mathcal{P}_{X|Y=\mathrm{n}}} \left[ \ell_\mathrm{n}(g(X)) \right]}{2} \tag{127}$$

(Scott & Zhang, 2020). Next, we show how to apply the framework to obtain the risk rewrite of UU learning under BER.

Recall that in Section 5.1.1, for classification risk, $M_{\mathrm{trsf}} = \begin{pmatrix} 1/\pi_\mathrm{p} & 0 \\ 0 & 1/\pi_\mathrm{n} \end{pmatrix}$ was chosen to link the base

distributions $B = \begin{pmatrix} \mathcal{P}_{X|Y=\mathrm{p}} \\ \mathcal{P}_{X|Y=\mathrm{n}} \end{pmatrix}$ and the risk-defining distributions $P = \begin{pmatrix} \mathcal{P}_{Y=\mathrm{p},X} \\ \mathcal{P}_{Y=\mathrm{n},X} \end{pmatrix}$:

$$\begin{pmatrix} \mathcal{P}_{X|Y=\mathrm{p}} \\ \mathcal{P}_{X|Y=\mathrm{n}} \end{pmatrix} = \begin{pmatrix} 1/\pi_\mathrm{p} & 0 \\ 0 & 1/\pi_\mathrm{n} \end{pmatrix} \begin{pmatrix} \mathcal{P}_{Y=\mathrm{p},X} \\ \mathcal{P}_{Y=\mathrm{n},X} \end{pmatrix}.$$

By definition (127), the risk-defining distributions become $\begin{pmatrix} \mathcal{P}_{X|Y=\mathrm{p}}/2 \\ \mathcal{P}_{X|Y=\mathrm{n}}/2 \end{pmatrix}$. Since the base distributions $B$

remain unchanged, we must redefine $M_{\mathrm{trsf}} := \begin{pmatrix} 2 & 0 \\ 0 & 2 \end{pmatrix}$, so that

$$\begin{pmatrix} \mathcal{P}_{X|Y=\mathrm{p}} \\ \mathcal{P}_{X|Y=\mathrm{n}} \end{pmatrix} = \begin{pmatrix} 2 & 0 \\ 0 & 2 \end{pmatrix} \begin{pmatrix} \mathcal{P}_{X|Y=\mathrm{p}}/2 \\ \mathcal{P}_{X|Y=\mathrm{n}}/2 \end{pmatrix}$$

satisfies $B = M_{\mathrm{trsf}} P$ for the BER setting.

With the newly defined $M_{\mathrm{trsf}} = \begin{pmatrix} 2 & 0 \\ 0 & 2 \end{pmatrix}$ in hand, we can follow Step 1 in Section 5.1.1 to obtain the data-generating process

$$\begin{pmatrix} \mathcal{P}_{\mathrm{U}_1} \\ \mathcal{P}_{\mathrm{U}_2} \end{pmatrix} = M_{\mathrm{UU}} \begin{pmatrix} 2 & 0 \\ 0 & 2 \end{pmatrix} \begin{pmatrix} \mathcal{P}_{X|Y=\mathrm{p}}/2 \\ \mathcal{P}_{X|Y=\mathrm{n}}/2 \end{pmatrix}.$$

Then, by assigning $M_{\mathrm{UU-BER}}^\dagger := \begin{pmatrix} 1/2 & 0 \\ 0 & 1/2 \end{pmatrix} M_{\mathrm{UU}}^{-1}$ and following the similar derivation in the proof of Theorem 21, we achieve the risk rewrite for UU learning under BER:

$$R_{\mathrm{BER}}(g) = \mathbb{E}_{\mathrm{U}_1} \left[ \bar{\ell}_{\mathrm{U}_1} \right] + \mathbb{E}_{\mathrm{U}_2} \left[ \bar{\ell}_{\mathrm{U}_2} \right],$$

where

$$\bar{\ell}_{U_1} = \frac{1-\gamma_2}{2(1-\gamma_1-\gamma_2)}\ell_p + \frac{-\gamma_2}{2(1-\gamma_1-\gamma_2)}\ell_n,$$

$$\bar{\ell}_{U_2} = \frac{-\gamma_1}{2(1-\gamma_1-\gamma_2)}\ell_p + \frac{1-\gamma_1}{2(1-\gamma_1-\gamma_2)}\ell_n.$$

## 6.2 Learning with Label Noise

In this section, we show that the framework can easily handle the risk rewrite under label noise. We will take PU learning as an example. We use $\mathcal{P}$ to denote the clean distributions and $\mathcal{Q}$ for the noisy ones. The scenario is formulated as follows. According to Lemma 4, the data-generating process of the noisy PU learning is of the form

$$\begin{pmatrix} \mathcal{Q}_{P'} \\ \mathcal{Q}_{U'} \end{pmatrix} = \begin{pmatrix} 1 & 0 \\ \pi'_p & \pi'_n \end{pmatrix} \begin{pmatrix} \mathcal{Q}_{X|Y'=p} \\ \mathcal{Q}_{X|Y'=n} \end{pmatrix},$$

where $\pi'_p = \mathcal{Q}_{Y'=p}$ and $\pi'_n = \mathcal{Q}_{Y'=n}$. We choose the MCD setting (Section 2.2.17) to formulate the label noise:

$$\begin{pmatrix} \mathcal{Q}_{X|Y'=p} \\ \mathcal{Q}_{X|Y'=n} \end{pmatrix} = \begin{pmatrix} 1-\alpha'_p & \alpha'_p \\ \alpha'_n & 1-\alpha'_n \end{pmatrix} \begin{pmatrix} \mathcal{P}_{X|Y=p} \\ \mathcal{P}_{X|Y=n} \end{pmatrix},$$

where $\alpha'_p \geq 0$ and $\alpha'_n \geq 0$ are parameters describing the degree of noise perturbation. As in the previous section, we also choose BER as the performance metric.

By cascading matrices, the framework effortlessly links the data-generating distributions and the risk-defining distributions:

$$\begin{pmatrix} \mathcal{Q}_{P'} \\ \mathcal{Q}_{U'} \end{pmatrix} = \begin{pmatrix} 1 & 0 \\ \pi'_p & \pi'_n \end{pmatrix} \begin{pmatrix} 1-\alpha'_p & \alpha'_p \\ \alpha'_n & 1-\alpha'_n \end{pmatrix} \begin{pmatrix} 2 & 0 \\ 0 & 2 \end{pmatrix} \begin{pmatrix} \mathcal{P}_{X|Y=p}/2 \\ \mathcal{P}_{X|Y=n}/2 \end{pmatrix}.$$

Knowing the contamination process, we apply the inversion method (Proposition 1) to construct the decontamination matrix

$$
\begin{aligned}
M^{\dagger}_{\text{PU-BER}} \; &:= \; \begin{pmatrix} 1/2 & 0 \\ 0 & 1/2 \end{pmatrix} \frac{1}{1-\alpha'_p-\alpha'_n} \begin{pmatrix} 1-\alpha'_n & -\alpha'_p \\ -\alpha'_n & 1-\alpha'_p \end{pmatrix} \begin{pmatrix} 1 & 0 \\ -\pi'_p/\pi'_n & 1/\pi'_n \end{pmatrix} \\
&= \; \begin{pmatrix} \frac{(1-\alpha'_n)\pi'_n+\alpha'_p\pi'_p}{2(1-\alpha'_p-\alpha'_n)\pi'_n} & \frac{-\alpha'_p}{2(1-\alpha'_p-\alpha'_n)\pi'_n} \\ \frac{-\alpha'_n\pi'_n-(1-\alpha'_p)\pi'_p}{2(1-\alpha'_p-\alpha'_n)\pi'_n} & \frac{1-\alpha'_p}{2(1-\alpha'_p-\alpha'_n)\pi'_n} \end{pmatrix} := \begin{pmatrix} c_1 & c_2 \\ c_3 & c_4 \end{pmatrix}.
\end{aligned}
$$

Then, applying equations (38) and (39), we can rewrite the BER for PU learning as

$$R_{\text{BER}}(g) = \mathbb{E}_{X \sim \mathcal{Q}_{P'}} \left[ \bar{\ell}_{P'} \right] + \mathbb{E}_{X \sim \mathcal{Q}_{U'}} \left[ \bar{\ell}_{U'} \right],$$

where

$$\bar{\ell}_{P'} = c_1\ell_p + c_3\ell_n,$$
$$\bar{\ell}_{U'} = c_2\ell_p + c_4\ell_n.$$

In the two subsections above, we have shown that the proposed framework can address risk rewrite under a different performance metric and a complex system. In the outlook part of the next section, we will further discuss the potential of the framework.

# 7 Conclusion and Outlook

We set out with the questions wishing to determine if there is a common way to interpret the formation of weak supervision and search for a generic treatment to solve WSL, to understand the essence of WSL. In response, we proposed a framework that unifies the formulations and analyses of a set of WSL scenarios to provide a common ground to connect, compare, and understand various weakly-supervised signals. The formulation component of the proposed framework, viewing WSL from a contamination perspective, associates a WSL data-generating process with a base distribution vector multiplied by a contamination matrix. By instantiating the contamination matrices of WSLs, we revealed a comprehensive reduction graph, Figure 1, connecting existing WSLs. Each vertex contains a contamination matrix and the section index of the WSL scenario which the matrix characterizes. Each edge represents the reduction relation of two WSLs. We can see three major branches from the abstract $M_{\mathrm{corr}}$, corresponding to Tables 7, 8, and 9 we discussed in Section 4. The analysis component of the proposed framework, tackling the problem from a decontamination viewpoint, working with the technical building blocks Theorems 1 and 2 constitute a generic treatment to solve the risk rewrite problem. Section 5 discussed in depth how the analysis component conducts risk rewrite and recovers existing results for WSLs.

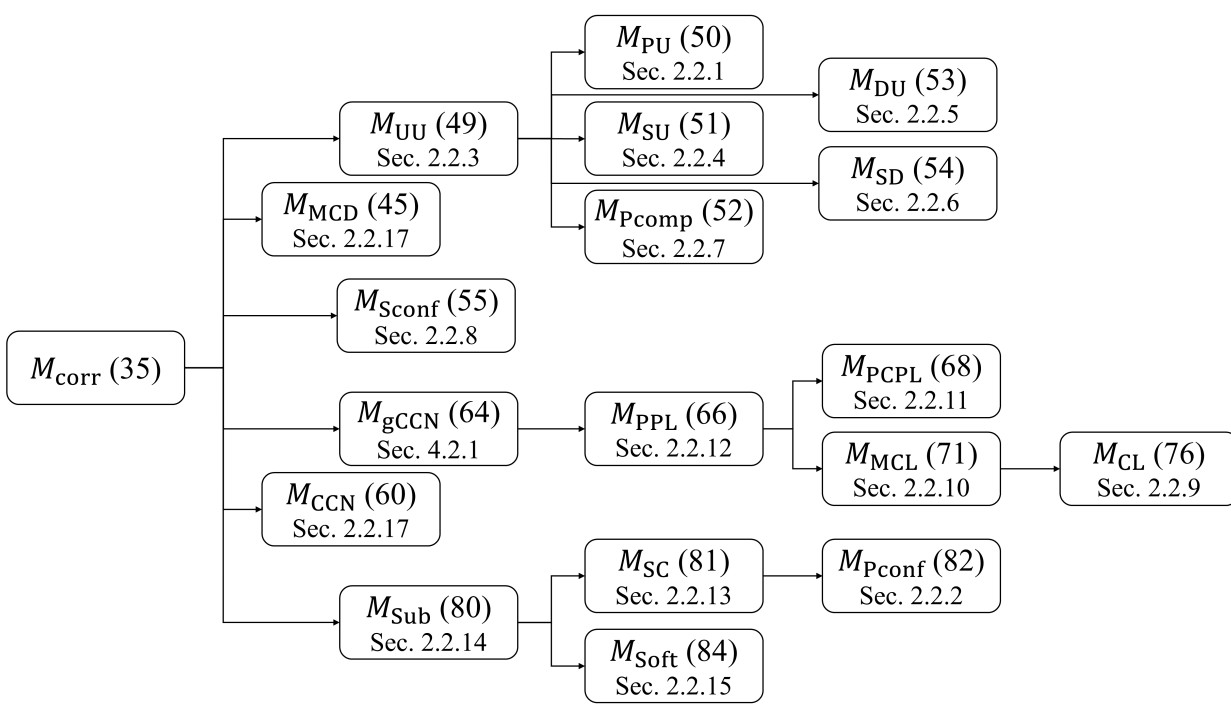

Figure 1: Depicting the reduction map from Tables 7, 8, and 9.

The application of the proposed framework results in a set of theorems. We summarize them in Table 10. The Formulation column consists of the results of the formulation component (35). The Decontamination and the Corrected losses columns correspond to the results of the analysis component ((37), (38), and (39)). The Recovery column justifies the framework by recovering results from the literature. Crucial results are marked red. Since the analyses of different scenarios are subsumed under a single framework, we now have a basis for transferring a technique developed for one scenario to another. In addition, these alternative proofs provide different ways of dissecting the risk, which in turn could aid in the development of a training objective by examining multiple risk decomposition approaches.

Table 10: Theorem Structure.

| Model | Formulation (Find $M$ s.t. $\bar{P} = MB$.) | Decontamination (Find $M^\dagger$ s.t. $P = M^\dagger \bar{P}$.) | Corrected losses (Rewrite via $\bar{L}^\top = L^\top M^\dagger$ and $\bar{P}$.) | Recovery |
|---|---|---|---|---|
| Abstract model | (35) | (37) Proposition 1 and Proposition 2 | (38) and (39) | |
| MCD | | | | |
| UU | Lemma 3 | Corollary 20 | Theorem 21 | (Notation swap.) |
| PU | Lemma 4 | (Immediate reduction.) | Corollary 22 | (Notation swap.) |
| SU | Lemma 5 | (Immediate reduction.) | Corollary 23 | Lemmas 24 and 25 |
| Pcomp | Lemma 6 | (Immediate reduction.) | Corollary 26 | (Notation swap.) |
| DU | Lemma 7 | (Immediate reduction.) | Corollary 27 | Lemmas 28 and 29 |
| SD | Lemma 8 | (Immediate reduction.) | Corollary 30 | Lemma 31 |
| Sconf | Lemma 9 | Lemmas 32 and 33 | Theorem 34 | (Notation swap.) |
| CCN | | | | |
| gCCN | Lemma 10 | (106) and Proposition 2 | Theorem 35 | (Notation swap.) |
| PPL | Lemma 11 | Lemma 36 | Corollary 37 | (Notation swap.) |
| PCPL | Lemma 12 | Corollary 38 | Corollary 38 | Lemma 39 |
| MCL | Lemma 13 | Corollary 40 | Corollary 40 | Theorem 42, Lemmas 41 and 43 |
| CL | Lemma 15 | (Immediate reduction.) | Corollary 44 | (Notation swap.) |
| Sub-Conf | Lemma 16 | Lemma 45 | Theorem 46 | (Notation swap.) |
| SC-Conf | Lemma 17 | (Immediate reduction.) | Corollary 47 | (Notation swap.) |
| Pconf | Lemma 18 | (Immediate reduction.) | Corollary 48 | (Notation swap.) |
| Soft | Lemma 19 | (Immediate reduction.) | Corollary 49 | (N/A.) |

The proposed framework is abstract and flexible; hence, we would like to discuss its potential from the following aspects. Firstly, the performance measure focused on in this paper is the classification risk. With proper choices of $P$ and $L$, our framework can be extended to other performance metrics, such as the balanced error rate, one-versus-rest risk, and cost-sensitive measures (Rifkin & Klautau, 2004; Zhang, 2004; Brodersen et al., 2010; du Plessis et al., 2014; Menon et al., 2015; Blanchard et al., 2016; Natarajan et al., 2017; Scott & Zhang, 2020). We have demonstrated the applicability of the proposed framework for the balanced error rate in Section 6. Secondly, we can explore the formulation capability by exploiting the power of matrix operations. Cascading matrices allow us to formulate complex scenarios, such as data containing preference relations collected in a noisy environment. Matrix addition allows us to categorize different contamination mechanisms into cases to capture the structural properties of a problem. A complicated scenario could undergo a sophisticated formulation procedure, but once we have the resulting contamination matrix, the problem boils down to calculating the corresponding decontamination matrix. Thirdly, the MCD scenarios discussed in this paper (Sections 4.1 and 5.1) belong to binary classification. A way of extending an MCD formulation to multiclass classification is to extend $M_{\mathrm{MCD}}$ (45) from a $2 \times 2$ matrix to a $K \times K$ one, in which $K^2 - K$ mixture rates are used to characterize the extended $M_{\mathrm{gMCD}}$: the $(i, j)$ entry is $\gamma_{i,j}$ if $i \neq j$ and is $1 - \sum_{j \neq i} \gamma_{i,j}$ for the $i$-th entry on the diagonal. Fourthly, the label-flipping probabilities $\mathcal{P}_{\bar{Y}|Y}$ in Natarajan et al. (2017) and Feng et al. (2020b) assume that the contaminated label $\bar{Y}$ is independent of $X$ condition on the ture label $Y$. The formulation matrices, $M_{\mathrm{CCN}}$ (60) and $M_{\mathrm{gCCN}}$ (64), in contrast, take $X$ into consideration. This formulation enables us to tackle the instance-dependent problem (Berthon et al., 2021) and the effect of sampling strategies such as SAR and SCAR (Elkan & Noto, 2008; Coudray et al., 2023) in the future. Fifthly, we hope that the analysis technique developed in Section 5.2, which combines marginal chain and properness, opens up a new possibility to search for invertibility-free methods for the risk rewrite problem. We also project its potential in research regarding the broader sense of contamination

and decontamination. Sixthly, the properness of Wu et al. (2023) provides an efficient technique to compute $\mathcal{P}_{Y|S,X}$ needed in $M_{\text{gCCN}}$ (106). It would be intriguing to know if there are any other alternatives. Finally but not least, the proposed framework operating under matrix multiplication belongs to a broader question of under what circumstances does a function $f^\dagger$ exist with $P = f^\dagger(\bar{P})$ if $\bar{P} = f(P)$.

## Acknowledgments

The authors were supported by the Institute for AI and Beyond, UTokyo. The first author would like to thank Professor Takashi Ishida (UTokyo) for valuable insights and discussions in extending the coverage of the framework, and the colleagues Xin-Qiang Cai, Masahiro Negishi, Wei Wang, and Yivan Zhang (in alphabetical order) for comments in improving the manuscript.

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

## A    Use Cases for This Paper

There are several ways to use this paper in the study of WSLs. We provide some common use cases as follows:

Use Case 1: If you want a quick overview of WSL. Reading Section 2 would provide a quick catch-up on the formulations and results of various WSL scenarios.

Use Case 2: If you want to get the high-level idea of this paper. One could start with Section 3, which provides the abstract form of the proposed framework. Then, Sections 4.1.1 and 5.1.1 give the practical application of the framework to UU learning. If you are more familiar with PU learning, then you are welcome to proceed to Sections 4.1.2 and 5.1.1 to see how our framework interprets PU learning. In addition, these reading steps will reveal the connection between UU and PU.

Use Case 3: If you want to know how to apply the framework to a specific scenario. One could read Sections 4.a.b and 5.a.b at the same time, and come back to the rest of Sections 4 and 5 only as needed. Note that "a.b" represents the index of a subsubsection.

Use Case 4: If you want to know the connections discovered and the analysis techniques developed in this paper. Section 4 provides detailed discussions of the formulations, and Figure 1 summarizes the relationship. Section 5 provides detailed explanations of how our framework is applied to rewrite the classification risk. The extensive analyses from Section 5 are summarized in Table 10.

## B    Notations

Table 11:   Notations and Aliases.

| Name of the notation | Expression | Aliases | Convention |
|---|---|---|---|
| Example | $(y, x)$ | | $(x, y)$ |
| Binary classes | $\{\mathrm{p}, \mathrm{n}\}$ | | $\{+1, -1\}$ |
| Multiple classes | $\{1, \cdots, K\}$ | $[K]$ | |
| Compound classes of $[K]$ | $2^{[K]} \backslash \{\emptyset, [K]\}$ | $\mathcal{S}$ | |
| A subset of classes | $\mathcal{Y}_s \subset [K]$ | | |
| Joint distribution | $\Pr(Y = y, X = x)$ | $\mathcal{P}_{Y=y,x}$, $\mathcal{P}_{Y=y,X}$, or $\mathcal{P}_{Y,X}$ | $\Pr(x, y)$ |
| Class prior | $\Pr(Y = y)$ | $\pi_y$ | |
| Marginal | $\Pr(X)$ | $\mathcal{P}_X$ | |
| Class-conditional | $\Pr(X = x \mid Y = y)$ | $\mathcal{P}_{x\|y}$, $\mathcal{P}_{X\|y}$, $\mathcal{P}_{x\|Y=y}$, or $\mathcal{P}_{X\|Y=y}$ | |
| Class probability | $\Pr(Y = y \mid X = x)$ | $\mathcal{P}_{Y=y\|x}$, $\mathcal{P}_{Y=y\|X}$, or $\mathcal{P}_{Y\|X}$ | $\eta(x)$ |
| Confidence | $\Pr(Y = y \mid X = x)$ | $r_y(X)$, $r_y(x)$, or $r(X)$ if $y = \mathrm{p}$ | $r^y(x)$ or $r(x)$ |
| Sample size probability | $\Pr(\|S\| = d)$ | $\mathcal{P}_{\|S\|=d}$ or $q_{\|S\|}$ | |
| Hypothesis and its space | $g \in \mathcal{G}$ | | |
| Loss of $g$ | $\ell_{Y=y}(g(x))$ | $\ell_y$, $\ell_y(X)$, or $\ell_Y(g(X))$ | $\ell(g(X), Y)$ |
| Classification risk | $\mathbb{E}_{Y,X}[\ell_Y(g(X))]$ | $R(g)$ | $\mathbb{E}_{X,Y}[\ell(g(X), Y)]$ |
| The $j$-th entry of vector $V$ | $(V)_j$ | $V_j$ | |
| Indicator function of $E$ | $\mathbb{I}[E]$ | | |
| Complement of set $s$ | $\mathcal{Y}\backslash s$ | $\bar{s}$ | |
| Identity matrix | $I$ | | |
| MCD parameters | $\gamma_{\mathrm{p}}$ and $\gamma_{\mathrm{n}}$ | | |
| UU parameters | $\gamma_1$ and $\gamma_2$ | | $1 - \theta$ and $\theta'$ |
| CCN parameters | $\mathcal{P}_{\tilde{Y}\|Y,X}$ | $\mathcal{P}_{S\|Y,X}$ or $\mathcal{P}_{\bar{S}\|Y,X}$ | $\rho_+$ and $\rho_-$ |

## C  Omitted Proofs in Section 4

Omitted proofs in Section 4 are provided in this appendix. We first restate a claim in the main body of the paper, and then provide the corresponding proof.

### C.1  Omitted Proofs in Section 4.1

**Proof of Lemma 5**

**Lemma 5.** *Let B (46) be the base distributions and*

$$\bar{P} := \begin{pmatrix} \mathcal{P}_{\tilde{S}} \\ \mathcal{P}_{U} \end{pmatrix}.$$

*Then, the contamination matrix*

$$M_{\mathrm{SU}} := \begin{pmatrix} \frac{\pi_{\mathrm{p}}^2}{\pi_{\mathrm{p}}^2 + \pi_{\mathrm{n}}^2} & \frac{\pi_{\mathrm{n}}^2}{\pi_{\mathrm{p}}^2 + \pi_{\mathrm{n}}^2} \\ \pi_{\mathrm{p}} & \pi_{\mathrm{n}} \end{pmatrix}, \tag{51}$$

*which satisfies $\bar{P} = M_{\mathrm{SU}}B$, characterizes the data-generating distributions $\bar{P}$.*

*Proof.* The proof steps follow that of Lemma 4. By definitions,

$$M_{\mathrm{SU}}B = \begin{pmatrix} \frac{\pi_{\mathrm{p}}^2}{\pi_{\mathrm{p}}^2 + \pi_{\mathrm{n}}^2} & \frac{\pi_{\mathrm{n}}^2}{\pi_{\mathrm{p}}^2 + \pi_{\mathrm{n}}^2} \\ \pi_{\mathrm{p}} & \pi_{\mathrm{n}} \end{pmatrix} \begin{pmatrix} \mathcal{P}_{X|Y=\mathrm{p}} \\ \mathcal{P}_{X|Y=\mathrm{n}} \end{pmatrix}.$$

Since

$$\frac{\pi_{\mathrm{p}}^2}{\pi_{\mathrm{p}}^2 + \pi_{\mathrm{n}}^2}\mathcal{P}_{X|Y=\mathrm{p}} + \frac{\pi_{\mathrm{n}}^2}{\pi_{\mathrm{p}}^2 + \pi_{\mathrm{n}}^2}\mathcal{P}_{X|Y=\mathrm{n}} = \frac{\pi_{\mathrm{p}}^2\mathcal{P}_{X|Y=\mathrm{p}} + \pi_{\mathrm{n}}^2\mathcal{P}_{X|Y=\mathrm{p}}}{\pi_{\mathrm{p}}^2 + \pi_{\mathrm{n}}^2} = \mathcal{P}_{\tilde{S}}$$

and

$$\pi_{\mathrm{p}}\mathcal{P}_{X|Y=\mathrm{p}} + \pi_{\mathrm{n}}\mathcal{P}_{X|Y=\mathrm{n}} = \mathcal{P}_X = \mathcal{P}_{\mathrm{U}},$$

the first entry of the resulting vector equals $\mathcal{P}_{\tilde{S}}$ and the second entry equals $\mathcal{P}_{\mathrm{U}}$, we achieve $\bar{P} = M_{\mathrm{SU}}B$.  □

**Proof of Lemma 6**

**Lemma 6.** *Let B (46) be the base distributions and*

$$\bar{P} := \begin{pmatrix} \mathcal{P}_{\mathrm{Sup}} \\ \mathcal{P}_{\mathrm{Inf}} \end{pmatrix}.$$

*Then, the contamination matrix*

$$M_{\mathrm{Pcomp}} := \begin{pmatrix} \frac{\pi_{\mathrm{p}}}{\pi_{\mathrm{p}} + \pi_{\mathrm{n}}^2} & \frac{\pi_{\mathrm{n}}^2}{\pi_{\mathrm{p}} + \pi_{\mathrm{n}}^2} \\ \frac{\pi_{\mathrm{p}}^2}{\pi_{\mathrm{p}}^2 + \pi_{\mathrm{n}}} & \frac{\pi_{\mathrm{n}}}{\pi_{\mathrm{p}}^2 + \pi_{\mathrm{n}}} \end{pmatrix}, \tag{52}$$

*which satisfies $\bar{P} = M_{\mathrm{Pcomp}}B$, characterizes the data-generating distributions $\bar{P}$.*

*Proof.* By definitions,

$$M_{\mathrm{Pcomp}}B = \begin{pmatrix} \frac{\pi_{\mathrm{p}}}{\pi_{\mathrm{p}}+\pi_{\mathrm{n}}^2} & \frac{\pi_{\mathrm{n}}^2}{\pi_{\mathrm{p}}+\pi_{\mathrm{n}}^2} \\ \frac{\pi_{\mathrm{p}}^2}{\pi_{\mathrm{p}}^2+\pi_{\mathrm{n}}} & \frac{\pi_{\mathrm{n}}}{\pi_{\mathrm{p}}^2+\pi_{\mathrm{n}}} \end{pmatrix} \begin{pmatrix} \mathcal{P}_{X|Y=\mathrm{p}} \\ \mathcal{P}_{X|Y=\mathrm{n}} \end{pmatrix}.$$

Since

$$\frac{\pi_{\mathrm{p}}}{\pi_{\mathrm{p}}+\pi_{\mathrm{n}}^2}\mathcal{P}_{X|Y=\mathrm{p}} + \frac{\pi_{\mathrm{n}}^2}{\pi_{\mathrm{p}}+\pi_{\mathrm{n}}^2}\mathcal{P}_{X|Y=\mathrm{n}} = \frac{\pi_{\mathrm{p}}\mathcal{P}_{X|Y=\mathrm{p}} + \pi_{\mathrm{n}}^2\mathcal{P}_{X|Y=\mathrm{n}}}{\pi_{\mathrm{p}}+\pi_{\mathrm{n}}^2} = \mathcal{P}_{\mathrm{Sup}}$$

and

$$\frac{\pi_{\mathrm{p}}^2}{\pi_{\mathrm{p}}^2+\pi_{\mathrm{n}}}\mathcal{P}_{X|Y=\mathrm{p}} + \frac{\pi_{\mathrm{n}}}{\pi_{\mathrm{p}}^2+\pi_{\mathrm{n}}}\mathcal{P}_{X|Y=\mathrm{n}} = \frac{\pi_{\mathrm{p}}^2\mathcal{P}_{X|Y=\mathrm{p}} + \pi_{\mathrm{n}}\mathcal{P}_{X|Y=\mathrm{n}}}{\pi_{\mathrm{p}}^2+\pi_{\mathrm{n}}} = \mathcal{P}_{\mathrm{Inf}},$$

the first entry of the resulting vector equals $\mathcal{P}_{\mathrm{Sup}}$ and the second entry of the resulting vector equals $\mathcal{P}_{\mathrm{Inf}}$, which establishes $M_{\mathrm{Pcomp}}B = \bar{P}$. □

**Proof of Lemma 7**

**Lemma 7.** *Let B (46) be the base distributions and*

$$\bar{P} = \begin{pmatrix} \mathcal{P}_{\tilde{\mathrm{D}}} \\ \mathcal{P}_{\mathrm{U}} \end{pmatrix}.$$

*Then, the contamination matrix*

$$M_{\mathrm{DU}} = \begin{pmatrix} 1/2 & 1/2 \\ \pi_{\mathrm{p}} & \pi_{\mathrm{n}} \end{pmatrix}, \tag{53}$$

*which satisfies $\bar{P} = M_{\mathrm{DU}}B$, characterizes the data-generating distributions $\bar{P}$.*

*Proof.* Similar to the proofs of Lemmas 5 and 6, we begin with

$$M_{\mathrm{DU}}B = \begin{pmatrix} 1/2 & 1/2 \\ \pi_{\mathrm{p}} & \pi_{\mathrm{n}} \end{pmatrix} \begin{pmatrix} \mathcal{P}_{X|Y=\mathrm{p}} \\ \mathcal{P}_{X|Y=\mathrm{n}} \end{pmatrix}.$$

Since $\left(\mathcal{P}_{X|Y=\mathrm{p}} + \mathcal{P}_{X|Y=\mathrm{n}}\right)/2 = \mathcal{P}_{\tilde{\mathrm{D}}}$ and $\pi_{\mathrm{p}}\mathcal{P}_{X|Y=\mathrm{p}} + \pi_{\mathrm{n}}\mathcal{P}_{X|Y=\mathrm{n}} = \mathcal{P}_X = \mathcal{P}_{\mathrm{U}}$, we have $M_{\mathrm{DU}}B = \bar{P}$. □

**Proof of Lemma 8**

**Lemma 8.** *Let B (46) be the base distributions and*

$$\bar{P} = \begin{pmatrix} \mathcal{P}_{\tilde{\mathrm{S}}} \\ \mathcal{P}_{\tilde{\mathrm{D}}} \end{pmatrix}.$$

*Then, the contamination matrix*

$$M_{\mathrm{SD}} = \begin{pmatrix} \frac{\pi_{\mathrm{p}}^2}{\pi_{\mathrm{p}}^2+\pi_{\mathrm{n}}^2} & \frac{\pi_{\mathrm{n}}^2}{\pi_{\mathrm{p}}^2+\pi_{\mathrm{n}}^2} \\ 1/2 & 1/2 \end{pmatrix} \tag{54}$$

*which satisfies $\bar{P} = M_{\mathrm{SD}}B$, characterizes the data-generating distributions $\bar{P}$.*

*Proof.* First, we begin with

$$M_{\mathrm{SD}}B = \begin{pmatrix} \frac{\pi_{\mathrm{p}}^2}{\pi_{\mathrm{p}}^2+\pi_{\mathrm{n}}^2} & \frac{\pi_{\mathrm{n}}^2}{\pi_{\mathrm{p}}^2+\pi_{\mathrm{n}}^2} \\ 1/2 & 1/2 \end{pmatrix} \begin{pmatrix} \mathcal{P}_{X|Y=\mathrm{p}} \\ \mathcal{P}_{X|Y=\mathrm{n}} \end{pmatrix}.$$

Then, we have the lemma by reusing the calculations in the proofs of Lemmas 5 and 7. □

### C.2 Omitted Proofs in Section 4.2

**Proof of Lemma 11**

**Lemma 11.** *Let the elements in $\mathcal{S}$ be $\{s_1, s_2, \ldots s_{|\mathcal{S}|}\}$. For each $j \in [|\mathcal{S}|]$, let the $j$-th entry of $\bar{P}$ be*

$$\bar{P}_j = \mathcal{P}_{S=s_j, X} := C(S = s_j, X) \sum_{k \in s_j} \mathcal{P}_{Y=k, X},$$

*which denotes the data-generating distribution of $(s_j, X)$. Assume the base distributions $B$ and the contamination matrix $M_{\mathrm{PPL}}$ are given by (63) and (66), respectively. Then, $M_{\mathrm{PPL}}$ satisfies $\bar{P} = M_{\mathrm{PPL}} B$ and characterizes PPL learning (26).*

*Proof.* For each $j \in [|\mathcal{S}|]$,

$$\left( M_{\mathrm{PPL}} B \right)_j = \sum_{k=1}^{K} C(s_j, X) \mathbb{I}\left[ Y = k \in s_j \right] \mathcal{P}_{Y=k, X} = C(s_j, X) \sum_{k \in s_j} \mathcal{P}_{Y=k, X} = \bar{P}_j.$$

Note that $C(s_j, X) \sum_{k \in s_j} \mathcal{P}_{Y=k, X}$ corresponds to (26) when the observed partial-label is $s_j$. □

**Proof of Lemma 12**

**Lemma 12.** *Let the elements in $\mathcal{S}$ be $\{s_1, s_2, \ldots s_{|\mathcal{S}|}\}$. For each $j \in [|\mathcal{S}|]$, let the $j$-th entry of $\bar{P}$ be*

$$\bar{P}_j = \mathcal{P}_{S=s_j, X} := \frac{1}{2^{K-1} - 1} \sum_{k \in s_j} \mathcal{P}_{Y=k, X},$$

*which denotes the data-generating distribution of $(s_j, X)$. Assume the base distributions $B$ and the contamination matrix $M_{\mathrm{PCPL}}$ are given by (63) and (68), respectively. Then, $M_{\mathrm{PCPL}}$ satisfies $\bar{P} = M_{\mathrm{PCPL}} B$ and characterizes PCPL learning (24).*

*Proof.* For each $j \in [|\mathcal{S}|]$,

$$\left( M_{\mathrm{PCPL}} B \right)_j = \sum_{k=1}^{K} \frac{1}{2^{K-1} - 1} \mathbb{I}\left[ Y = k \in s_j \right] \mathcal{P}_{Y=k, X} = \frac{1}{2^{K-1} - 1} \sum_{k \in s_j} \mathcal{P}_{Y=k, X} = \bar{P}_j.$$

Note that $\frac{1}{2^{K-1}-1} \sum_{k \in s_j} \mathcal{P}_{Y=k, X}$ corresponds to (24) when the observed partial-label is $s_j$. □

**Proof of lemma 13**

**Lemma 13.** *Suppose the base distributions $B$, the contamination matrix $M_{\mathrm{MCL}}$, and the data-generating distributions $\bar{P}$ are given by (63), (71), and (70), respectively. Then, $M_{\mathrm{MCL}}$ satisfies $\bar{P} = M_{\mathrm{MCL}} B$ and characterizes MCL (22).*

*Proof.* For each $j \in [N]$, we have

$$
\begin{aligned}
\left( M_{\mathrm{MCL}} B \right)_j &= \sum_Y \frac{\mathcal{P}_{|\bar{S}|=|\bar{s}_j|}}{\binom{K-1}{|\bar{s}_j|}} \mathbb{I}\left[Y \notin \bar{s}_j\right] \mathcal{P}_{Y,X} \\
&= \sum_Y \frac{\sum_{d=1}^{K-1} \mathbb{I}\left[|\bar{s}_j|=d\right] \mathcal{P}_{|\bar{s}_j|=d}}{\binom{K-1}{|\bar{s}_j|}} \mathbb{I}\left[Y \notin \bar{s}_j\right] \mathcal{P}_{Y,X} \\
&= \sum_{d=1}^{K-1} \frac{\mathcal{P}_{|\bar{s}_j|=d}}{\binom{K-1}{|\bar{s}_j|}} \sum_Y \mathbb{I}\left[Y \notin \bar{s}_j\right] \mathcal{P}_{Y,X} \mathbb{I}\left[|\bar{s}_j|=d\right] \\
&= \sum_{d=1}^{K-1} \frac{\mathcal{P}_{|\bar{s}_j|=d}}{\binom{K-1}{|\bar{s}_j|}} \sum_{Y \notin \bar{s}_j} \mathcal{P}_{Y,X} \mathbb{I}\left[|\bar{s}_j|=d\right] \\
&= \mathcal{P}_{\bar{S}=\bar{s}_j,X}.
\end{aligned}
$$

$\square$

## D  Omitted Proofs in Section 5

Omitted proofs in Section 5 are provided in this appendix. We first restate a claim in the main body of the paper, and then provide the corresponding proof.

### D.1  Omitted Proofs in Section 5.1

**Proof of Corollary 22**

**Corollary 22.** *For PU learning, the classification risk can be rewritten as*

$$
R(g) = \mathbb{E}_{\mathrm{P}}\left[\bar{\ell}_{\mathrm{P}}\right] + \mathbb{E}_{\mathrm{U}}\left[\bar{\ell}_{\mathrm{U}}\right], \tag{90}
$$

*where*

$$
\begin{aligned}
\bar{\ell}_{\mathrm{P}} &= \pi_{\mathrm{p}} \ell_{\mathrm{p}} - \pi_{\mathrm{p}} \ell_{\mathrm{n}}, \\
\bar{\ell}_{\mathrm{U}} &= \ell_{\mathrm{n}}.
\end{aligned} \tag{91}
$$

*Proof.* According to Table 7, $M_{\mathrm{PU}}$ is a child of $M_{\mathrm{UU}}$ on the reduction graph. Thus, replacing the subscripts $\{\mathrm{U}_1, \mathrm{U}_2\}$ of $\bar{P}$ and $\bar{L}$ with $\{\mathrm{P}, \mathrm{U}\}$ and assigning $\gamma_1 = 0$ and $\gamma_2 = \pi_{\mathrm{p}}$ as what we choose in Section 4.1.2, we follow the proof of Theorem 21 to conduct the risk rewrite: We first obtain the corrected losses (91) by plugging the assigned values into (86). Then, repeating the steps in (89), we achieve (90). $\square$

**Proof of Corollary 23**

**Corollary 23.** *Assume $\pi_{\mathrm{p}} \neq 1/2$. For SU learning, the classification risk can be rewritten as*

$$
R(g) = \mathbb{E}_{\tilde{\mathrm{S}}}\left[\bar{\ell}_{\tilde{\mathrm{S}}}\right] + \mathbb{E}_{\mathrm{U}}\left[\bar{\ell}_{\mathrm{U}}\right],
$$

*where*

$$
\begin{aligned}
\bar{\ell}_{\tilde{\mathrm{S}}} &= \frac{\pi_{\mathrm{p}}^2 + \pi_{\mathrm{n}}^2}{2\pi_{\mathrm{p}} - 1} \ell_{\mathrm{p}} - \frac{\pi_{\mathrm{p}}^2 + \pi_{\mathrm{n}}^2}{2\pi_{\mathrm{p}} - 1} \ell_{\mathrm{n}}, \\
\bar{\ell}_{\mathrm{U}} &= -\frac{\pi_{\mathrm{n}}}{2\pi_{\mathrm{p}} - 1} \ell_{\mathrm{p}} + \frac{\pi_{\mathrm{p}}}{2\pi_{\mathrm{p}} - 1} \ell_{\mathrm{n}}.
\end{aligned} \tag{92}
$$

*Proof.* By Table 7, $M_{\mathrm{SU}}$ is a child of $M_{\mathrm{UU}}$. Substituting the subscripts $\{\mathrm{U}_1, \mathrm{U}_2\}$ with subscripts $\{\tilde{\mathrm{S}}, \mathrm{U}\}$ and choosing $\gamma_1 = \frac{\pi_{\mathrm{n}}^2}{\pi_{\mathrm{p}}^2 + \pi_{\mathrm{n}}^2}$ and $\gamma_2 = \pi_{\mathrm{p}}$ as we did in Section 4.1.3, we obtain the corrected losses (92) by plugging the assigned values into (86). We note that $\pi_{\mathrm{p}} \neq 1/2$ ensures the choices of $\gamma_1$ and $\gamma_2$ above satisfy the $\gamma_1 + \gamma_2 \neq 1$ assumption discussed in Section 4.1.1. Then, we achieve the rewrite $R(g) = \mathbb{E}_{\tilde{\mathrm{S}}}\left[\bar{\ell}_{\tilde{\mathrm{S}}}\right] + \mathbb{E}_{\mathrm{U}}\left[\bar{\ell}_{\mathrm{U}}\right]$ by repeating the derivation for (89). $\square$

**Proof of Lemma 25**

**Lemma 25.** *Let $(x, x') \sim \mathcal{P}_{\mathrm{S}}$ defined by (10). Then, $\mathbb{E}_{\mathrm{S}}\left[\frac{\mathcal{L}(X)}{2}\right] = \mathbb{E}_{\mathrm{S}}\left[\frac{\mathcal{L}(X')}{2}\right]$.*

*Proof.* For clarity, we simplify $\mathcal{P}_{\mathrm{S}}$ as $c_1 \mathcal{P}_{X|Y=\mathrm{p}} \mathcal{P}_{X'|Y=\mathrm{p}} + c_2 \mathcal{P}_{X|Y=\mathrm{n}} \mathcal{P}_{X'|Y=\mathrm{n}}$, with $c_1 = \frac{\pi_{\mathrm{p}}^2}{\pi_{\mathrm{p}}^2 + \pi_{\mathrm{n}}^2}$ and $c_2 = \frac{\pi_{\mathrm{n}}^2}{\pi_{\mathrm{p}}^2 + \pi_{\mathrm{n}}^2}$. The lemma follows from

$$
\begin{aligned}
&\mathbb{E}_{\mathrm{S}}\left[\mathcal{L}(X)\right] \\
&= \int_{x \in \mathcal{X}} \int_{x' \in \mathcal{X}} \mathcal{P}_{\mathrm{S}} \mathcal{L}(x) \, \mathrm{d}x' \, \mathrm{d}x \\
&= \int_{x \in \mathcal{X}} \int_{x' \in \mathcal{X}} \left(c_1 \mathcal{P}_{x|Y=\mathrm{p}} \mathcal{P}_{x'|Y=\mathrm{p}} + c_2 \mathcal{P}_{x|Y=\mathrm{n}} \mathcal{P}_{x'|Y=\mathrm{n}}\right) \mathcal{L}(x) \, \mathrm{d}x' \, \mathrm{d}x \\
&= c_1 \int_{x \in \mathcal{X}} \mathcal{P}_{x|Y=\mathrm{p}} \mathcal{L}(x) \, \mathrm{d}x \int_{x' \in \mathcal{X}} \mathcal{P}_{x'|Y=\mathrm{p}} \, \mathrm{d}x' + c_2 \int_{x \in \mathcal{X}} \mathcal{P}_{x|Y=\mathrm{n}} \mathcal{L}(x) \, \mathrm{d}x \int_{x' \in \mathcal{X}} \mathcal{P}_{x'|Y=\mathrm{n}} \, \mathrm{d}x' \\
&= c_1 \int_{x \in \mathcal{X}} \mathcal{P}_{x|Y=\mathrm{p}} \mathcal{L}(x) \, \mathrm{d}x + c_2 \int_{x \in \mathcal{X}} \mathcal{P}_{x|Y=\mathrm{n}} \mathcal{L}(x) \, \mathrm{d}x,
\end{aligned}
$$

and similarly,

$$
\mathbb{E}_{\mathrm{S}}\left[\mathcal{L}(X')\right] = c_1 \int_{x' \in \mathcal{X}} \mathcal{P}_{x'|Y=\mathrm{p}} \mathcal{L}(x') \, \mathrm{d}x' + c_2 \int_{x' \in \mathcal{X}} \mathcal{P}_{x'|Y=\mathrm{n}} \mathcal{L}(x') \, \mathrm{d}x'.
$$

$\square$

**Proof of Corollary 26**

**Corollary 26.** *For Pcomp learning, the classification risk can be rewritten as*

$$
R(g) = \mathbb{E}_{\mathrm{Sup}}\left[\bar{\ell}_{\mathrm{Sup}}\right] + \mathbb{E}_{\mathrm{Inf}}\left[\bar{\ell}_{\mathrm{Inf}}\right],
$$

*where*

$$
\begin{aligned}
\bar{\ell}_{\mathrm{Sup}} &= \ell_{\mathrm{p}} - \pi_{\mathrm{p}} \ell_{\mathrm{n}}, \\
\bar{\ell}_{\mathrm{Inf}} &= -\pi_{\mathrm{n}} \ell_{\mathrm{p}} + \ell_{\mathrm{n}}.
\end{aligned}
\tag{96}
$$

*Proof.* Since $M_{\mathrm{UU}}$ reduces to $M_{\mathrm{Pcomp}}$ with $\gamma_1 = \frac{\pi_{\mathrm{n}}^2}{\pi_{\mathrm{p}} + \pi_{\mathrm{n}}^2}$ and $\gamma_2 = \frac{\pi_{\mathrm{p}}^2}{\pi_{\mathrm{p}}^2 + \pi_{\mathrm{n}}}$ according to Table 7, we replace the subscripts $\{\mathrm{U}_1, \mathrm{U}_2\}$ with $\{\mathrm{Sup}, \mathrm{Inf}\}$ and instantiate (86) with the assigned values to obtain the corrected losses $\bar{\ell}_{\mathrm{Sup}}$ and $\bar{\ell}_{\mathrm{Inf}}$ (96). Then, repeating the same steps in (89), we have the corollary. $\square$

**Proof of Corollary 27**

**Corollary 27.** *Assume $\pi_{\mathrm{p}} \neq 1/2$. For DU learning, the classification risk can be rewritten as*

$$
R(g) = \mathbb{E}_{\tilde{\mathrm{D}}}\left[\bar{\ell}_{\tilde{\mathrm{D}}}\right] + \mathbb{E}_{\mathrm{U}}\left[\bar{\ell}_{\mathrm{U}}\right],
$$

*where*

$$
\begin{aligned}
\bar{\ell}_{\tilde{\mathrm{D}}} &= 2\pi_{\mathrm{p}} \pi_{\mathrm{n}} \left(\frac{1}{\pi_{\mathrm{n}} - \pi_{\mathrm{p}}} \ell_{\mathrm{p}} - \frac{1}{\pi_{\mathrm{n}} - \pi_{\mathrm{p}}} \ell_{\mathrm{n}}\right), \\
\bar{\ell}_{\mathrm{U}} &= -\frac{\pi_{\mathrm{p}}}{\pi_{\mathrm{n}} - \pi_{\mathrm{p}}} \ell_{\mathrm{p}} + \frac{\pi_{\mathrm{n}}}{\pi_{\mathrm{n}} - \pi_{\mathrm{p}}} \ell_{\mathrm{n}}.
\end{aligned}
\tag{98}
$$

*Proof.* By Table 7, $M_{\mathrm{DU}}$ is reduced from $M_{\mathrm{UU}}$. Thus, replacing $\{\mathrm{U}_1, \mathrm{U}_2\}$ with $\{\tilde{\mathrm{D}}, \mathrm{U}\}$, and assigning $\gamma_1 = 1/2$ and $\gamma_2 = \pi_{\mathrm{p}}$, we obtain the corrected losses (98) by plugging the assigned values into (86). Note that $\pi_{\mathrm{p}} \neq 1/2$ implies that the $\gamma_1$ and $\gamma_2$ assignments are feasible. Then, repeating the steps in (89), we have the corollary. $\square$

**Proof of Lemma 28**

**Lemma 28.** *Given B (46) and following the DU learning notations, we have*

$$M'_{\mathrm{DU}}B = \begin{pmatrix} \mathcal{P}_{\tilde{\mathrm{D}}} \\ \mathcal{P}_{\mathrm{U}} \end{pmatrix} = \bar{P},$$

*where*

$$M'_{\mathrm{DU}} := \begin{pmatrix} \frac{\int_{x' \in \mathcal{X}} \mathcal{P}_{x'|Y=\mathrm{n}} \mathrm{d}x'}{2} & \frac{\int_{x' \in \mathcal{X}} \mathcal{P}_{x'|Y=\mathrm{p}} \mathrm{d}x'}{2} \\ \pi_{\mathrm{p}} & \pi_{\mathrm{n}} \end{pmatrix}.$$

*Proof.* Since $\int_{x' \in \mathcal{X}} \mathcal{P}_{x'|Y=\mathrm{n}} \mathrm{d}x' = 1$ and $\int_{x' \in \mathcal{X}} \mathcal{P}_{x'|Y=\mathrm{p}} \mathrm{d}x' = 1$, we have $M'_{\mathrm{DU}} = M_{\mathrm{DU}}$ and hence $M'_{\mathrm{DU}}B = M_{\mathrm{DU}}B = \begin{pmatrix} \mathcal{P}_{\tilde{\mathrm{D}}} \\ \mathcal{P}_{\mathrm{U}} \end{pmatrix}$. The last equality follows from Lemma 7. $\qquad\square$

**Proof of Lemma 29**

**Lemma 29.** *Let $(x, x') \sim \mathcal{P}_{\mathrm{D}}$ defined in (12). Then, $\mathbb{E}_{\mathrm{D}}\left[\frac{\mathcal{L}(X)}{2}\right] = \mathbb{E}_{\mathrm{D}}\left[\frac{\mathcal{L}(X')}{2}\right]$.*

*Proof.* Recall $\mathcal{P}_{\mathrm{D}} = \frac{1}{2}(\mathcal{P}_{x|Y=\mathrm{p}}\mathcal{P}_{x'|Y=\mathrm{n}} + \mathcal{P}_{x|Y=\mathrm{n}}\mathcal{P}_{x'|Y=\mathrm{p}})$. Following the similar argument in Lemma 25,

$$\begin{aligned} \mathbb{E}_{\mathrm{D}}\left[\frac{\mathcal{L}(X)}{2}\right] &= \int_{x \in \mathcal{X}} \int_{x' \in \mathcal{X}} \left(\mathcal{P}_{x|Y=\mathrm{p}}\mathcal{P}_{x'|Y=\mathrm{n}} + \mathcal{P}_{x|Y=\mathrm{n}}\mathcal{P}_{x'|Y=\mathrm{p}}\right) \frac{\mathcal{L}(x)}{4} \, \mathrm{d}x'\mathrm{d}x \\ &= \int_{x \in \mathcal{X}} \left(\mathcal{P}_{x|Y=\mathrm{p}} + \mathcal{P}_{x|Y=\mathrm{n}}\right) \frac{\mathcal{L}(x)}{4} \, \mathrm{d}x \end{aligned}$$

and

$$\mathbb{E}_{\mathrm{D}}\left[\frac{\mathcal{L}(X')}{2}\right] = \int_{x' \in \mathcal{X}} \left(\mathcal{P}_{x'|Y=\mathrm{n}} + \mathcal{P}_{x'|Y=\mathrm{p}}\right) \frac{\mathcal{L}(x')}{4} \, \mathrm{d}x'$$

prove the lemma. $\qquad\square$

**Proof of Lemma 30**

**Corollary 30.** *Assume $\pi_{\mathrm{p}} \neq 1/2$. For SD learning, the classification risk can be rewritten as*

$$R(g) = \mathbb{E}_{\tilde{\mathrm{S}}}\left[\bar{\ell}_{\tilde{\mathrm{S}}}\right] + \mathbb{E}_{\tilde{\mathrm{D}}}\left[\bar{\ell}_{\tilde{\mathrm{D}}}\right],$$

*where*

$$\begin{aligned} \bar{\ell}_{\tilde{\mathrm{S}}} &= \left(\pi_{\mathrm{p}}^2 + \pi_{\mathrm{n}}^2\right)\left(\frac{\pi_{\mathrm{p}}}{\pi_{\mathrm{p}} - \pi_{\mathrm{n}}}\ell_{\mathrm{p}} - \frac{\pi_{\mathrm{n}}}{\pi_{\mathrm{p}} - \pi_{\mathrm{n}}}\ell_{\mathrm{n}}\right), \\ \bar{\ell}_{\tilde{\mathrm{D}}} &= 2\pi_{\mathrm{p}}\pi_{\mathrm{n}}\left(-\frac{\pi_{\mathrm{n}}}{\pi_{\mathrm{p}} - \pi_{\mathrm{n}}}\ell_{\mathrm{p}} + \frac{\pi_{\mathrm{p}}}{\pi_{\mathrm{p}} - \pi_{\mathrm{n}}}\ell_{\mathrm{n}}\right). \end{aligned} \qquad (101)$$

*Proof.* By Table 7, $M_{\mathrm{SD}}$ is reduced from $M_{\mathrm{UU}}$. Thus, replacing $\{\mathrm{U}_1, \mathrm{U}_2\}$ with $\{\tilde{\mathrm{S}}, \tilde{\mathrm{D}}\}$, and assigning $\gamma_1 = \frac{\pi_{\mathrm{n}}^2}{\pi_{\mathrm{p}}^2 + \pi_{\mathrm{n}}^2}$ and $\gamma_2 = 1/2$, we obtain the corrected losses (101) by plugging the assigned values into (86). Note that $\pi_{\mathrm{p}} \neq 1/2$ implies that the $\gamma_1$ and $\gamma_2$ assignments are feasible. Then, repeating the steps in (89), we have the corollary. $\qquad\square$

**Proof of Lemma 31**

**Lemma 31.** *Given $B$ (46) and following the SD learning notations, we have*

$$M'_{\mathrm{SD}}B = \begin{pmatrix} \mathcal{P}_{\tilde{\mathrm{S}}} \\ \mathcal{P}_{\tilde{\mathrm{D}}} \end{pmatrix} = \bar{P},$$

*where*

$$M'_{\mathrm{SD}} := \begin{pmatrix} \dfrac{\pi_{\mathrm{p}}^2 \int_{x' \in \mathcal{X}} \mathcal{P}_{x'|Y=\mathrm{p}} \mathrm{d}x'}{\pi_{\mathrm{p}}^2 + \pi_{\mathrm{n}}^2} & \dfrac{\pi_{\mathrm{n}}^2 \int_{x' \in \mathcal{X}} \mathcal{P}_{x'|Y=\mathrm{n}} \mathrm{d}x'}{\pi_{\mathrm{p}}^2 + \pi_{\mathrm{n}}^2} \\ \dfrac{\int_{x' \in \mathcal{X}} \mathcal{P}_{x'|Y=\mathrm{n}} \mathrm{d}x'}{2} & \dfrac{\int_{x' \in \mathcal{X}} \mathcal{P}_{x'|Y=\mathrm{p}} \mathrm{d}x'}{2} \end{pmatrix}.$$

*Proof.* Since $\int_{x' \in \mathcal{X}} \mathcal{P}_{x'|Y=\mathrm{p}} \mathrm{d}x' = 1$ and $\int_{x' \in \mathcal{X}} \mathcal{P}_{x'|Y=\mathrm{n}} \mathrm{d}x' = 1$, we have $M'_{\mathrm{SD}} = M_{\mathrm{SD}}$ and hence $M'_{\mathrm{SD}}B = M_{\mathrm{SD}}B = \begin{pmatrix} \mathcal{P}_{\tilde{\mathrm{S}}} \\ \mathcal{P}_{\tilde{\mathrm{D}}} \end{pmatrix}$. The last equality follows from Lemma 8. $\qquad\square$

**Proof of Lemma 32**

**Lemma 32.** *Assume the formulation $\bar{P} = M_{\mathrm{Sconf}}B$ (58) is given. Suppose a vector of corrected losses $\bar{L}^{\top}$ of the form $\begin{pmatrix} \tilde{\ell}_1(x) & \tilde{\ell}_2(x) \end{pmatrix}$ is independent of $x'$. Then, we have*

$$\int_{x' \in \mathcal{X}} \bar{L}^{\top} \bar{P} \, \mathrm{d}x' = \bar{L}^{\top} \tilde{M}_{\mathrm{Sconf}} P, \tag{103}$$

*where*

$$\tilde{M}_{\mathrm{Sconf}} = \begin{pmatrix} \int_{x'} \dfrac{\pi_{\mathrm{p}}^2 \mathcal{P}_{x'|\mathrm{p}} - \pi_{\mathrm{n}}^2 \mathcal{P}_{x'|\mathrm{n}}}{r - \pi_{\mathrm{n}}} \mathrm{d}x' & \int_{x'} \dfrac{\pi_{\mathrm{n}}^2 \mathcal{P}_{x'|\mathrm{n}} - \pi_{\mathrm{n}}^2 \mathcal{P}_{x'|\mathrm{p}}}{r - \pi_{\mathrm{n}}} \mathrm{d}x' \\ \int_{x'} \dfrac{\pi_{\mathrm{p}}^2 \mathcal{P}_{x'|\mathrm{n}} - \pi_{\mathrm{p}}^2 \mathcal{P}_{x'|\mathrm{p}}}{\pi_{\mathrm{p}} - r} \mathrm{d}x' & \int_{x'} \dfrac{\pi_{\mathrm{p}}^2 \mathcal{P}_{x'|\mathrm{p}} - \pi_{\mathrm{n}}^2 \mathcal{P}_{x'|\mathrm{n}}}{\pi_{\mathrm{p}} - r} \mathrm{d}x' \end{pmatrix}.$$

*Proof.* **of Lemma 32.** We replace $\bar{P}$ using $\bar{P} = M_{\mathrm{Sconf}}B$ (58). Since $\bar{L}$ is independent of $x'$, we can move the integral over $x'$ into $M_{\mathrm{Sconf}}$ to obtain

$$
\begin{aligned}
\int_{x' \in \mathcal{X}} \bar{L}^{\top} \bar{P} \, \mathrm{d}x' &= \int_{x'} \bar{L}^{\top} \begin{pmatrix} \dfrac{\pi_{\mathrm{p}}(\pi_{\mathrm{p}}^2 \mathcal{P}_{x'|\mathrm{p}} - \pi_{\mathrm{n}}^2 \mathcal{P}_{x'|\mathrm{n}})}{r - \pi_{\mathrm{n}}} & \dfrac{\pi_{\mathrm{p}}(\pi_{\mathrm{n}}^2 \mathcal{P}_{x'|\mathrm{n}} - \pi_{\mathrm{n}}^2 \mathcal{P}_{x'|\mathrm{p}})}{r - \pi_{\mathrm{n}}} \\ \dfrac{\pi_{\mathrm{n}}(\pi_{\mathrm{p}}^2 \mathcal{P}_{x'|\mathrm{n}} - \pi_{\mathrm{p}}^2 \mathcal{P}_{x'|\mathrm{p}})}{\pi_{\mathrm{p}} - r} & \dfrac{\pi_{\mathrm{n}}(\pi_{\mathrm{p}}^2 \mathcal{P}_{x'|\mathrm{p}} - \pi_{\mathrm{n}}^2 \mathcal{P}_{x'|\mathrm{n}})}{\pi_{\mathrm{p}} - r} \end{pmatrix} \begin{pmatrix} \mathcal{P}_{X|\mathrm{p}} \\ \mathcal{P}_{X|\mathrm{n}} \end{pmatrix} \mathrm{d}x' \\
&= \bar{L}^{\top} \begin{pmatrix} \int_{x'} \dfrac{\pi_{\mathrm{p}}^2 \mathcal{P}_{x'|\mathrm{p}} - \pi_{\mathrm{n}}^2 \mathcal{P}_{x'|\mathrm{n}}}{r - \pi_{\mathrm{n}}} \mathrm{d}x' & \int_{x'} \dfrac{\pi_{\mathrm{n}}^2 \mathcal{P}_{x'|\mathrm{n}} - \pi_{\mathrm{n}}^2 \mathcal{P}_{x'|\mathrm{p}}}{r - \pi_{\mathrm{n}}} \mathrm{d}x' \\ \int_{x'} \dfrac{\pi_{\mathrm{p}}^2 \mathcal{P}_{x'|\mathrm{n}} - \pi_{\mathrm{p}}^2 \mathcal{P}_{x'|\mathrm{p}}}{\pi_{\mathrm{p}} - r} \mathrm{d}x' & \int_{x'} \dfrac{\pi_{\mathrm{p}}^2 \mathcal{P}_{x'|\mathrm{p}} - \pi_{\mathrm{n}}^2 \mathcal{P}_{x'|\mathrm{n}}}{\pi_{\mathrm{p}} - r} \mathrm{d}x' \end{pmatrix} \begin{pmatrix} \pi_{\mathrm{p}} \mathcal{P}_{X|\mathrm{p}} \\ \pi_{\mathrm{n}} \mathcal{P}_{X|\mathrm{n}} \end{pmatrix}.
\end{aligned}
$$

Since $\pi_{\mathrm{p}} \mathcal{P}_{X|\mathrm{p}} = \mathcal{P}_{X,Y=\mathrm{p}}$ and $\pi_{\mathrm{n}} \mathcal{P}_{X|\mathrm{n}} = \mathcal{P}_{X,Y=\mathrm{n}}$, $\begin{pmatrix} \pi_{\mathrm{p}} \mathcal{P}_{X|\mathrm{p}} \\ \pi_{\mathrm{n}} \mathcal{P}_{X|\mathrm{n}} \end{pmatrix} = P$. Furthermore, comparing the equality in the above derivation with (103), we have

$$\tilde{M}_{\mathrm{Sconf}} = \begin{pmatrix} \int_{x'} \dfrac{\pi_{\mathrm{p}}^2 \mathcal{P}_{x'|\mathrm{p}} - \pi_{\mathrm{n}}^2 \mathcal{P}_{x'|\mathrm{n}}}{r - \pi_{\mathrm{n}}} \mathrm{d}x' & \int_{x'} \dfrac{\pi_{\mathrm{n}}^2 \mathcal{P}_{x'|\mathrm{n}} - \pi_{\mathrm{n}}^2 \mathcal{P}_{x'|\mathrm{p}}}{r - \pi_{\mathrm{n}}} \mathrm{d}x' \\ \int_{x'} \dfrac{\pi_{\mathrm{p}}^2 \mathcal{P}_{x'|\mathrm{n}} - \pi_{\mathrm{p}}^2 \mathcal{P}_{x'|\mathrm{p}}}{\pi_{\mathrm{p}} - r} \mathrm{d}x' & \int_{x'} \dfrac{\pi_{\mathrm{p}}^2 \mathcal{P}_{x'|\mathrm{p}} - \pi_{\mathrm{n}}^2 \mathcal{P}_{x'|\mathrm{n}}}{\pi_{\mathrm{p}} - r} \mathrm{d}x' \end{pmatrix}$$

that completes the proof. $\qquad\square$

**Proof of Lemma 33**

**Lemma 33.** *Let*

$$\tilde{M}_{\text{Sconf}}^{\dagger} := \begin{pmatrix} \frac{r - \pi_{\text{n}}}{\pi_{\text{p}} - \pi_{\text{n}}} & 0 \\ 0 & \frac{\pi_{\text{p}} - r}{\pi_{\text{p}} - \pi_{\text{n}}} \end{pmatrix}.$$

*Then,*

$$\tilde{M}_{\text{Sconf}}^{\dagger} \tilde{M}_{\text{Sconf}} = I.$$

*Proof.* We prove the lemma by examining each entry of $\tilde{M}_{\text{Sconf}}^{\dagger} \tilde{M}_{\text{Sconf}}$. The value of $(1,1)$ entry is

$$
\begin{aligned}
\frac{r - \pi_{\text{n}}}{\pi_{\text{p}} - \pi_{\text{n}}} \int_{x'} \frac{\pi_{\text{p}}^2 \mathcal{P}_{x'|\text{p}} - \pi_{\text{n}}^2 \mathcal{P}_{x'|\text{n}}}{r - \pi_{\text{n}}} \mathrm{d}x' &= \frac{1}{\pi_{\text{p}} - \pi_{\text{n}}} \left( \pi_{\text{p}}^2 \int_{x'} \mathcal{P}_{x'|\text{p}} \mathrm{d}x' - \pi_{\text{n}}^2 \int_{x'} \mathcal{P}_{x'|\text{n}} \mathrm{d}x' \right) \\
&= \frac{\pi_{\text{p}}^2 - \pi_{\text{n}}^2}{\pi_{\text{p}} - \pi_{\text{n}}} = 1.
\end{aligned}
$$

The $(2,2)$ entry has value

$$
\begin{aligned}
\frac{\pi_{\text{p}} - r}{\pi_{\text{p}} - \pi_{\text{n}}} \int_{x'} \frac{\pi_{\text{p}}^2 \mathcal{P}_{x'|\text{p}} - \pi_{\text{n}}^2 \mathcal{P}_{x'|\text{n}}}{\pi_{\text{p}} - r} \mathrm{d}x' &= \frac{1}{\pi_{\text{p}} - \pi_{\text{n}}} \left( \pi_{\text{p}}^2 \int_{x'} \mathcal{P}_{x'|\text{p}} \mathrm{d}x' - \pi_{\text{n}}^2 \int_{x'} \mathcal{P}_{x'|\text{n}} \mathrm{d}x' \right) \\
&= \frac{\pi_{\text{p}}^2 - \pi_{\text{n}}^2}{\pi_{\text{p}} - \pi_{\text{n}}} = 1.
\end{aligned}
$$

The $(1,2)$ entry is zero since $\int_{x'} \left( \pi_{\text{n}}^2 \mathcal{P}_{x'|\text{n}} - \pi_{\text{n}}^2 \mathcal{P}_{x'|\text{p}} \right) \mathrm{d}x' = 0$. Similarly, since $\int_{x'} \left( \pi_{\text{p}}^2 \mathcal{P}_{x'|\text{n}} - \pi_{\text{p}}^2 \mathcal{P}_{x'|\text{p}} \right) \mathrm{d}x' = 0$, the $(2,1)$ entry is also zero. □

## D.2 Omitted Proofs in Section 5.2

**Proof of Corollary 38**

**Corollary 38.** *The decontamination matrix $M_{\text{PCPL}}^{\dagger}$ for PCPL equals $M_{\text{PPL}}^{\dagger}$. If we define the corrected losses as $\bar{L}^{\top} := L^{\top} M_{\text{PCPL}}^{\dagger}$, the classification risk for PCPL learning can be rewritten as*

$$R(g) = \mathbb{E}_{S,X} \left[ \bar{\ell}_S \right],$$

*where*

$$\bar{\ell}_S = \sum_{i \in S} \frac{\mathcal{P}_{Y=i|X}}{\sum_{a \in S} \mathcal{P}_{Y=a|X}} \ell_{Y=i}. \tag{110}$$

*Proof.* The proof follows the standard argument: First, find out $M_{\text{PCPL}}^{\dagger}$, then construct the corrected losses to rewrite the risk.

Since $M_{\text{PCPL}}$ is reduced from $M_{\text{PPL}}$, we can exploit Lemma 36. Note that the only difference $C(S, X)$ between the formulations of PCPL and PPL cancels itself out in the derivation of $\mathcal{P}_{Y=i|S=s_j,X}$ (please refer to the proof of Lemma 36 for a detailed derivation), the $(i,j)$ entry of $M_{\text{PCPL}}^{\dagger}$ coincides with that of $M_{\text{PPL}}^{\dagger}$ for all $i$ and $j$, proving the first assertion.

Since $M_{\text{PPL}}^{\dagger}$ and $M_{\text{PCPL}}^{\dagger}$ are identical, $L^{\top} M_{\text{PPL}}^{\dagger} = L^{\top} M_{\text{PCPL}}^{\dagger}$ gives (110):

$$
\begin{aligned}
\bar{\ell}_{S=s_j} = \left( L^{\top} M_{\text{PCPL}}^{\dagger} \right)_j &= \sum_{i=1}^{K} \frac{\mathcal{P}_{Y=i|X} \mathbb{I}\left[ Y = i \in s_j \right]}{\sum_{a \in s_j} \mathcal{P}_{Y=a|X}} \ell_{Y=i} \\
&= \sum_{i \in s_j} \frac{\mathcal{P}_{Y=i|X}}{\sum_{a \in s_j} \mathcal{P}_{Y=a|X}} \ell_{Y=i}.
\end{aligned}
$$

Being identical to $M_{\mathrm{PPL}}^{\dagger}$ also means that $M_{\mathrm{PCPL}}^{\dagger}$ is derived from $M_{\mathrm{gCCN}}^{\dagger}$. Thus, we can continue (107) to rewrite the risk by repeating the proof of Corollary 37:

$$R(g) = \int_{x \in \mathcal{X}} \bar{L}^{\top} \bar{P} \mathrm{d}x = \int_{x \in \mathcal{X}} \sum_{j=1}^{|\mathcal{S}|} \mathcal{P}_{S=s_j,x} \bar{\ell}_{S=s_j} \mathrm{d}x = \mathbb{E}_{S,X} \left[ \bar{\ell}_S \right].$$

$\square$

**Proof of Lemma 39**

**Lemma 39.** *Let $(s, s')$ be a pair of partial-labels satisfying $s = \mathcal{Y} \backslash s'$. Then,*

$$\mathcal{P}_{S=s,X} \bar{\ell}_{S=s} + \mathcal{P}_{S=s',X} \bar{\ell}_{S=s'} = \mathcal{P}_{S=s,X} \sum_{i=1}^{K} \frac{\mathcal{P}_{Y=i|X} \ell_{Y=i}}{\sum_{a \in s} \mathcal{P}_{Y=a|X}}.$$

*Proof.* Given $M_{\mathrm{PCPL}}$ (68), we apply $\bar{P} = M_{\mathrm{PCPL}} B$ to obtain

$$\mathcal{P}_{S=s',X} = \frac{\sum_{k=1}^{K} \mathbb{I}\left[Y = k \in s'\right] \mathcal{P}_{Y=k,X}}{2^{K-1} - 1}.$$

We also have

$$\bar{\ell}_{S=s'} = \frac{\sum_{i \in s'} \mathcal{P}_{Y=i|X} \ell_{Y=i}}{\sum_{a \in s'} \mathcal{P}_{Y=a|X}}$$

according to (110). Since

$$\frac{\sum_{k=1}^{K} \mathbb{I}\left[Y = k \in s'\right] \mathcal{P}_{Y=k,X}}{\sum_{a \in s'} \mathcal{P}_{Y=a|X}} = \mathcal{P}_X = \frac{\sum_{k=1}^{K} \mathbb{I}\left[Y = k \in s\right] \mathcal{P}_{Y=k,X}}{\sum_{a \in s} \mathcal{P}_{Y=a|X}},$$

$$\begin{aligned}
\mathcal{P}_{S=s',X} \bar{\ell}_{S=s'} &= \frac{\sum_{k=1}^{K} \mathbb{I}\left[Y = k \in s'\right] \mathcal{P}_{Y=k,X}}{2^{K-1} - 1} \frac{\sum_{i \in s'} \mathcal{P}_{Y=i|X} \ell_{Y=i}}{\sum_{a \in s'} \mathcal{P}_{Y=a|X}} \\
&= \frac{\sum_{k=1}^{K} \mathbb{I}\left[Y = k \in s\right] \mathcal{P}_{Y=k,X}}{2^{K-1} - 1} \frac{\sum_{i \in s'} \mathcal{P}_{Y=i|X} \ell_{Y=i}}{\sum_{a \in s} \mathcal{P}_{Y=a|X}}.
\end{aligned}$$

Thus,

$$\begin{aligned}
\mathcal{P}_{S=s,X} \bar{\ell}_{S=s} + \mathcal{P}_{S=s',X} \bar{\ell}_{S=s'} &= \frac{\sum_{k=1}^{K} \mathbb{I}\left[Y = k \in s\right] \mathcal{P}_{Y=k,X}}{2^{K-1} - 1} \frac{\sum_{i \in s} \mathcal{P}_{Y=i|X} \ell_{Y=i}}{\sum_{a \in s} \mathcal{P}_{Y=a|X}} \\
&\quad + \frac{\sum_{k=1}^{K} \mathbb{I}\left[Y = k \in s\right] \mathcal{P}_{Y=k,X}}{2^{K-1} - 1} \frac{\sum_{i \in s'} \mathcal{P}_{Y=i|X} \ell_{Y=i}}{\sum_{a \in s} \mathcal{P}_{Y=a|X}} \\
&= \mathcal{P}_{S=s,X} \frac{\sum_{i \in s} \mathcal{P}_{Y=i|X} \ell_{Y=i} + \sum_{i \in s'} \mathcal{P}_{Y=i|X} \ell_{Y=i}}{\sum_{a \in s} \mathcal{P}_{Y=a|X}} \\
&= \mathcal{P}_{S=s,X} \sum_{i=1}^{K} \frac{\mathcal{P}_{Y=i|X} \ell_{Y=i}}{\sum_{a \in s} \mathcal{P}_{Y=a|X}}
\end{aligned}$$

proves the lemma. $\square$

**Proof of Corollary 40**

**Corollary 40.** *The $(i, j)$ entry of the decontamination matrix $M_{\mathrm{MCL}}^{\dagger}$ is of the form*

$$\mathcal{P}_{Y=i|\bar{S}=\bar{s}_j,X} = \frac{\mathcal{P}_{Y=i|X}\mathbb{I}\left[Y=i \notin \bar{s}_j\right]}{\sum_{a \notin \bar{s}_j} \mathcal{P}_{Y=a|X}}. \tag{111}$$

*Define the corrected losses $\bar{L}^{\top} := L^{\top} M_{\mathrm{MCL}}^{\dagger}$. Then, for MCL learning, the classification risk can be rewritten as*

$$R(g) = \mathbb{E}_{\bar{S},X}\left[\bar{\ell}_{\bar{S}}\right],$$

*where*

$$\bar{\ell}_{\bar{S}} = \sum_{i \notin \bar{S}} \frac{\mathcal{P}_{Y=i|X}}{\sum_{a \notin \bar{S}} \mathcal{P}_{Y=a|X}} \ell_{Y=i}. \tag{112}$$

*Proof.* The proof follows the standard strategy in Section 5.2: We will first find out $M_{\mathrm{MCL}}^{\dagger}$, and then construct the corrected losses $\bar{L}$ to rewrite the risk.

Based on the notion in (71), we denote the $(j, i)$ entry of $M_{\mathrm{MCL}}$ as

$$\frac{\mathcal{P}_{|\bar{S}|=|\bar{s}_j|}}{\binom{K-1}{|\bar{s}_j|}}\mathbb{I}\left[Y=i \notin \bar{s}_j\right] = C(\bar{s}_j, X)\mathbb{I}\left[Y=i \notin \bar{s}_j\right] = \mathcal{P}_{\bar{S}=\bar{s}_j|Y=i,X}. \tag{128}$$

Expressing $M_{\mathrm{MCL}}$ via (128) allows us to apply the argument for (106) to show that the $(i, j)$ entry of $M_{\mathrm{MCL}}^{\dagger}$ is of the form $\mathcal{P}_{Y=i|\bar{S}=\bar{s}_j,X}$. Specifically, assigning $(M)_{j,i}$ in (40) as $\mathcal{P}_{\bar{S}=\bar{s}_j|Y=i,X}$ and applying marginal chain (i.e., Proposition 2), we obtain

$$\begin{aligned}
\left(M_{\mathrm{MCL}}^{\dagger} M_{\mathrm{MCL}} P\right)_i &= \sum_{j=1}^{|\mathcal{S}|} \left(M_{\mathrm{MCL}}^{\dagger}\right)_{i,j} \sum_{k=1}^{K} (M_{\mathrm{MCL}})_{j,k} P_k \\
&= \sum_{j=1}^{|\mathcal{S}|} \mathcal{P}_{Y=i|\bar{S}=\bar{s}_j,X} \sum_{k=1}^{K} \mathcal{P}_{\bar{S}=\bar{s}_j|Y=k,X}\mathcal{P}_{Y=k,X} \\
&= \mathcal{P}_{Y=i,X} = P_i.
\end{aligned}$$

Then, we follow the same argument in Lemma 36 to calculate $\mathcal{P}_{Y=i|\bar{S}=\bar{s}_j,X}$ subject to (128). Note that $\mathcal{P}_{\bar{S}|Y,X} = C(\bar{S}, X)\mathbb{I}\left[Y \notin \bar{S}\right]$ in (128) implies

$$\begin{aligned}
\sum_{b \in \bar{S}} \mathcal{P}_{\bar{S},Y=b|X} &= \sum_{b \in \bar{S}} \mathcal{P}_{\bar{S}|Y=b,X}\mathcal{P}_{Y=b|X} \\
&= \sum_{b \in \bar{S}} C(\bar{S}, X)\mathbb{I}\left[Y=b \notin \bar{S}\right]\mathcal{P}_{Y=b|X} \\
&= 0.
\end{aligned}$$

Thus, $\mathcal{P}_{\bar{S}|X} = \sum_{b \in \bar{S}} \mathcal{P}_{\bar{S},Y=b|X} + \sum_{a \notin \bar{S}} \mathcal{P}_{\bar{S},Y=a|X} = \sum_{a \notin \bar{S}} \mathcal{P}_{\bar{S},Y=a|X}$. The fact further implies

$$\begin{aligned}
\mathcal{P}_{Y|\bar{S},X} = \frac{\mathcal{P}_{\bar{S},Y|X}}{\mathcal{P}_{\bar{S}|X}} &= \frac{\mathcal{P}_{\bar{S}|Y,X}\mathcal{P}_{Y|X}}{\sum_{a \notin \bar{S}} \mathcal{P}_{\bar{S}|Y=a,X}\mathcal{P}_{Y=a|X}} \\
&= \frac{C(\bar{S}, X)\mathbb{I}\left[Y \notin \bar{S}\right]\mathcal{P}_{Y|X}}{\sum_{a \notin \bar{S}} C(\bar{S}, X)\mathbb{I}\left[Y=a \notin \bar{S}\right]\mathcal{P}_{Y=a|X}} \\
&= \frac{\mathcal{P}_{Y|X}\mathbb{I}\left[Y \notin \bar{S}\right]}{\sum_{a \notin \bar{S}} \mathcal{P}_{Y=a|X}}.
\end{aligned}$$

Therefore, for $Y = i$ and $\bar{S} = \bar{s}_j$, we achieve

$$\mathcal{P}_{Y=i|\bar{S}=\bar{s}_j,X} = \frac{\mathcal{P}_{Y=i|X}\mathbb{I}\left[Y = i \notin \bar{s}_j\right]}{\sum_{a \notin \bar{s}_j} \mathcal{P}_{Y=a|X}}$$

that proves (111).

With $M_{\mathrm{MCL}}^\dagger$ in hand, we repeat the same argument in Corollary 37 to obtain

$$
\begin{aligned}
\bar{\ell}_{\bar{S}=\bar{s}_j} = \left(L^\top M_{\mathrm{MCL}}^\dagger\right)_j &= \sum_{i=1}^{K} \frac{\mathcal{P}_{Y=i|X}\mathbb{I}\left[Y = i \notin \bar{s}_j\right]}{\sum_{a \notin \bar{s}_j} \mathcal{P}_{Y=a|X}}\ell_{Y=i} \\
&= \sum_{i \notin \bar{s}_j} \frac{\mathcal{P}_{Y=i|X}}{\sum_{a \notin \bar{s}_j} \mathcal{P}_{Y=a|X}}\ell_{Y=i}
\end{aligned}
$$

and

$$
\begin{aligned}
R(g) &= \int_{x \in \mathcal{X}} L^\top P\,\mathrm{d}x = \int_{x \in \mathcal{X}} L^\top M_{\mathrm{MCL}}^\dagger M_{\mathrm{MCL}} P\,\mathrm{d}x \\
&= \int_{x \in \mathcal{X}} \bar{L}^\top \bar{P}\,\mathrm{d}x = \int_{x \in \mathcal{X}} \sum_{j=1}^{|\mathcal{S}|} \mathcal{P}_{\bar{S}=\bar{s}_j,x} \bar{\ell}_{\bar{S}=\bar{s}_j}\,\mathrm{d}x = \mathbb{E}_{\bar{S},X}\left[\bar{\ell}_{\bar{S}}\right]
\end{aligned}
$$

to complete the risk rewrite of MCL. $\qquad\square$

### D.3 Omitted Proofs in Section 5.3

**Proof of Corollary 47**

**Corollary 47.** *For SC-Conf learning, the classification risk can be written as*

$$R(g) = \pi_{y_{\mathrm{s}}}\mathbb{E}_{X|Y=y_{\mathrm{s}}}\left[\sum_{i=1}^{K} \frac{r_i(X)}{r_{y_{\mathrm{s}}}(X)}\ell_i\right].$$

*Proof.* The corollary follows from notation substitution and the same argument for Theorem 46. Specifically, we replace $M_{\mathrm{Sub}}$ with $M_{\mathrm{SC}}$, $\mathcal{Y}_{\mathrm{s}}$ with $y_{\mathrm{s}}$, and $\frac{\mathcal{P}_{Y=i|X}}{\mathcal{P}_{Y\in\mathcal{Y}_{\mathrm{s}}|X}}\ell_i$ with $\frac{\mathcal{P}_{Y=i|X}}{\mathcal{P}_{Y=y_{\mathrm{s}}|X}}\ell_i$. $\qquad\square$

**Proof of Corollary 48**

**Corollary 48.** *For Pconf learning, the classification risk can be written as*

$$R(g) = \pi_{\mathrm{p}}\mathbb{E}_{\mathrm{P}}\left[\ell_{\mathrm{p}} + \frac{1 - r(X)}{r(X)}\ell_{\mathrm{n}}\right].$$

*Proof.* Since

$$M_{\mathrm{Pconf}}^\dagger M_{\mathrm{Pconf}} = \begin{pmatrix} \frac{\mathcal{P}_{Y=\mathrm{p}|X}}{\mathcal{P}_{Y=\mathrm{p}|X}} & 0 \\ 0 & \frac{\mathcal{P}_{Y=\mathrm{p}|X}}{\mathcal{P}_{Y=\mathrm{n}|X}} \end{pmatrix} \begin{pmatrix} \frac{\mathcal{P}_{Y=\mathrm{p}|X}}{\mathcal{P}_{Y=\mathrm{p}|X}} & 0 \\ 0 & \frac{\mathcal{P}_{Y=\mathrm{n}|X}}{\mathcal{P}_{Y=\mathrm{p}|X}} \end{pmatrix} = I,$$

we define $\bar{L}^\top := L^\top M_{\mathrm{Pconf}}^\dagger$ and apply (39) to rewrite the risk as follows

$$
\begin{aligned}
R(g) &= \int_{x \in \mathcal{X}} L^\top P\,\mathrm{d}x = \int_{x \in \mathcal{X}} \bar{L}^\top \bar{P}\,\mathrm{d}x = \int_{x \in \mathcal{X}} \begin{pmatrix} \ell_{\mathrm{p}} & \frac{1-r(X)}{r(X)}\ell_{\mathrm{n}} \end{pmatrix} \begin{pmatrix} \mathcal{P}_{Y=\mathrm{p},X} \\ \mathcal{P}_{Y=\mathrm{p},X} \end{pmatrix}\,\mathrm{d}x \\
&= \mathcal{P}_{Y=\mathrm{p}}\mathbb{E}_{X|Y=\mathrm{p}}\left[\ell_{\mathrm{p}} + \frac{1-r(X)}{r(X)}\ell_{\mathrm{n}}\right] \\
&= \pi_{\mathrm{p}}\mathbb{E}_{\mathrm{P}}\left[\ell_{\mathrm{p}} + \frac{1-r(X)}{r(X)}\ell_{\mathrm{n}}\right].
\end{aligned}
$$

$\square$

**Proof of Corollary 49**

**Corollary 49.** *For soft-label learning, the classification risk can be written as*

$$R(g) = \mathbb{E}_X \left[ \sum_{i=1}^{K} \mathcal{P}_{Y=i|X} \ell_i \right] = \mathbb{E}_X \left[ \sum_{i=1}^{K} r_i(X) \ell_i \right].$$

*Proof.* Defining $\bar{L}^\top := L^\top M_{\text{Soft}}^\dagger$ and recalling $\bar{P}$ in (83), we apply (39) to obtain

$$\begin{aligned}
R(g) &= \int_{x \in \mathcal{X}} L^\top P \mathrm{d}x = \int_{x \in \mathcal{X}} \bar{L}^\top \bar{P} \mathrm{d}x = \int_{x \in \mathcal{X}} \sum_{i=1}^{K} \mathcal{P}_{Y=i|X} \ell_i \cdot \mathcal{P}_X \mathrm{d}x \\
&= \mathbb{E}_X \left[ \sum_{i=1}^{K} \mathcal{P}_{Y=i|X} \ell_i \right].
\end{aligned}$$

$\square$

