# OpenReview forum: "Unified Risk Analysis for Weakly Supervised Learning"
_TMLR — Accepted by TMLR_

### Review · Reviewer_vtCp · 2024-11-01

**Summary Of Contributions:**

This paper introduces a unified framework for weakly supervised learning (WSL), utilizing a contamination-based perspective for constructing supervised problems and a decontamination approach for analysis. The framework aims to encompass diverse WSL scenarios under a single methodology, addressing both theoretical consistency and practical applications.

**Audience:**

Yes

**Broader Impact Concerns:**

None.

**Claims And Evidence:**

Yes

**Requested Changes:**

1. The paper introduces a substantial amount of notation and theoretical constructs early on, particularly in Section 3, without sufficiently defining key terms like the data-generating distribution, $P$ and $\bar{P}$. This could make it difficult for readers to follow the framework. I recommend defining these concepts clearly and concisely at the start of Section 3 to improve readability.

2. I suggest that the authors offer more specific guidance on potential future research directions. Rather than presenting this as just a general framework, I think it would be beneficial if the authors gave a few concrete paths forward for both theoretical analysis and practical applications in WSL. For instance, could this framework inspire new types of risk analysis or model improvements?

3. The core idea of using a matrix to correct data labels aligns closely with the noise adaptation layer [R1] proposed by Goldberger et al., where a matrix is also used to modify class labels to handle noise. Could you clarify how your approach differs from or expands upon the noise adaptation layer?

[R1] Jacob Goldberger and Ehud Ben-Reuven. Training deep neural-networks using a noise adaptation layer. ICLR, 2017.

**Strengths And Weaknesses:**

1. The approach of viewing supervised problem formulation as a contamination process and problem analysis as a decontamination process is both novel and conceptually engaging.

2. By providing a structure that covers various WSL settings, the framework supports researchers in analyzing different weak supervision scenarios under a consistent methodology.

3. This framework could be useful in advancing empirical risk minimization across different weakly supervised contexts, potentially reducing the complexity of handling multiple WSL settings separately.

---

> ### Author Response · Authors · 2024-11-17
> **Response to the first requested change from Reviewer vtCp**
>
> We thank the reviewer for providing valuable suggestions to improve the submission. We will respond to each of the reviewer's Requested Changes. Please let us know if you have any additional concerns or suggestions.
>
> > The paper introduces a substantial amount of notation and theoretical constructs early on, particularly in Section 3, without sufficiently defining key terms like the data-generating distribution, $P$ and $\bar{P}$. This could make it difficult for readers to follow the framework. I recommend defining these concepts clearly and concisely at the start of Section 3 to improve readability.
>
> We will revise the main text to improve readability by adding the following paragraph below the first paragraph of Section 3. Some of the overlapped definitions in Sections 3.1 and 3.2 will be removed.
>
> Before introducing the framework, we first define several abstract notations that will be used throughout the paper. There are three main characters in the framework. They are the vector of data-generating distributions $\bar{P}$, the vector of base distributions $B$, and the vector of risk-defining distributions $P$. For completeness, we also denote $L$ the vector of loss functions. Since the proposed framework takes a matrix multiplication approach to formulate a data generation process, we vectorize the data-generating distributions of each scenario listed in Tables 2 and 3 and abstract them using a pseudonym $\bar{P}$. For instance, in PU learning (4), $\bar{P}$ is a two-dimensional vector $(p_{\mathrm{P}}$ $p_{\mathrm{U}})^\top$. The vector of base distributions $B$ consists of distributions of the (ground truth) supervised data. One can think of the entries in $B$ as base elements to form a (observational) data-generating distribution. An entry of $B$ could be of the form $p(Y, X)$ or $p(X|Y)$. $P$ and $L$, on the other hand, are fixed vectors throughout the paper. The $k$-th entry of $P$ is $p(Y=k, X)$ and the $k$-th entry of $L$ is $\ell_Y=k(g(X))$.
>
> (The probability notion $\mathcal{P}$ in the paper is replaced by $p$ because of a bug in OpenReview's rendering system.)

---

> ### Author Response · Authors · 2024-11-17
> **Response to the second requested change from Reviewer vtCp**
>
> > I suggest that the authors offer more specific guidance on potential future research directions. Rather than presenting this as just a general framework, I think it would be beneficial if the authors gave a few concrete paths forward for both theoretical analysis and practical applications in WSL. For instance, could this framework inspire new types of risk analysis or model improvements?
>
> Future directions would emerge from the development of new forms of $\bar{P} = M_{\mathrm{corr}} B$ and $B = M_{\mathrm{trsf}} P$.
> - The current paper only discusses minimizing the classification risk. By modifying M_trsf, the framework can accommodate various performance measures such as the balance error rate, the one-versus-rest risk, and cost-sensitive measures (Brodersen et al., 2010; du Plessis et al., 2014; Menon et al., 2015; Blanchard et al., 2016; Rifkin & Klautau, 2004; Zhang, 2004; Natarajan et al., 2017; Scott & Zhang, 2020).
> - $M_{\mathrm{corr}}$ described in this paper is quite limited. One could leverage cascading matrices to formulate a new learning scenario. For instance, applying a label-flipping matrix followed by a $M_{\mathrm{PU}}$ (50) describes a noisy PU learning scenario. One could also use matrix addition to define conditional contaminations. Moreover, $M_{\mathrm{MCD}}$ (45) modeling a binary classification scenario can be extended to multi-class classification.
> - Unlike works that assume conditional independence (Natarajan et al., 2017; Feng et al., 2020b), $M_{\mathrm{CCN}}$ (60) and $M_{\mathrm{gCCN}}$ (64) take $X$ into account. This degree of freedom allows the study of instance-dependent scenarios and the effect of sampling strategies such as SAR and SCAR (Elkan & Noto, 2008; Coudray et al., 2023).
> - The analysis strategy in Section 5.2, which combines marginal chain and properness, does not rely on matrix inversion. The new technique opens a new possibility to search for non-invertibility based methods for the risk rewrite problem.
>
> In addition to the above potential outlooks, a distinguishing feature of this paper is that it provides a new lens to examine WSL from a different angle. As a result, the proposed framework provides two risk rewrites of MCL, Corollary 40 and Theorem 42, in Section 5.4.2. Furthermore, by comparing the entries between matrices, Lemma 14 can be used to show the connection between MCL, PPL and PCPL. We hope that the matrix-based approach of the framework provides a tool that can break a problem down into components that can be examined more closely and carefully, so that we can gain new insights and understanding of WSL.
>
> References
>
> Coudray, O., Keribin, C., Massart, P., and Pamphile, P. Risk bounds for positive-unlabeled learning under the selected at random assumption. Journal of Machine Learning Research, 24(107):1–31, 2023.
>
> Elkan, C. and Noto, K. Learning classifiers from only positive and unlabeled data. In Proceedings of the 14th ACM SIGKDD International Conference on Knowledge Discovery and Data Mining, pp. 213–220, 2008.
>
> Rifkin, R. and Klautau, A. In defense of one-vs-all classification. Journal of Machine Learning Research, 5:101–141, 2004.
>
> Zhang, T. Statistical analysis of some multi-category large margin classification methods. Journal of Machine Learning Research, 5:1225–1251, 2004.
>
> We will incorporate the response to the concluding section in the final revision.

---

> ### Author Response · Authors · 2024-11-17
> **Response to the third requested change from Reviewer vtCp**
>
> > The core idea of using a matrix to correct data labels aligns closely with the noise adaptation layer [R1] proposed by Goldberger et al., where a matrix is also used to modify class labels to handle noise. Could you clarify how your approach differs from or expands upon the noise adaptation layer?
>
> > [R1] Jacob Goldberger and Ehud Ben-Reuven. Training deep neural-networks using a noise adaptation layer. ICLR, 2017.
>
> Both papers aim at learning a noise matrix that models how a true label $y$ is converted to a noisy label $z$ (here we adopt the notation of [R1]). The difference is how the idea of decontamination is implemented. In our paper, a noise matrix is used to compute a decontamination matrix that aims to cancel out the effect of the noise matrix. This can be considered as a model independent approach. On the other hand, [R1] exploits a strength of the neural network that once the lower labeling layer $w()$ and the upper noise adaptation layer $w_{\mathrm{noise}}()$ are well trained, the labeling layer $w()$ produces the predictions to the true labels.
>
> Mathematically, let $y=w(x)$ be the labeling function and $z=w_{\mathrm{noise}}(y)$ be the noise matrix (i.e., the noise adaptation layer). The idea behind [R1] is that if $z=w_{\mathrm{noise}}(w(x))$ is well trained, then $w(x)$ is the required classifier for $y$. This idea is visualized in Figure 1 of [R1]. In contrast, the analysis component of our framework, discussed in Section 5, shows how to construct a $w_{\mathrm{noise}}^{-1}$ function such that $w_{\mathrm{noise}}^{-1}(w_{\mathrm{noise}}(y)) = y$. From the perspective of the proposed framework, $w_{\mathrm{noise}}^{-1}$ corresponds to a decontamination matrix.
>
> Note that [R1] tested different models for the noise adaptation layer (e.g., the s-model and the c-model). Therefore, one can apply [R1] to solve WSLs by replacing the noise adaptation layer with a weakly supervised adaptation model, where the label transition model can be inspired by a contamination matrix listed in Figure 1 of our paper.

---

> ### Comment · Reviewer_vtCp · 2024-12-28
> **Response to the authors**
>
> Thank you for the detailed responses. Your revisions and clarifications have effectively addressed my concerns.

---

> > ### Author Response · Authors · 2024-12-29
> > **Thank you**
> >
> > Thank you for your response. We will revise the draft to reflect the discussions with the reviewers and the action editor.
> >
> > Best regards,
> >
> > Authors

---

### Review · Reviewer_NSQr · 2024-11-24

**Summary Of Contributions:**

This paper studies weakly supervised learning (WSL). The authors first provide a comprehensive review of many existing binary and multiclass formulations of WSL, and then on top of them they propose a general framework that can capture all different types of WSL, which is depicted in Figure 1.

**Audience:**

Yes

**Broader Impact Concerns:**

No concerns on the ethical implications of the work.

**Claims And Evidence:**

Yes

**Requested Changes:**

I would like to invite the authors to make some changes to the paper in the following aspects.

1. (Weakness 1) The technical proofs of the lemmas and theorems can be simplified or moved to the appendix. Otherwise readers might find it difficult to follow the high-level idea of the paper.

2. (Weakness 2) The motivating questions mentioned on page 2, such as "can a unique interpretation be found to explain the mechanism
behind WSL", as well as the main theoretical techniques used to address them make this paper more like a technical report rather than a journal paper. Could the authors provide more discussions on what insights the proposed framework or the techniques can bring to the community?

3. (Weakness 3) Could the authors provide some discussions on how the proposed framework can make a difference in the machine learning community, from an empirical perspective?

**Strengths And Weaknesses:**

Strengths:

1. This paper provides a good review of existing WSL frameworks.

2. The authors propose a unified framework for many formulations of WSL, which helps readers better understand that there is a common way to express WSL problems and solve them.

Weaknesses:

1. The main body of this paper is too heavy for readers to follow.

2. The main techniques (matrix analysis, combinatorics, etc) used in this paper are quite standard.

3. Although the authors provide a good way to interpret the formulation of WSL, it is unclear to the readers why/how it is important to the community from an empirical perspective.

---

> ### Author Response · Authors · 2024-12-11
> **Response to the first requested change from Reviewer NSQr**
>
> We appreciate your comments on how to improve the submission. We will respond to each of the reviewer's Requested Changes. Please let us know if you have any additional concerns or suggestions.
>
> > (Weakness 1) The technical proofs of the lemmas and theorems can be simplified or moved to the appendix. Otherwise readers might find it difficult to follow the high-level idea of the paper.
>
> On submission, we thought about readability and moved proofs to Appendices B and C. These are proofs using similar techniques or ideas. Several proofs of technical lemmas have also been moved to the appendix. The remaining proofs in the main text fall into one of three main categories: (a) explaining the application of the framework, (b) new proof techniques, and (c) having referred equations for later discussion. We provide a list of the proofs in the main text and briefly explain their significance in the last paragraph of this response (please see (*1)).
>
> We conjecture the difficulty comes from the section organization, for which we suggest the following to provide a more comfortable readability:
>
> - (1) Incorporate the following clarification into the last paragraph of Section 1.
>
> The current organization of Sections 4 and 5 aims at connecting multiple WSL scenarios under one framework. We remark that this paper can serve multiple purposes for the study of WSLs. A summary of possible use cases of the paper is provided in Appendix A.
>
> - (2) Add a new section to the appendix.
>
> Appendix A. Ways to Use the Paper
>
> There are several ways to use this paper in the study of WSLs. We provide some common use cases as follows:
>
> Use Case 1: If you want a quick overview of WSL. Reading Section 2 would provide a quick catch-up on the formulations and results of various WSL scenarios.
>
> Use Case 2: If you want to get the high-level idea of this paper. One could start with Section 3, which provides the abstract form of the proposed framework. Then, sections 4.1.1 and 5.1.1 give the practical application of the framework to UU learning. If you are more familiar with PU learning, then you are welcome to proceed to Sections 4.1.2 and 5.1.2 to see how our framework interprets PU learning. In addition, these reading steps will reveal the connection between UU and PU.
>
> Use Case 3: If you want to know how to apply the framework to a specific scenario. One could read Sections 4.a.b and 5.a.b at the same time and come back to the rest of Sections 4 and 5 only as needed. Note that “a.b” represents the index of a subsubsection.
>
> Use Case 4: If you want to know the connections discovered and the analysis techniques developed in this paper. Section 4 provides detailed discussions of the formulations, and Figure 1 summarizes the relationship. Section 5 provides detailed explanations on how our framework is applied to rewrite the classification risk. The extensive analyses from Section 5 are summarized in Table 10.
>
> ***
> (*1) Crucial proofs in the main text
> - (a) Results showing subtle adjustments to the application of the framework and results appear for the first time: Propositions 1 and 2; proofs in Sections 4.1.1, 4.1.2, 4.2.1, and 4.3.1; proofs in Sections 5.1.1, 5.2.1, 5.2.2, and 5.3.1.
> - (b) Derivations of new proof technique(s): Proofs in Sections 4.1.6 (Lemma 10), 4.2.4 (Lemma 14), 5.1.3 (Lemma 24), 5.1.6 (Theorem 34), and 5.2.4 (Lemma 42).
> - (c) Proofs with referenced equation(s): Lemma 10 and equation (88) in the proof of Theorem 21.

---

> ### Author Response · Authors · 2024-12-11
> **Response to the second requested change from Reviewer NSQr**
>
> > (Weakness 2) The motivating questions mentioned on page 2, such as "can a unique interpretation be found to explain the mechanism behind WSL", as well as the main theoretical techniques used to address them make this paper more like a technical report rather than a journal paper. Could the authors provide more discussions on what insights the proposed framework or the techniques can bring to the community?
>
> The submission generates two insights.
>
> The first is that seemingly different WSL formulations actually share the same data generation idea. In particular, the comprehensive relationship graph shown in Figure 1 subsumes WSLs under a unique formulation framework and reveals connections between scenarios that were previously unknown to the community. Figure 1 is our answer to the research question “From a formulation perspective, can a unique interpretation be found to explain the mechanism behind WSL?”
>
> The second insight is that the derivations of different risk rewrites, which previously seemed irrelevant, can indeed be systematically analyzed. This insight corresponds to the general strategy proposed in Section 3.2. We also illustrate the subtle adjustments to develop simplified and intuitive proofs for the existing risk rewrites in Section 5. The extensive alternative proofs are summarized in Table 10, which answers our third research question “Does a methodology exist to address as many WSLs as possible?” In terms of technical novelty, to the best of our knowledge, the marginal chain method is the first loss correction method developed without the need for the invertibility assumption.

---

> ### Author Response · Authors · 2024-12-11
> **Response to the third requested change from Reviewer NSQr**
>
> > (Weakness 3) Could the authors provide some discussions on how the proposed framework can make a difference in the machine learning community, from an empirical perspective?
>
> Thank you for pointing out the importance of practical applicability. Although the paper only unifies WSLs down to the risk rewrite level, it could be beneficial when constructing a practical training objective from a risk rewrite. Since the analyses of different scenarios are subsumed under a single framework, we now have a basis for transferring a technique developed for one scenario to another. In addition, the alternative proofs provide different ways of dissecting the risk, so that new training objectives could be constructed by examining respective parts of a different risk decomposition.
>
> We also responded to another reviewer with several outlooks and insights. Please refer to the reply titled “[Response to the second requested change from Reviewer vtCp][1]” for more details.
>
> [1]: https://openreview.net/forum?id=RGsdAwWuu6&noteId=NGltcVFSfJ

---

> > ### Comment · Action_Editor_Rg3A · 2024-12-25
> > **Suggestions for the appendix**
> >
> > Dear Reviewer NSQr,
> >
> > Could you clarify whether the proposed changes by the authors would constitute an acceptable resolution of your issues, or whether any further modifications are necessary?
> >
> > In particular, would the incorporation of a map of the paper at the beginning of the appendix as the authors propose be sufficient, or would a more drastic reorganisation of the sections be necessary?
> >
> >
> >
> > Best wishes,
> >
> > AE

---

> > > ### Comment · Reviewer_NSQr · 2024-12-28
> > > **Response to the Action Editor**
> > >
> > > Dear Action Editor,
> > >
> > > I have read the response and think the suggestions proposed by the authors are acceptable. I agree that moving most proofs to the appendix seems not reasonable. Adding a section for 'Ways to Use the Paper' would be great for the readers.
> > >
> > > I am also satisfied with the authors' response on the theoretical insights and practical applicability.
> > >
> > > Best,
> > > Reviewer NSQr

---

> > ### Comment · Reviewer_NSQr · 2024-12-28
> > **Response to the authors**
> >
> > Dear authors,
> >
> > Thank you for your efforts in the response. I think adding a section for 'Ways to Use the Paper' is reasonable. I'm also satisfied with the response on the theoretical insights and practical applicability. Please incorporate these discussions in the revision.
> >
> > Best,
> > Reviewer NSQr

---

> > > ### Author Response · Authors · 2024-12-29
> > > **Thank you**
> > >
> > > Thank you for your response. We will revise the draft to reflect the discussions with the reviewers and the action editor.
> > >
> > > Best regards,
> > >
> > > Authors

---

> ### Comment · Action_Editor_Rg3A · 2025-01-09
> **Official recommendation (deadline yesterday AoE)**
>
> Dear Reviewer NSQr,
>
>
> Many thanks for your help carefully reviewing the paper and interacting with the authors during the rebuttal phase.
>
> Please submit your official recommendation in the system when you can, as this is necessary for me to be able to process the article with a decision.
>
>
> Best regards,
>
> AE

---

### Review · Reviewer_DJLT · 2024-12-11

**Summary Of Contributions:**

This paper presents a unified risk analysis for weakly supervised learning (WSL). The framework involves a formulation component and an analysis component. The former provides a unifying interpretation for WSL data generation process, and produces three reduction graphs to show the connection between different WSL formulations. The latter uses the decontamination concept to build a general method to conduct risk rewrites for different WSLs. The paper presents theoretical analysis to show that the risk rewrites in the framework recover existing results.

**Audience:**

Yes

**Broader Impact Concerns:**

No broader impact concerns.

**Claims And Evidence:**

Yes

**Requested Changes:**

The paper includes some proofs in the main text. To further improve the organization, it may be better to include all the proofs in the appendix.

Page 7: it seems that the definition of $P_{sup}$ and $P_{inf}$ are the same as both $x$ and $x'$ play the same role. The authors should check it.

Page 64： there are several missing periods at the end of equations

Page 74: "as follows." should be "as follows"

**Strengths And Weaknesses:**

Strength:

- The paper presents the first framework to show the connection between different WSL methods. This approach provides an insightful viewpoint to study different WSLs and understand the underlying mechanisms.
- The developed framework is powerful and recovers existing results.

Weakness:

- The paper only considers the classification risk as the performance measure. As a comparison, existing methods consider various performance measures such as the balanced error rate, the receiver operating characteristic curve and the cost-sensitive measures.
- It is not clear to me whether the proposed framework also implies new WSL methods that can outperform the existing WSL methods in terms of accuracy and scalability.
- For multi-class classification problems, page 57 shows that one needs to compute a K by K matrix, and therefore it does not apply to problems with a large number of classes.

---

> ### Author Response · Authors · 2024-12-12
> **Response to the weaknesses from Reviewer DJLT**
>
> Thank you for the comments for improving the submission. We are addressing your concerns as follows. If you have additional suggestions or concerns, please let us know.
>
> > Weakness 1: The paper only considers the classification risk as the performance measure. As a comparison, existing methods consider various performance measures such as the balanced error rate, the receiver operating characteristic curve and the cost-sensitive measures.
>
> Thank you for pointing out the choice of performance metrics. The primary goal of this submission is to unify WSLs. Thus, we choose the most common metric, classification risk, to communicate our discovery. We address the adaptation of our framework to other performance metrics in Section 6 (the second and third sentences on page 57).
>
> > Weakness 2: It is not clear to me whether the proposed framework also implies new WSL methods that can outperform the existing WSL methods in terms of accuracy and scalability.
>
> We would not choose the word “implication” to justify this submission as the current submission does not provide constructive guidance on accuracy and scalability. However, it does open up several new ways of approaching WSL. In short, they are modeling a complex system, devising an invertibility-free approach, technique transfer, and risk decomposition. Please refer to the responses “[Response to the second requested change from Reviewer vtCp][1]” and “[Response to the third requested change from Reviewer NSQr][2]” for further elaboration.
>
> [1]: https://openreview.net/forum?id=RGsdAwWuu6&noteId=NGltcVFSfJ
> [2]: https://openreview.net/forum?id=RGsdAwWuu6&noteId=mcsnpMPast
> > Weakness 3: For multi-class classification problems, page 57 shows that one needs to compute a K by K matrix, and therefore it does not apply to problems with a large number of classes.
>
> No, the current paper does not scale. To be consistent with our goal of finding a methodology, we assume that all requested parameters are given (the last sentence in the first paragraph of Section 5). However, if we are given problem-specific knowledge (e.g., structural information, clusters, heavy-tailed property, etc.), it is possible to decompose a large $K$ by $K$ matrix into smaller ones to address the scalability challenge.

---

> ### Author Response · Authors · 2024-12-12
> **Response to the requested changes from Reviewer DJLT**
>
> > The paper includes some proofs in the main text. To further improve the organization, it may be better to include all the proofs in the appendix.
>
> The proofs are kept in the main text because (1) they contain subtle adjustments to illustrate the applicability of the framework, (2) they are new proof techniques, or (3) they have an informative equation that is referred to in a later discussion. To address the concern about the organization, we propose adding an appendix section to illustrate several use cases of this paper. We have a response to Reviewer NSQr regarding this concern. Please see the reply titled “[Response to the first requested change from Reviewer NSQr][1]” for more details.
>
> [1]: https://openreview.net/forum?id=RGsdAwWuu6&noteId=HzTYZ8koud
>
> > Page 7: it seems that the definition of $P_{sup}$ and $P_{inf}$ are the same as both x and x’ play the same role. The authors should check it.
>
> We reviewed the draft again. $P_{sup}$ and $P_{inf}$ are different. Although $x$ and $x’$ play the same role, $(x, x’)$ and $(x’, x)$ have different meanings. In the Pcomp formulation, we always place the instance that is more likely to be labeled positive on the left entry of the pair. We also checked with the original paper and confirmed that $P_{sup}$ and $P_{inf}$ at the bottom of page 21 are the same as those derived in Theorem 2 of (Feng et al., 2021) (with notation substitution).
>
> > Page 64： there are several missing periods at the end of equations
>
> > Page 74: "as follows." should be "as follows"
>
> Thank you for pointing out the typos. We have made the corrections.

---

> > ### Comment · Reviewer_DJLT · 2024-12-28
> > **Thanks for your response**
> >
> > Thanks for the response to my comments. I am satisfied with these response. I also agree to keep the proof in the maintext as this is better to illustrate the unifying framework.

---

> > > ### Author Response · Authors · 2024-12-29
> > > **Thank you**
> > >
> > > Thank you for your reply. We will revise the draft accordingly in the revision.
> > >
> > > Best regards,
> > >
> > > Authors

---

### Comment · Action_Editor_Rg3A · 2024-12-25
**Discussion**

Dear Reviewers,


Please make sure to read the rebuttal by the authors and respond with any additional questions or requests for the revision.

Best wishes,

AE

---

### Author Response · Authors · 2025-01-02
**A draft revision is uploaded**

Dear Reviewers and Action Editor,

We have revised the submission according to the discussions and re-uploaded the revision. Modified parts are highlighted in blue. For section-wide changes, we color only the section title. The changed parts are summarized below:

- Page 2
  - Reviewer NSQr: Insights.

- Page 3 + Appendix A (page 65)
  - Reviewer NSQr: Use cases.

- Pages 14–15
  - Reviewer vtCp: Definitions of $\bar{P}$, $P$ and $B$
  - Action Editor Rg3A: The distinction between $P$ and $B$. The role of $M_{trsf}$.

- Section 6 (pages 56–57)
  - Action Editor Rg3A: Question 3 on applying the framework.

- Pages 58–59
  - Reviewer vtCp, NSQr, and DJLT: Applications and future directions.

Please let us know if the revision meets your requirements or not. Thank you very much.

Best regards,

Authors

---

> ### Comment · Action_Editor_Rg3A · 2025-01-06
>
> Dear Authors,
>
>
> Thanks for the summary and the additional details. I am quite happy with the new calculations on pages 56 and 57.
>
> Best regards,
>
>
> AE
>
>
> Dear Reviewer NSQr,
>
>
> Please check the updated pdf to see whether it complies with your requests and engage with the authors if necessary before submitting your recommendation.
>
> Best regards,
>
>
> AE

---

### Decision · Action_Editor_Rg3A · 2025-01-09

**Recommendation:** Accept as is

**Comment:**

This paper introduces a unifying framework for many weakly supervised learning scenarios such as Positive Labeled (PL) or Unlabeled-Unlabeled (UU) learning. The core idea is to associate each learning setting with a transformation matrix which transforms the 'vector' of class-wise distributions into the 'vector' of distributions associated with each partial labels. Whilst the paper is relatively long, the flow is reasonably natural.

Whilst the truly new results for specific WSL settings in the present submission are limited to risk rewrites (which might explain why the reviewers unanimously oppose acceptance at the ICLR journal to conference track), I believe the unifying formulation introduced in this paper has the potential to become an established standard (which explains the unanimous decision to accept). Further down the road, this could help prove deeper results about and draw connections between various WSL settings and help positively shape the direction of future research. Overall, this is an excellent paper and contribution to the community. Thus, I am happy to recommend a survey certification in addition to the acceptance decision.

**Audience:**

This topic falls within the area of (broader) learning theory. It is certainly on topic for TMLR's audience.

**Claims And Evidence:**

The statements are accompanied by rigorous proofs, and include the main ideas in the main paper.

---

> ### Author Response · Authors · 2025-01-16
> **Thank you**
>
> Dear Action Editor and Reviewers,
>
> We would like to thank you for your time, valuable comments, and constructive suggestions for improving the submission. We appreciate the acceptance and certification.
>
> Best regards,
>
> Authors